# Impact of wind profiles on ground-generation airborne wind energy system performance

Markus Sommerfeld[1], Martin Dörenkämper[2], Jochem De Schutter[3], and Curran Crawford[1]

[1]Institute for Integrated Energy Systems, University of Victoria, British Columbia, Canada
[2]Fraunhofer Institute for Wind Energy Systems (IWES), Oldenburg, Germany
[3]Systems Control and Optimization Laboratory IMTEK, University of Freiburg, Germany

**Correspondence:** Markus Sommerfeld (msommerf@uvic.ca)

**Abstract.** This study investigates the performance of pumping mode ground-generation airborne wind energy systems(AWESs) by determining cyclical, feasible, power-optimal flight trajectories based on realistic vertical wind velocity profiles. These 10-minute profiles, derived from mesoscale weather simulations at an offshore and an onshore site in Europe, are incorporated into an optimal control model that maximizes average cycle power by optimizing the trajectory. To reduce the computational cost, representative wind conditions are determined based on k-means clustering. The results describe the influence of wind speed magnitude and profile shape on the power, tether tension, tether reeling speed and kite trajectory during a pumping cycle. The effect of mesoscale-simulated wind profiles on power curves is illustrated by comparing them to logarithmic wind profiles. Offshore, the results are in good agreement, while onshore power curves differ due to more frequent non-monotonic wind conditions. Results are references against a simplified quasi-steady-state model and wind turbine model. This study investigates how power curves based on mesoscale-simulated wind profiles are affected by the choice of reference height. Our data shows that optimal operating heights are generally below 400 m with most AWESs operating at around 200 m.

## 1 Introduction

Airborne wind energy systems (AWESs) aspire to harvest stronger and less turbulent winds at mid-altitude, here defined as heights between 100 and 1000 m, presumably beyond what is achievable with conventional wind turbines (WTs). The prospects of higher energy yield combined with reduced capital cost motivate the development of this new class of renewable energy technology (Lunney et al., 2017; Fagiano and Milanese, 2012). Unlike conventional WTs, which in recent decades have converged to a single concept with three blades and a conical tower, several different AWES concepts and designs are being investigated by numerous companies and research institutes (Cherubini et al., 2015; Vermillion et al., 2021; Fagiano et al., 2022). These kite-inspired systems consist of three main components: one or more tethered aircraft or kites, one or more ground stations, and one or more tethers to connect the flying components to the ground. This study focuses on the two-phase cyclic, ground-generation concept, also referred to as pumping mode (Luchsinger, 2013). During the reel-out or production phase, the kite pulls a tether from a drum on the ground, which is connected to a generator, thereby producing electricity. This is followed by the reel-in phase during which the kite returns to its initial position and reduces its aerodynamic forces in order to de-power. There are several ways to reduce the aerodynamic forces on a kite, such as adjusting its angle of attack or flying it out of the

wind window. Various other concepts such as fly-gen, aerostat, or rotary lift are not considered in this study (Cherubini et al., 2015). Since this technology is still in a relatively early stage of development, validation and comparison of power production estimates is difficult. Several studies (van der Vlugt et al., 2013; Schelbergen et al., 2020; Vander Lind, 2013; Ranneberg et al., 2018) compared computed power curves with experimental performance data. At present, there is no standardized power curve definition or reference design that would allow for comparison between different concepts and conventional wind turbines.

However, the goal of Eijkelhof and Schmehl (2022); Eijkelhof et al. (2020) was to create a reference design for a multi-MW AWES. It is not the goal of this study to determine a general power curve, but rather to investigate the power variation of a specific design derived from realistic wind profiles.

Recent consensus among the scientific community defined a power curve as the maximum average cycle power, which is the combination of a consecutive reel-out and reel-in phases, as a function of wind speed at pattern height, which is the time-

averaged height during the reel-out power-producing phase (Airborne Wind Europe, 2021). Together with the site-specific wind resource, wind park planners and manufacturers can use power curves to estimate annual energy production (AEP), which can be combined with a cost model to determine the levelized cost of electricity (LCOE) and financial viability (Malz et al., 2020a). Unlike conventional WTs, where the wind speed probability distribution at hub height is used to determine AEP, AWES continuously change their operating height, making it difficult to determine AEP with this approach. AWES

performance highly dependents on the shape and magnitude of the wind speed profile over the operating height range. Using simple wind profile approximations, such as logarithmic or exponential wind speed profiles, can provide an estimate of long-term average conditions. However, these approximations cannot account for the wide range of profile shapes that occur over short periods of time or changes that occur on a daily or seasonal basis (Emeis, 2013). This can reduce the accuracy of the predicted power output. However, such wind speed profiles can be employed to estimate average performance and are the

standard in most AWES power estimation studies.

In their study, van der Vlugt et al. (2013) described TU Delft's 20 kW inflatable wing technology demonstrator and compared a statistically derived power curve to results from a theoretical performance model. The wind and power models used in this study were taken from Fechner and Schmehl (2013). The wind model is based on a standard exponential wind speed profile approximation, while the power model uses a multi-phase QSM. A follow up study (van der Vlugt et al., 2019), added more

detail to specific cycle trajectories. Heilmann and Houle (2013) used exponential wind speed profiles and a standard Rayleigh distribution to estimate performance and cost. Their power curve is modeled based on a QSM by Luchsinger (2013) using the averaged flight path height of the kite as wind speed reference height. An LCOE between 40 and 110 Euro/MWh was estimated for different annual average wind speeds. Faggiani and Schmehl (2018) used a similar pumping-mode QSM to estimate power output based on wind speed at the operating height of the kite. They developed a cost model to estimate the achievable LCOE

of an entire kite wind farm. Their analyses showed that the cost of energy decreases and the quality of the electrical power increases with increasing number of kites. Ranneberg et al. (2018) studied the performance of a soft kite pumping mode AWES by determining its power curves at various reference heights for different logarithmic wind profiles. The study found that the yield variation for these logarithmic wind profiles was quite small. Additionally, the yield for a specific site was estimated using detailed wind speed profiles from COSMO-DE and the results were found to be consistent with more detailed simulations of

the Enerk'ite EK30 prototype (EnerKíte GmbH, 2022). Leuthold et al. (2018) investigated power-optimal trajectories of a ground-generation multikite configuration for a range of logarithmic wind speed profiles. Three distinct operational regions were identified: Region I where power is used to maintain altitude, Region II where power harvesting increases up to design wind speed, and Region III where power extraction is intentionally limited due to the physical constraints of the system. Licitra et al. (2019) estimated the performance and power curve of a fixed-kite ground-generation AWES by generating power-optimal trajectories using a power law approximation of the wind speed profile. The results were validated against data from Ampyx Power AP2 (Licitra, 2018; Malz et al., 2019; Ampyx Power BV, 2020). Eijkelhof and Schmehl (2022) found that mass had a detrimental effect on power-optimal trajectories for large-scale single-kite fixed-wing AWES. To determine power curves, the authors used normalized average offshore wind speed profiles from the Ijmuiden measurement tower. Sommerfeld et al. (2022) used the same methodology and wind data as in this study to examine the effects of size scaling and improvements in aerodynamic efficiency on a single-kite fixed-wing reference system. The authors found that it is likely better to deploy multiple smaller-scale devices rather than a single large-scale system, because of negative mass scaling effects. De Schutter et al. (2019) analyzed the performance of utility scale, stacked multikite systems using logarithmic wind speed profiles as boundary conditions for a nonlinear optimization problem. The authors used the same optimization framework as in the present investigation. They found that this multikite strategy could make power generation largely independent wing size. Malz et al. (2020b) efficiently estimated the performance of single-kite drag-mode AWES for large wind data sets by combining an optimal control performance model with smart initialization and machine learning. Wind speed profiles from the MERRA-2 reanalysis model (Gelaro et al., 2017) were clustered into characteristic profile shapes and interpolated using Lagrangian polynomials (Section 4.4). The authors showed that ordering the wind parameters by the wind speed at the average operational height (300 m) significantly reduced the computation time. Aull et al. (2020) explored the design and sizing of fly-gen rigid-kite systems based on a steady-state model with simple aerodynamic and mass-scaling approximations. At each scale, the relationships between size, efficiency, power output, and cost were determined. The wind resource was described by an exponential wind shear model with a Weibull distribution. The main conclusion was that physics and economics favor a larger number of small units. Bechtle et al. (2019) used ERA5 reanalysis data to assess wind resources at high altitudes throughout Europe. They described the available wind energy without considering a specific conversion mechanism and included a description of wind speed and probability at various heights. The effect of variable height harvesting was demonstrated for a location in the English Channel. Schelbergen et al. (2020) proposed a clustering procedure to obtain wind statistics from the Dutch Offshore Wind Atlas (DOWA) data set. Principal component analysis and k-means clustering were used to determine representative wind speed profile shapes. To estimate the AEP of a small-scale pumping AWES located at Cabauw in the center of the Netherlands, several power curves were derived for each wind speed profile shape using a flexible-kite, pumping mode QSM developed by van der Vlugt et al. (2019). Faggiani and Schmehl (2018) studied the economic impact of various design aspects of wind parks, including the spatial stacking of systems, the number of units, the size of kites, and phase-shifted operation. The performance of the system was estimated using a QSM developed by Schmehl et al. (2013); van der Vlugt et al. (2019), assuming a range of wind speeds at the operating height of the kite. The AEP LCOE could be assessed by combining a detailed cost model with an assumed Weibull probability distribution. The study found that increasing the number of kites had several scale effects,

such as decreasing the cost of energy and increasing the quality of electrical power. Wind speed profiles are governed by weather phenomena, environmental and location-dependent conditions on a multitude of temporal and spatial scales. The preferred means of determining wind conditions for wind energy converters are long-term, high-resolution measurements. At mid-altitudes, these measurements can only be obtained through long-range remote sensing methods such as LiDAR (light detection and ranging) or SoDAR (sonic detection and ranging). Measuring wind conditions at mid-altitudes is costly and difficult, due to reduced data availability (Sommerfeld et al., 2019a). Additionally, publicly available measurements are scarce. Therefore, wind data in this study is derived from Weather Research and Forecasting model (WRF) mesoscale simulations (Skamarock et al., 2008). However, the described trajectory optimization methodology can be applied to any wind data set such as wind atlas data or measurements. Numerical mesoscale weather prediction models such as WRF, which is well known for conventional WT siting applications (Salvação and Guedes Soares, 2018; Dörenkämper et al., 2020), are used to estimate wind conditions on time scales of a few minutes to years. Sommerfeld et al. (2019b) compared the simulated onshore data used in this study, located in northern Germany near the city of Pritzwalk, to LiDAR measurements and found a good agreement between both data sets. Data from the FINO3 research platform in the North Sea can be used as a reference for the simulated offshore conditions in this study. The present study investigates the performance of AWES subject to 10-minute average wind data, which is the standard for conventional WT, while the New European Wind Atlas (NEWA) only provides 30-minute average data (Witha et al., 2019). We use this higher-resolution wind data because the higher temporal, spatial, and vertical resolution reduces averaging and allows for the investigation of more realistic wind conditions.

This paper's main contribution is the examination of how realistic onshore and offshore wind profiles, compared to a standard log profile, affect the power-optimal performance of AWES, and how the choice of reference height impacts the power curve, particularly given the wide range of wind speed profile shapes. This study is a continuation of previous analyses of LiDAR measurements (Sommerfeld et al., 2019a) and WRF simulations (Sommerfeld et al., 2019b) at the onshore location. To demonstrate the validity and applicability of the data, several wind characteristics are described. These include annual wind speed probability distributions up to 1000 m, annual wind direction statistics and wind speed profile shapes. The data are categorized using k-means clustering (Lloyd, 1982; Hartigan and Wong, 1979) which classifies the wind data at each location into groups of similar wind speed and vertical profile shape. Three representative 10-minute wind velocity vectors $\mathbf{U}$ are sampled from each of these 20 clusters (total 60 vectors out of 52,560 wind data points per location) and serve as boundary conditions for the awebox trajectory optimization (De Schutter et al., 2020). The profiles with the 5th, 50th, and 95th percentile of wind speed at an a priori guess of the pattern trajectory height $z_{\mathrm{pth}} \approx 100\,\mathrm{m} \leq z \leq 400\,\mathrm{m}$ (Airborne Wind Europe, 2021) are selected because they encompass the most probable wind conditions within each cluster while excluding non-representative extremes. Subsection 6.3 verifies that choice and compares the impact of reference height on the power curve. This drastically reduces the computational cost as only few selected profiles are used to represent the entire spectrum with sufficient resolution to investigate the variation of power.

The awebox optimization model allows for the investigation of dynamic performance parameters, such as aircraft trajectories, tether tension, tether reeling speed and power which highly depend on the wind conditions. The aircraft model is based on the well investigated and published Ampyx Power AP2 prototype (Licitra, 2018; Malz et al., 2019; Ampyx Power BV, 2020),

adjusted to a projected surface area of $A = 20$ m$^2$ to generate results for more realistic and probable devices. The maximized power curves, estimated based on average cycle powers and wind speed at reference height $\overline{U}_{\mathrm{ref}} = U(z_{\mathrm{ref}})$, are compared to performance predictions using a simple AWES QSM and a steady-state WT model. The variation in performance caused by realistic wind data is referenced against predictions based on simple logarithmic wind speed profiles. The a priori guess of $100$ m $\leq z_{\mathrm{ref}} \leq 400$ m is confirmed by comparing the impact of different reference heights on the power curves.

The structure of this research is as follows. Section 2 introduces the mesoscale WRF model, analyzes the offshore and on-shore wind resource, introduces the k-means clustering algorithm and summarizes the results of the clustered wind vectors. Section 3 introduces the dynamic AWES model, which includes an aircraft and tether model as well as ground station con-straints. Section 4 describes the `awebox` optimization toolbox, summarizes the aircraft parameters, system constraints, and initial conditions. This is followed by a description of the WT and AWES reference models in Section 5. The results presented
in Section 6 include flight trajectories and time series data for various performance parameters, as well as a statistical analysis of the tether length, operating altitude, and power curve estimations. Section 7 summarizes the findings and concludes with an outlook and motivation for future work.

## 2   Wind conditions

Subsection 2.1 introduces the model and setup of the onshore and offshore mesoscale WRF simulations. Subsection 2.2 ana-
lyzes wind statistics to give an insight into the wind regime at both locations. Clustering, which is introduced in Subsection 2.3, is used to identify groups of similar vertical wind profiles and to select representative profiles from these groups. This sig-nificantly reduces the computational cost as only few selected profiles are necessary to represent the wind regime. Subsections 2.4 and 2.5 describe the resulting clusters and their statistical correlation with temporal and meteorological phenomena.

### 2.1   Mesoscale simulations

This study compares AWES performance for two specific onshore and offshore locations in Europe (Figure 1). Wind condi-tions for the chosen years are assumed to be representative of these locations. However, the wind data has not been compared to long-term wind atlas data and has not been corrected using long-term simulations. The onshore data represent wind con-ditions at the Pritzwalk Sommersberg airport (lat: $53°10'47.00''$N, long: $12°11'20.98''$E) in northern Germany and comprise 12 months of WRF simulation data between September 2015 and September 2016. The area surrounding the airport consists
mainly of flat agricultural land with the town of Pritzwalk in the south and is a suitable location for the generation of wind energy (Sommerfeld et al., 2019a, b). The FINO3 research platform in the North Sea (lat: $55°11,7'$N, long: $7°9,5'$ E) was chosen as a representative offshore location due to its proximity to several offshore wind farms and the amount of compre-hensive reference measurements (Peña et al., 2015). The offshore simulation covers the time frame between September 2013 and September 2014. The mesoscale simulations used the Weather Research and Forecasting (WRF) model (Powers et al.,
2023). The onshore simulation was performed with version 3.6.1 (Skamarock et al., 2008) prior to the 2018 release of WRF version 4.0.2 (Skamarock et al., 2021), in which offshore simulations were computed. The setup of the model was adapted

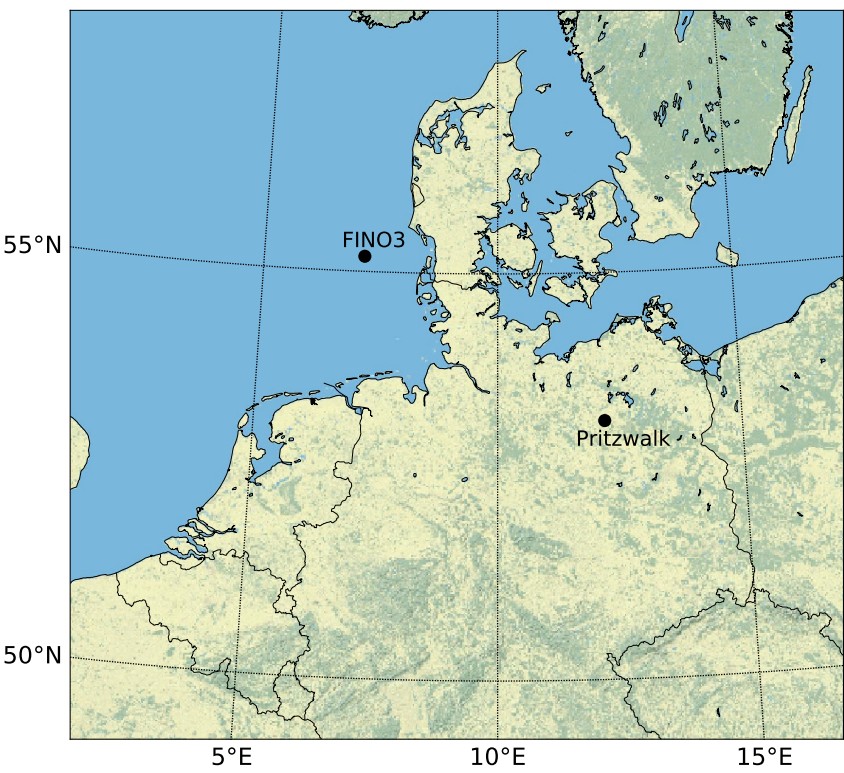

**Figure 1.** Map of northern Germany with the representative onshore (Pritzwalk) and offshore (FINO3) locations highlighted by black dots.

and constantly improved for wind energy applications by the authors of the present manuscript within for various projects and applications in recent years (Dörenkämper et al., 2015, 2017; Dörenkämper et al., 2020; Hahmann et al., 2020; Sommerfeld et al., 2019b). The focus of this study is not on the detailed comparison between mesoscale models, but on AWES performance
subject to realistic onshore and offshore wind conditions. Both WRF models provide adequate temporal and spatial resolution for preliminary performance assessment, even though the setup and time frame are different.

    Each simulation consists of three nested domains around their respective location (black dot Figure 1). The innermost domain (D03), with the finest resolution, is nested within the middle domain (D02), which in turn is nested within the outermost domain (D01) with the coarsest resolution. The simulations use one-way nesting where the outer domains define the boundary
conditions for the inner domains. Atmospheric boundary conditions are defined by ERA-Interim (Dee et al., 2011) for the onshore location and by ERA5 (Hersbach and Dick, 2016) reanalysis data for the offshore location, while sea surface parameters (such as sea surface temperature and sea ice analysis) for the offshore location are based on OSTIA (Donlon et al., 2012). These

data sets have proven to provide good results for wind energy-relevant heights and sites (Olauson, 2018; Hahmann et al., 2020).

Four-dimensional data assimilation (FDDA), also known as analysis nudging, nudges the simulation of the outer domain towards the atmospheric boundary conditions throughout the simulation time, to reduce numerical drifting and provide smoother boundary conditions. Both simulations use the Mellor–Yamada–Nakanishi–Niino (MYNN) 2.5 (Nakanishi and Niino, 2009) level scheme for the planetary boundary layer (PBL) physics. The onshore simulation was carried out in a single 12-month simulation run from 2015-09-01 to 2016-08-31. The offshore simulation period covered a 410 day period from 2013-08-30 to 2014-10-14 and was divided into 41 simulations of 10 days each, with an additional 24 hours of spin-up time per run. Spin-up is the period in which the model produces results that may not be reliable due to initialization using coarser global atmospheric reanalysis data. WRF calculates the vertical coordinate using a hybrid hydrostatic pressure coordinate, which is a function of surface and atmospheric pressure (Skamarock et al., 2021). The data at each vertical, terrain-following pressure coordinate (sigma level) is converted to geometric heights using the postprocessing methodology described by Dörenkämper et al. (2020). Table 1 summarizes the key model settings used in this study. All simulations were performed on the *EDDY* High-Performance Computing clusters at the University of Oldenburg (Carl von Ossietzky Universität Oldenburg, 2018).

**Table 1.** Key setup parameters of the onshore and offshore mesoscale WRF simulations to generate the wind data used in this study.

| Model Parameter | Settings | |
|---|---|---|
| | Onshore | Offshore |
| WRF model version | 3.5.1 | 4.0.2 |
| Time period | 2015-09-01 to 2016-08-31 | 2013-08-30 to 2014-10-14 |
| Reanalysis data set | ERA-Interim | ERA5 & OSTIA |
| Horizontal grid size (D01, D02, D03) | 120 ×120, 121 ×121, 121 ×121 | 150 ×150, 151 ×151, 151 ×151 |
| Horizontal Resolution (D01, D02, D03) | 27 km, 9 km, 3 km | 18 km, 6 km, 2 km |
| Vertical grid levels | 60 sigma levels (about 25 below 2 km) | 60 sigma levels (about 25 below 2 km) |
| Nesting | one-way | one-way |
| Initialisation strategy | single run | 240 h runs plus 24 h spin-up time |
| Nudging | Analysis nudging (FDDA) | Analysis nudging (FDDA) |
| PBL scheme | MYNN level 2.5 | MYNN level 2.5 |

## 2.2 Wind regime

Figure 2 depicts the wind roses of the computed annual wind conditions at 100 m (a, b) and 500 m (c, d) height onshore (left) and offshore (right). The dominant wind direction at both locations is southwest, turning clockwise with increasing altitude.

Directional variability decreases and the wind speed $U$, which is the magnitude of the wind vector $\mathbf{U}$ increases with height, following the expected trends in the northern hemisphere (Arya and Holton, 2001; Stull, 1988). The average onshore wind direction turns about $14°$ between 100 and 500 m, whereas average offshore wind direction only veers approximately $5°$. The

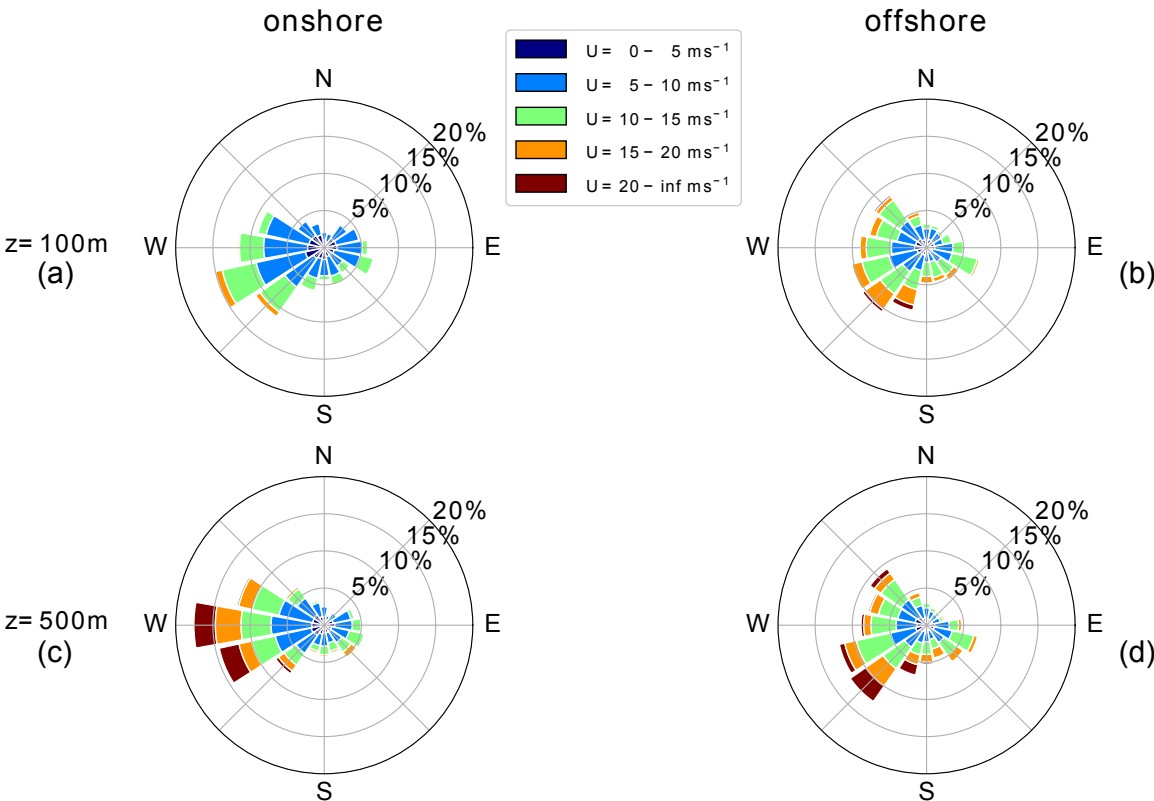

**Figure 2.** Wind roses of annual wind direction and speed statistics at Pritzwalk (onshore) and FINO3 (offshore) for heights of 100 and 500 m during the simulated years.

offshore wind direction turns approximately $10°$ additional degrees above 500 m, resulting in roughly the same westerly wind direction at around 1000 m. Due to the prevailing unstable conditions offshore accompanied by strong vertical mixing, the investigated heights show less veer than onshore. Wind shear at the offshore location is lower compared to the onshore location due to lower surface roughness.

Figure 3 shows the annual horizontal wind speed probability distributions at each height level for both locations. These distributions give insight into the wind statistics at specific heights, but not into the statistics of the wind profile shapes. The nonlinear color gradient allows for the representation of the entire relative probability range. Onshore (a) $U$ are relatively low and have a fairly narrow deviation below 300 m, due to dominant surface effects. Above this height the distribution broadens, but a high probability of low wind speeds remains for the entire height range. The distributions show bimodal characteristics caused by different atmospheric stratification. Low wind speeds are commonly associated with unstable and high wind speeds with neutral or stable atmospheric conditions.

Such multimodal distributions at higher altitudes are better described by the sum of two or more probability distributions, as standard Weibull or Rayleigh distributions cannot capture this phenomenon (Sommerfeld et al., 2019a). Offshore $U$ (b) display

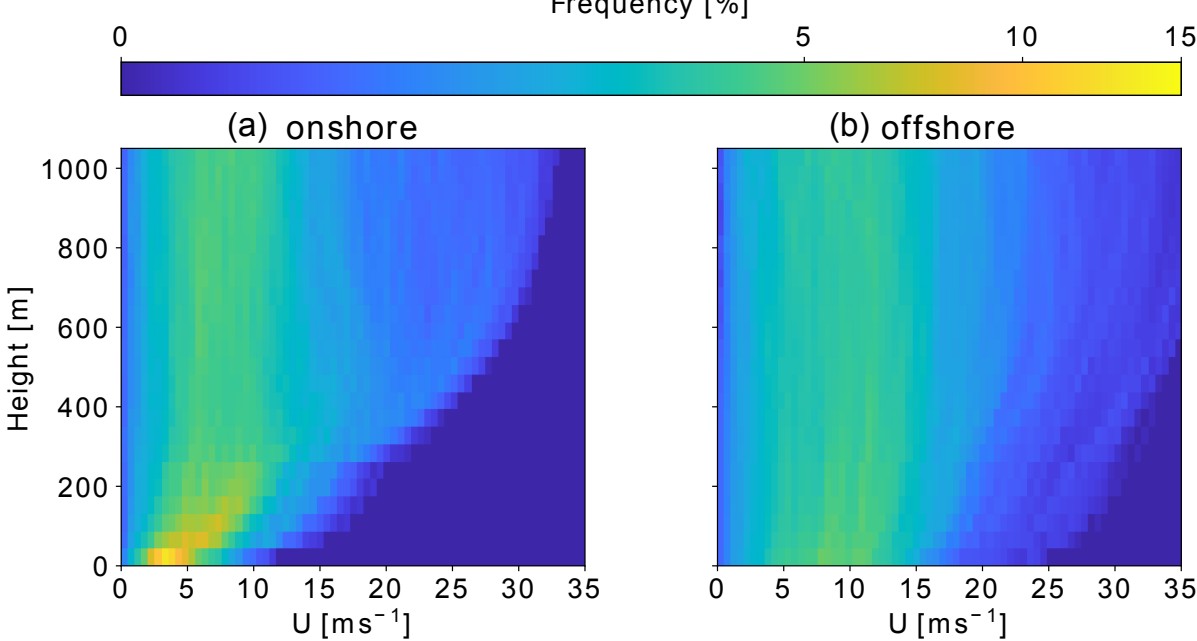

**Figure 3.** Comparison of WRF-simulated annual wind speed $U$ probability distributions at each height level between onshore (a) and offshore (b) up to 1000 m. The nonlinear color scheme represents the high probability of low altitude, particularly onshore, while still differentiating the lower, wide spread frequencies at higher altitudes.

a wider distribution at all heights as they are less affected by surface effects. Similarly to onshore, the offshore frequency distribution also shows a high probability of lower $U$ (between 5-10 $\mathrm{ms}^{-1}$) at all heights. Higher $U$ at lower altitudes benefit conventional WT and weakens the argument for offshore AWES, as one of their benefits would be to harness energy from the stronger winds at higher altitudes. However, other reasons for placing AWES offshore are the safety and land use regulations and the potential cost benefits of a smaller support structure (offshorewind.biz, 2018; Lunney et al., 2017; Ellis and Ferraro,
210    2016).

    The Obukhov length $\mathcal{L}$ (Obukhov, 1971; Sempreviva and Gryning, 1996)

$$\mathcal{L} = \frac{-u_*^3 \theta_{\mathrm{v}}}{kg} \left( \frac{1}{Q_{\mathrm{S}}} + \frac{0.61}{Q_{\mathrm{L}}\theta} \right), \tag{1}$$

commonly characterizes the near-surface atmospheric stability, which highly affects the shape of the wind speed profile $U$, which is the magnitude of the wind velocity profile $\mathbf{U}$. We extent the concept to mid-altitudes between 100 and 1000 m.
The Obukhov length is a function of $u_*$ is the simulated friction velocity, $\theta_{\mathrm{v}}$ the virtual potential temperature, $\theta$ the potential temperature, $Q_{\mathrm{S}}$ the kinematic virtual sensible surface heat flux, $Q_{\mathrm{L}}$ the kinematic virtual latent heat flux, $k$ the von Kármán constant and $g$ the gravitational acceleration. Various stability classifications based on the Obukhov length have been defined for different wind energy sites. Table 2 summarizes the Obukhov length bin widths used by Floors et al. (2011) and the frequency of occurrence of each stability class onshore and offshore, consistent with Sommerfeld et al. (2019b).

Neutral stratification occurs approximately 20% of the year at both locations. The lower heat capacity of the land surface leads to a faster heat transfer and a quicker surface cool-off which favors the development of stable stratification ($\approx$17% onshore vs $\approx$6% offshore). The offshore location has a higher probability of unstable conditions, which is likely caused by a warmer ocean surface compared to the air above (Archer et al., 2016).

**Table 2.** Stability classes based on Obukhov length $\mathcal{L}$ (bins from Floors et al. (2011)) and associated annual probability at Pritzwalk (onshore) and FINO3 (offshore), based on WRF simulations.

| Stability class | $\mathcal{L}$ [m] | onshore | offshore |
|---|---|---|---|
| Unstable (U) | $-200 \leq \mathcal{L} \leq -100$ | 7.27% | 13.66% |
| Nearly unstable (NU) | $-500 \leq \mathcal{L} \leq -200$ | 7.09% | 16.34% |
| Neutral (N) | $|\mathcal{L}| \geq 500$ | 20.71% | 22.82% |
| Nearly stable (NS) | $200 \leq \mathcal{L} \leq 500$ | 12.56% | 5.15% |
| Stable (S) | $50 \leq \mathcal{L} \leq 200$ | 17.24% | 6.20% |
| Very stable (VS) | $10 \leq \mathcal{L} \leq 50$ | 10.04% | 2.96% |
| Other | $-100 \leq \mathcal{L} \leq 10$ | 25.09% | 32.87% |

Both unstable and stable conditions can lead to non-logarithmic and non-monotonic $U$ profiles. Unstable conditions are
often accompanied by almost uniform $U$ profiles due to increased mixing, whereas low-level jets (LLJs) can develop during the nocturnal stable onshore boundary layer (Banta, 2008). Both locations have a high chance of unassigned conditions (labeled as "Other") which are mostly associated with low wind speeds.

## 2.3 Clustering of wind conditions

An accepted method to describe the near-surface atmosphere is atmospheric stability, commonly quantified by the Obukhov
length (Obukhov, 1971; Sempreviva and Gryning, 1996), which exclusively uses surface data (Section 2.2 and Equation (1)). Previous studies (Sommerfeld et al., 2019a, b) showed that Obukhov-length-classified wind speed profiles $U$ diverge with height, especially during neutral and stable conditions, which indicates vertically heterogeneous atmospheric stability and suggests that surface-based stability categorization is insufficient for higher altitudes. Unlike classifying wind regimes by atmospheric stability, which requires additional temperature and heat flux data, clustering only uses wind data at multiple
heights to group profiles by similarity. This results in more cohesive profile groups (Schelbergen et al., 2020). Therefore, clustering can also be applied to wind-only measurements such as LiDAR.

The k-means clustering algorithm (Pedregosa et al., 2011) used in this study is chosen for its ease of use and scalability, due to the high dimensionality of the data set. Many other algorithms produce similar results, but a comparison between clustering algorithms is beyond the scope of this research. The wind velocity profiles $\mathbf{U}'$ (Figure 4, black) are rotated such that the main
wind component $\overline{U}(z_{\mathrm{ref}}) = \overline{U}(100\,\mathrm{m} \leq z \leq 400\,\mathrm{m})$ points in the positive $x$ direction $\mathbf{U}$ (blue), in order to remove directional dependencies. The velocity components at each height level are decomposed into $u$ in the main wind direction and into $v$

perpendicular to it (red). The wind speed profile $U$ is not shown. This results in more homogeneous clusters, and simplifies the comparison of wind data and `awebox` results. It is analogous to assuming omnidirectional operation.

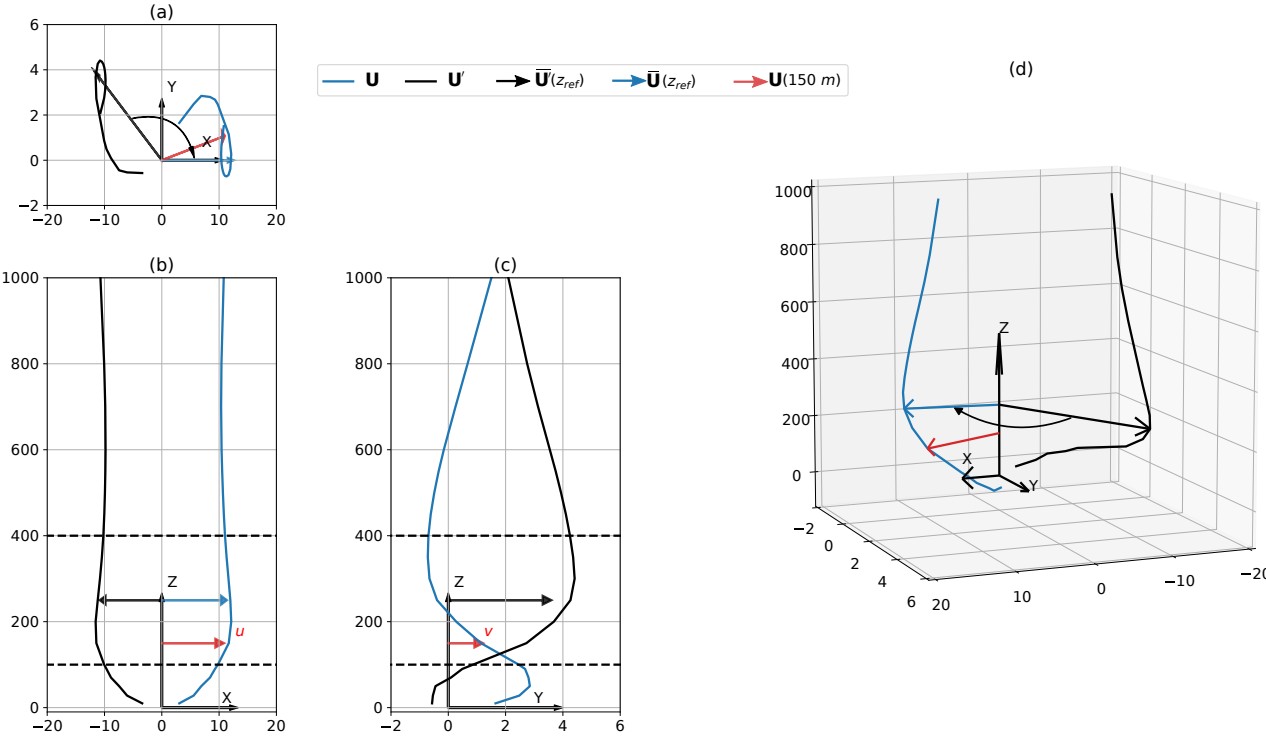

**Figure 4.** Representative wind velocity profile $\mathbf{U}'$ in its original direction and rotated so that the primary wind direction of the resulting profile $\mathbf{U}$ points in the x direction. Panel (a) shows the top view, panel (b) shows the front view, panel (c) shows the side view and panel (d) shows the isometric view. The primary wind direction is defined by the average wind vectors $\overline{\mathbf{U}'}(\mathbf{z}_{\mathrm{ref}})$ (black) and $\overline{\mathbf{U}}(z_{\mathrm{ref}})$ (blue) between 100 and 400 m (dashed lines). As an example, the wind vector at $z = 150$ m (red) decomposed into $u$ and $v$.

The wind velocity data up to 1000 m comprises of data points at 30 height levels and in 2 directions. The clustering algorithm
assigns each data point to one of the k clusters represented by their respective cluster mean, also called the centroid. These
centroids are chosen such that they minimize the sum of the Euclidean distances to every data point within each cluster. This
cost function is also referred to as inertia or within-cluster sum-of-squares. As such, the centroids are usually not actual data
points, but rather the clusters' average, and will at best coincide with a data point by chance. The cluster labels are the result of
random initialization and has no mathematical meaning. We therefore sort and label the clusters by average $\overline{U}(z_{\mathrm{ref}})$ between
100 m - 400 m for the following analyses in Subsection 2.4. The variable k refers to the fixed and predefined number of
clusters. The choice of k significantly affects the accuracy of the wind resource description, as well as the computational cost.
The choice of k is informed by the elbow method, named after the characteristic line chart that resembles an arm, and the
silhouette score. The "elbow" (the point of inflection of the curve) is a good indication that the underlying model fits well

for the corresponding number of clusters. k can be chosen at a point where the inertia reduction becomes marginally small or
decreases linearly (Pedregosa et al., 2011).

Absolute values of inertia (Figure 5 (a)) are not a normalized metric and scale with the size of the considered data set. On
the other hand, the silhouette coefficients (Figure 5 (b, d)) are normalized between -1 (worst) and 1 (best). They indicate the
membership of a data point to its cluster in comparison to other clusters, i.e. the proximity of each data point in one cluster
to data points in neighboring clusters (Pedregosa et al., 2011). A negative value suggests that a data point could be assigned
to the wrong cluster. The silhouette score, depicted by a dashed red vertical line, is the average of all silhouette coefficients
for a fixed number of clusters k. For visualization purposes, k=10 clusters have been chosen. Each cluster is sorted and color
coded according to average $\overline{U}(z_{\mathrm{ref}})$ between 100 m and 400 m and color, same as Figure 6. Performing these silhouette score
analyses for multiple k results in the trend shown in Figure 5 (c). A k value of 20 seems to be a good choice for the available
data sets. This is because the decrease in inertia for a higher number of clusters is only moderate, suggesting that the additional
computational cost may not be worthwhile. Similarly, the silhouette score remains almost constant for higher numbers of
clusters. Therefore, k = 20 has been chosen for the analyses in Section 6.

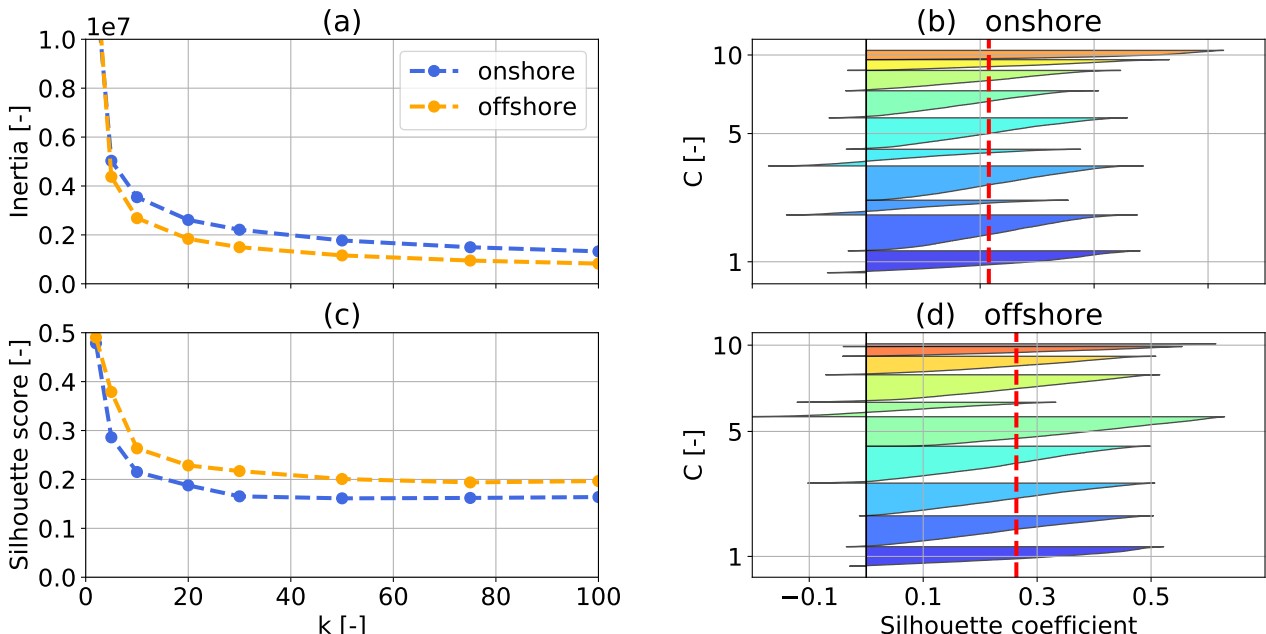

**Figure 5.** k-means clustering inertia over of number of clusters k (a) for one year of onshore (blue) and offshore (orange) wind velocity
profiles **U** up to 1000 m. The onshore (b) and offshore (d) silhouette coefficients express the distance to neighbouring clusters and are color
coded according to average $\overline{U}(z_{\mathrm{ref}})$ between 100 and 400 m, same as in Figures 6 9, 10 and 11. The red dashed line represents the silhouette
score, which is the average silhouette coefficient. Silhouette score (c) over number number of cluster k for both locations. The number of
clusters k = 10 has been chosen for presentation purposes only. The analyses in the results section use k = 20 clusters.

## 2.4 Analyses of clustered profiles

For visualization purposes, the following subsections describe the wind conditions at both locations using only k = 10 clusters. The analyses in the results section (Section 6) use k = 20 clusters.

Figures 6 (a) and 6 (b) show the average of the clustered wind speed profiles $U$, also referred to as centroids. Their colors corresponds to the average $\overline{U}(z_{\mathrm{ref}})$ between heights of 100 and 400 m. All WRF-simulated $U$ are depicted in gray. The cluster probabilities (Figures 6 (c, d)) are sorted by average centroid speed within the considered height range, represented by their colors and labels ($C = 1 - 10$).

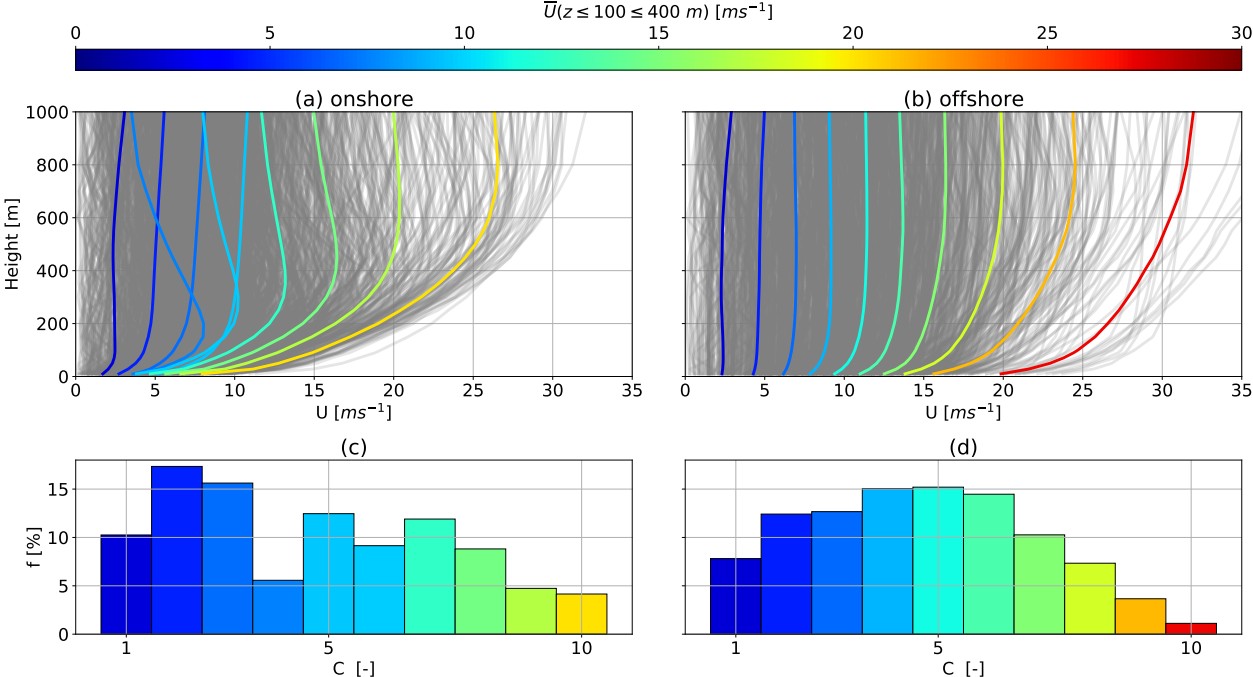

**Figure 6.** Onshore (left) and offshore (right) average annual wind speed profiles (centroids) resulting from k-means clustering for k = 10 (a,b). All comprising WRF-simulated wind speed profiles $U$ are depicted in gray. The centroids are sorted, labeled and color coded in ascending order of average $\overline{U}(z_{\mathrm{ref}})$ between heights of 100 and 400 m. The corresponding cluster frequency of occurrence f for each cluster C is shown in (c) and (d) below.

     As expected, offshore $U$ (Figure 6 (b)) at low-altitudes are higher and wind shear is lower than onshore (Figure 6 (a)).
In general, offshore centroids are more monotonic, as they do not exhibit a distinct $U$ peak (i.e. LLJs), and achieve higher maximum wind speeds than onshore. The $U$ profiles within each cluster cover a relatively small range, suggesting consistent clusters. Figure 7 (onshore) and Figure 8 show the distribution of $U$ within each of the clusters. At both locations, the first two clusters (Figures 7 (a, b) and 8 (a, b)) exhibit very low wind shear with a low and almost constant $U$ above 200 m. These low wind speed clusters amount to approximately 25 % onshore Figure 6 (c) and 20% offshore Figure 6 (d), as can be seen in the

corresponding cluster frequency of occurrence f. A standard logarithmic wind profile does not accurately describe such almost constant profiles which could lead to an overestimation of $U$ at higher altitudes. AWESs must either be capable of functioning at low $U$ or be able to safely land and take off autonomously. Onshore clusters 4 and 5 (Figures 7 (d, e) and 8 (d, e)) seem to mostly consist of non-monotonic profiles as these centroids show a distinct LLJ nose at about 200 m and 300 m. The offshore centroids of clusters 7 and 8 (Figures 7 (g, h) and 8 (g, h)) also show a slight wind shear inversion at higher altitudes.

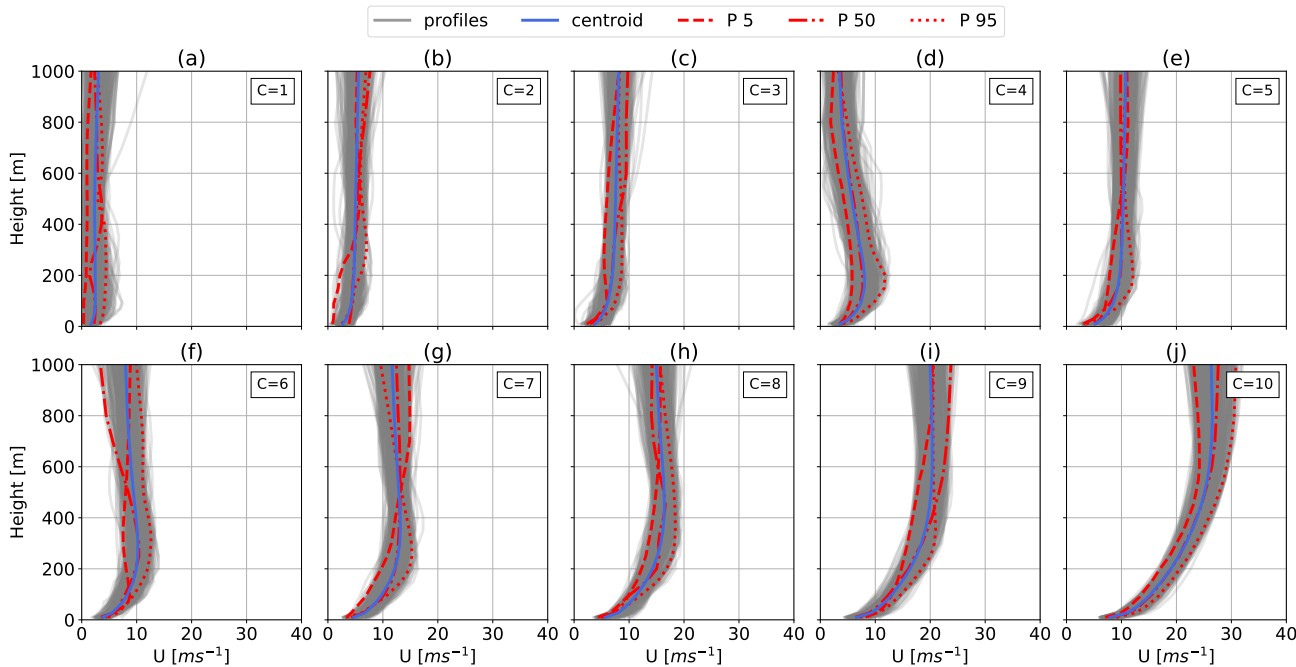

**Figure 7.** Vertical onshore $U$ categorized using the *k*-means clustering algorithm. The analyses in Section 6 employ k = 20 clusters. Here k = 10 is chosen for visualization purposes. The average profile (centroid) is shown in blue and the profiles associated with this cluster are shown in gray. Clusters 1 to 10 (a-j) are sorted and labeled in ascending order of average centroid $\overline{U}(z_{\mathrm{ref}})$ between 100 m and 400 m. The corresponding cluster frequency f for each cluster C is shown in Figure 6. The in the optimization toolbox implemented $U$ are highlighted with a red line.

Clusters C = 1 (a) to C = 10 (j) are sorted by the average centroid (blue line) wind speed between $\overline{U}(z_{\mathrm{ref}} = 100 - 400 \text{ m})$. The red lines indicate the profiles associated with the 5th, 50th and 95th percentile of $\overline{U}(z_{\mathrm{ref}})$ within each cluster. To reduce computational cost, only these profiles are later implemented into the `awebox` optimization toolbox. We selected these profiles because they are less likely to be irregular outliers than the extrema of the cluster, while still representing the variation within their respective cluster. These profiles illustrate the variations within their respective cluster and are not average profiles like the cluster centroids or scaled or semi-empirical approximations such as the logarithmic wind profile. Evidently, the magnitude of the wind speed plays a dominant role in the clustering process. A clearer wind profile shape distinction could be achieved by normalizing the data before clustering it (Molina-García et al., 2019; Schelbergen et al., 2020).

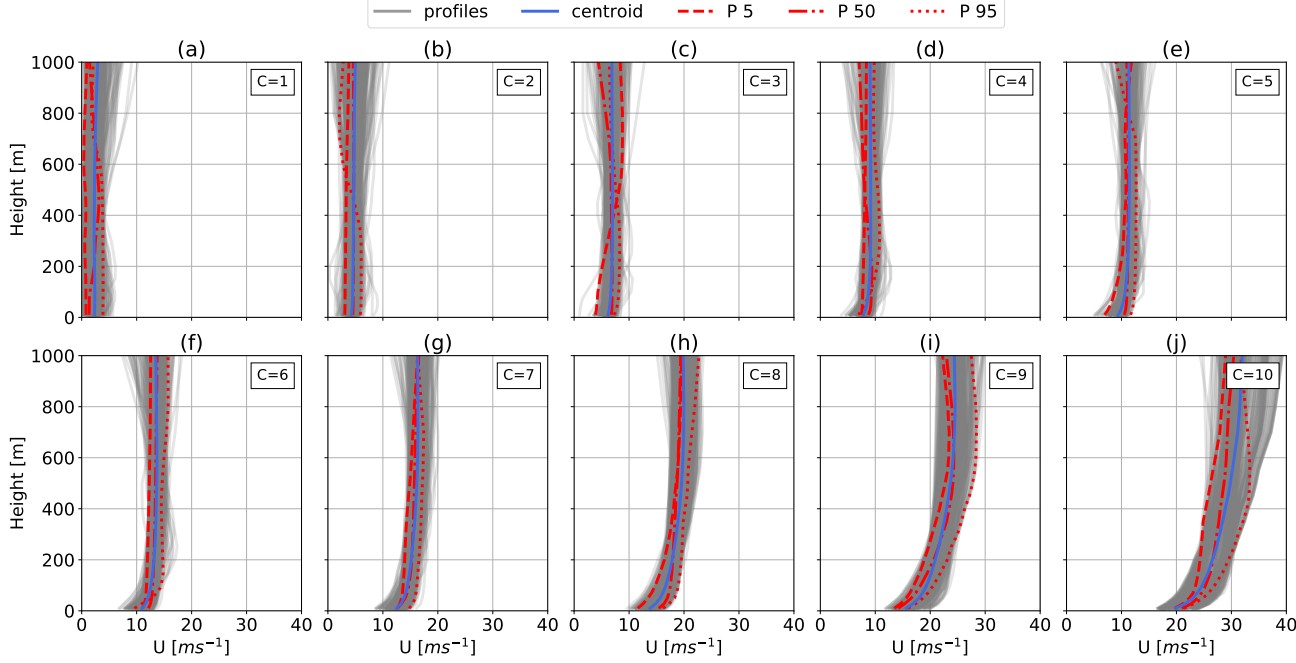

**Figure 8.** Vertical offshore wind speed profiles categorized using the *k*-means clustering algorithm. The analyses in Section 6 employ k = 20 clusters. Here k = 10 is chosen for visualization purposes. The average profile (centroid) is shown in blue and the profiles associated with this cluster are shown in gray. Clusters 1 to 10 (a-j) are sorted and labeled in ascending order of average centroid $\overline{\overline{U}}(z_{\mathrm{ref}})$ between 100 m and 400 m. The corresponding cluster frequency f for each cluster C is shown in Figure 6. The in the optimization toolbox implemented $U$ are highlighted with a red line.

## 2.5 Analysis of clustered statistics

This subsection examines the relationship between the clusters and monthly, diurnal, and atmospheric stability. These analyses reveal patterns that give insight into the wind regime and the resulting changes in AWES performance. Subsequent sections examine wind data from k = 20 clusters, while here only k = 10 clusters are chosen for presentation purposes. Clusters are sorted in ascending order of average centroid wind speed $\overline{U}(z_{\mathrm{ref}} = 100 - 400 \text{ m})$ and color coded as in Figure 5 and Figure 6.

Both locations exhibit a clear annual pattern as shown in Figure 9. High wind speeds are more common during winter while low wind speeds are more prevalent in summer. This is likely due to the seasonal difference in surface heating and the resulting differences in atmospheric mixing. The two onshore and offshore clusters associated with the highest wind speed are almost exclusively present during November to February.

Offshore data indicate minimal diurnal variation as shown in Figure 10, with only a slight increase in the frequency of lower wind speed clusters during the day.Onshore clusters, on the other hand, are more dependent on the diurnal cycle with a

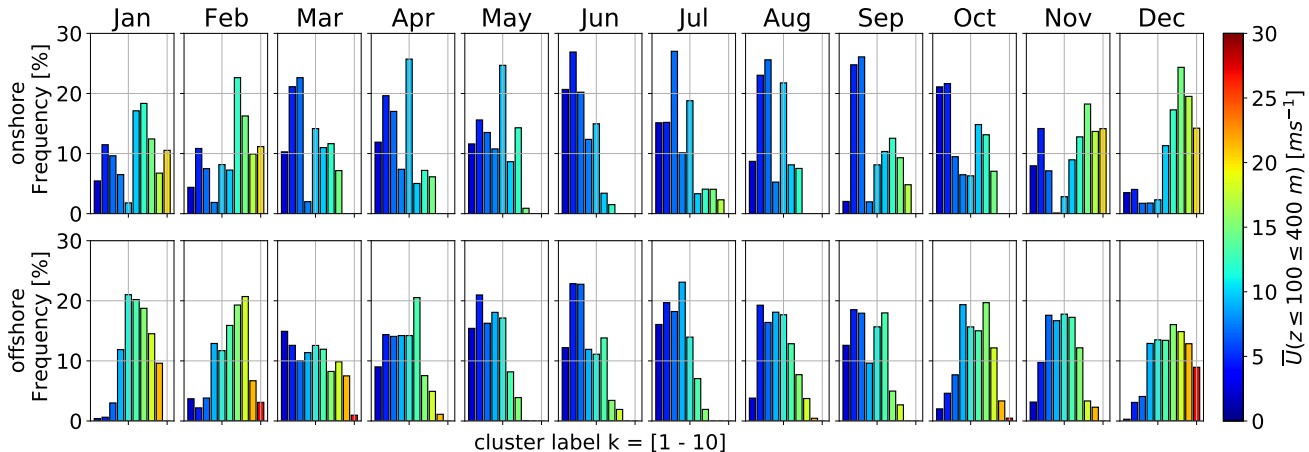

**Figure 9.** Monthly frequency of k-means clustered onshore (top) and (offshore) wind velocity profiles **U** for a representative k = 10. All clusters are sorted and color coded according to their $U(z_{\mathrm{ref}} = 100 - 400\ \mathrm{m})$. The corresponding centroid associated with each cluster can be found in Figure 6.

higher likelihood of low-speed clusters after sunrise. The frequency of onshore cluster 4, which includes a LLJ nose (Figure 305    6), decreases to almost zero during the day and rises at night, supporting the notion that this cluster is linked to nocturnal LLJs.

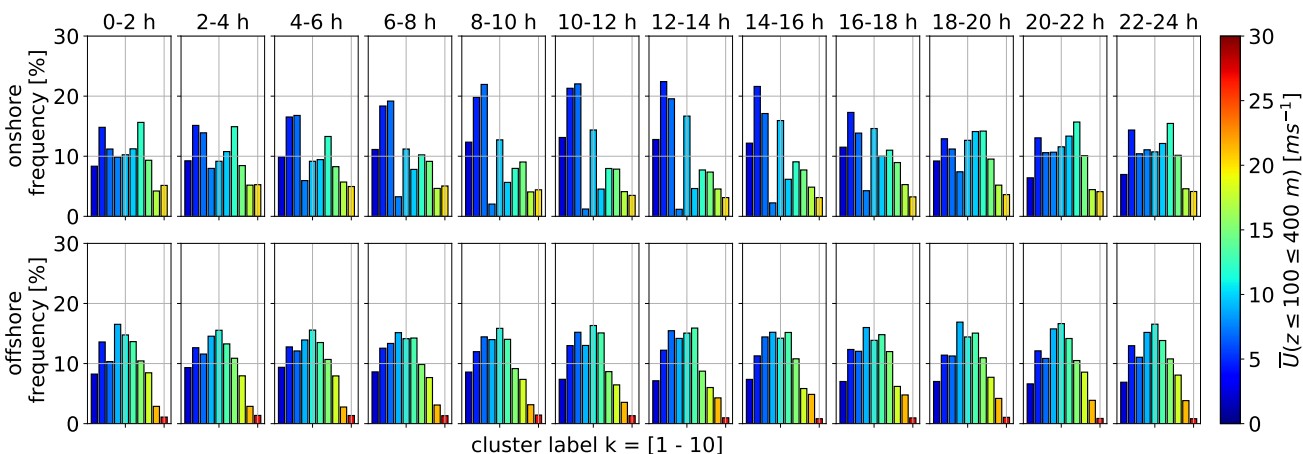

**Figure 10.** Diurnal frequency of k-means clustered onshore (top) and (offshore) wind velocity profiles **U** for a representative k = 10. All clusters are sorted and color coded according to their $U(z_{\mathrm{ref}} = 100 - 400\ \mathrm{m})$. The corresponding centroid associated with each cluster can be found in Figure 6.

Thwe wind velocity clusters (Figure 11) show a correlation with atmospheric stability. Low wind speed clusters make up about 20% to 30% of the annual wind resource. These clusters exhibit Obukhov lengths close to zero (probably caused by very low friction velocity $u_*$) and are classified as "other" according to Floors et al. (2011) (Table 2). Unstable (U) and near unstable (NU) conditions are associated slightly higher wind speeds. The highest wind speeds develop during neutral (N) and near stable (NS) conditions. It should be acknowledged that strong winds driven by large pressure gradients can lead to neutral stratification. LLJ profiles associated with onshore cluster 4 are most likely to develop during stable (S) and very stable (VS) conditions.

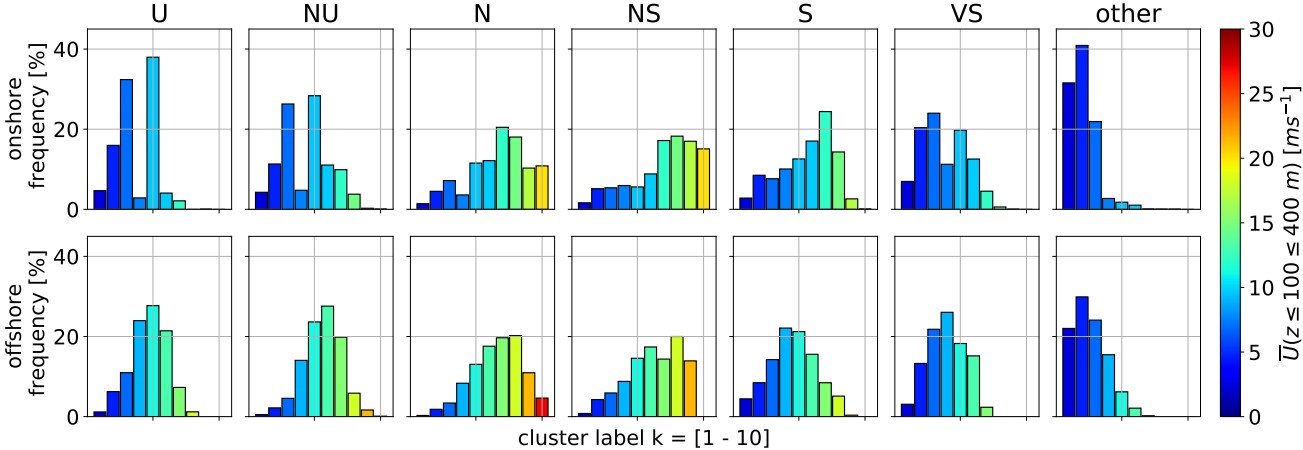

**Figure 11.** Atmospheric stability (U: unstable, NU: nearly unstable, N: neutral, NS: nearly stable, S: stable, VS: very stable) distribution of k-means clustered onshore (top) and (offshore) wind velocity profiles $\mathbf{U}$ for a representative k = 10. The associated stability classes are based on Obukhov length (Table 2). All clusters are sorted and color coded according to their $U(z_{\mathrm{ref}} = 100 - 400\ \mathrm{m})$. The corresponding centroid associated with each cluster can be found in Figure 6.

In summary, k-means clustering can effectively group wind velocity profiles with similar characteristics up to high altitudes. These clusters are correlated with seasonal and diurnal changes as well as atmospheric stability. The magnitude of the wind velocity profiles appears to have a greater impact on the resulting clusters than the shape of the profile. The algorithm is able to identify less common, non-monotonic profile shapes, for example profiles with LLJs, can be identified. Normalizing the profiles before clustering can provide more information about the different shapes of the vertical profiles, but this was not done in this study.

## 3 Dynamic AWES model

This section introduces the dynamic AWES model used in the `awebox` trajectory optimization toolbox (De Schutter et al., 2020). Subsection 3.1 provides a summary of the system configuration. The aerodynamic model is presented in Subsection 3.2, while the aircraft mass model is introduced in Subsection 3.3.

### 3.1 Model configuration

The rigid-body model considers a six degrees of freedom (DOF) fixed-wing aircraft which is connected to the ground via a straight tether. The introduction of the tether reduces the DOF to five (Terink et al., 2011). The model uses precomputed second order polynomials to describe the aerodynamic coefficients (Subsection 3.2) which are controlled via aileron-, elevator- and rudder-deflection (Malz et al., 2019).

The longitudinal dynamics of the tether is controlled via the tether jerk $\dddot{l}$ from which the tether acceleration $\ddot{l}$, reeling speed $\dot{l}$ and length $l$ are determined. The tether is modeled as a single solid rod which neither supports compressive forces nor bending moments (De Schutter et al., 2019). The rod is divided into 10 segments. Tether drag is calculated individually for each segment, using the local apparent wind speed (Bronnenmeyer, 2018). The tether drag of every segment is equally distributed between the two endpoints. This leads to an underestimation of total tether drag at the kite (Leuthold et al., 2018). The ground station dynamics are not modeled explicitly, but are implemented using a set of constraints. These constraints serve as an example of a system rather than representing a fully optimized design. A reel-in speed of $\dot{l}_{\mathrm{in}} = 15 \ \mathrm{ms}^{-1}$ and reel-out speed of $\dot{l}_{\mathrm{out}} = 10 \ \mathrm{ms}^{-1}$ are assumed to be realistic winch motor constraints based on information provided by a ground station manufacturer and literature review. This results in a reel-out to reel-in ratio of $\frac{2}{3}$. A maximum tether acceleration of $\ddot{l} = 20 \ \mathrm{ms}^{-2}$ is imposed to comply with generator torque limits. The tether diameter is selected to be able to withstand three times (safety factor of SF = 3) the maximum tether tension of $F_{\mathrm{tether}}^{\mathrm{max}} = 50$ kN. This results in a rated average cycle power of about $P_{\mathrm{rated}} \approx 260 - 300$ kW, according to `awebox` simulations.

### 3.2 Aerodynamic model

The presented model utilizes the Ampyx Power AP2 aerodynamic coefficients from De Schutter et al. (2020); Malz et al. (2019); Ampyx Power BV (2020). The AP2 reference is scaled from a projected wing surface area of $A_{AP2} = 3 \ \mathrm{m}^2$ to $A = 20 \ \mathrm{m}^2$, to generate results for more realistic and probable devices, while the aspect ratio is kept constant at $AR = 10$. The total combined drag coefficient of the aircraft and tether $c_{\mathrm{D,total}}$:

$$c_{\mathrm{D,total}} = c_{\mathrm{D,kite}} + \frac{1}{4}\frac{ld}{A}c_{\mathrm{D,tether}}. \tag{2}$$

depends on the diameter $d$ and length $l$ of the tether, as well as the projected surface area $A$ and the aerodynamic drag coefficient $c_{\mathrm{D,kite}}$ of the kite. To illustrate the effect of a longer tether, we utilize a simple tether drag estimation for a cylindrical tether with constant diameter and an aerodynamic tether drag coefficient $c_{\mathrm{D,tether}}$ of 1.0. This value would be even higher for braided

tethers. Assuming a constant and uniform wind speed, the line integral along the tether results in a total effective drag coefficient

of

$ld/4A$ accounts for the different reference areas for $c_{D,kite}$ and $c_{D,tether}$. See Houska and Diehl (2007), Argatov and Silvennoinen (2013) and van der Vlugt et al. (2019) for details. Figure 12 visualized the tether drag impact on the aerodynamic coefficients for tether lengths up to $l = 1000$ m.

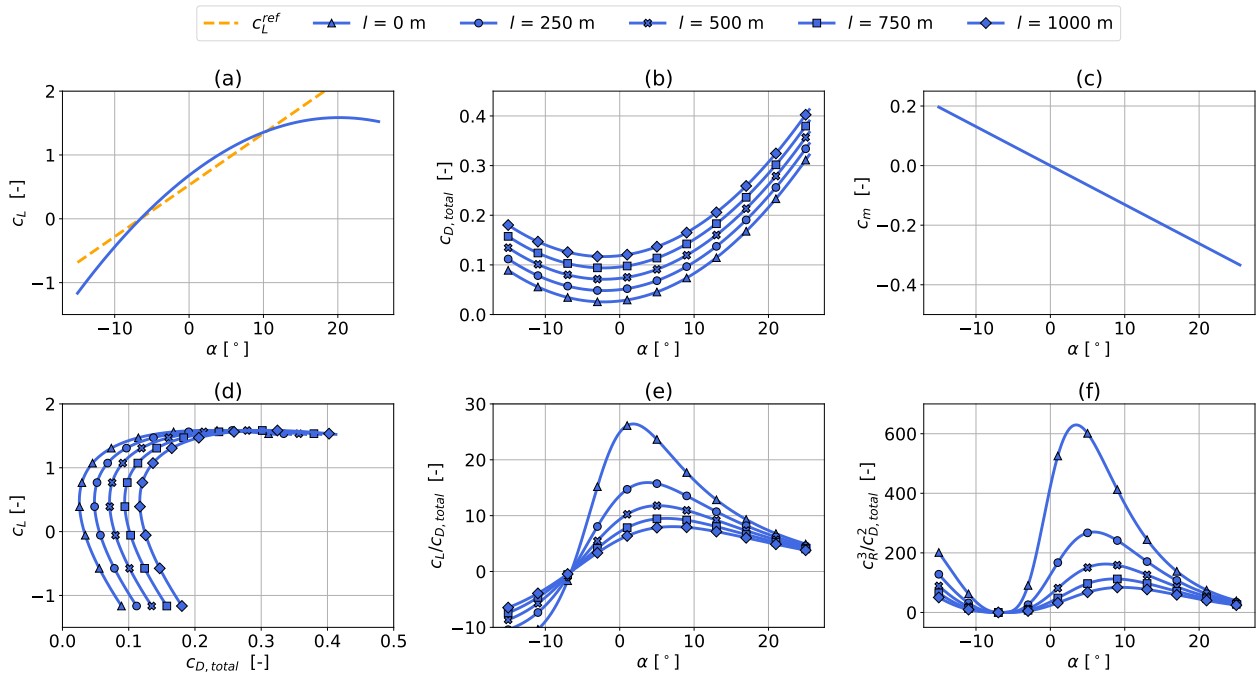

**Figure 12.** Ampyx Power AP2 reference kite aerodynamic lift $c_L$ (a) and drag $c_{D,total}$ coefficients (b) (Malz et al., 2019; Ampyx Power BV, 2020), including tether drag according to Equation (2), for a projected wing surface area $A = 20$ m$^2$ and tether diameter of $d = 7.8$ mm. Tether length varies between 250 m and 1000 m. (c) shows the pitch moment coefficient $c_m$ as a function of angle of attack $\alpha$. The bottom figures display lift over drag (d), glide ratio over angle of attack (e) and $c_R^3/c_{D,total}^2$ over angle of attack.

The lift coefficient $c_L$ (Figure 12 (a)) is approximated as a second-order polynomial function of angle of attack $\alpha$, to simulate
stall effects. A polynomial description is necessary for the entire range of angle of attack, as the optimization algorithm requires a two-times differentiable function. For the sake of simplicity, a piecewise, continuous and differentiable function has not been implemented. As a result, the implemented $c_L$ (blue) slightly exceeds the linear (orange) lift coefficient $c_L^{ref}$ of the AP2 reference (Malz et al., 2019) between $-5 \leq \alpha \leq 10°$. The side slip angle $\beta$ is included in the model, but variation of aerodynamic coefficients due to $\beta$ are neglected. The pitch moment (Figure 12 (c)) is assumed to behave linearly. Changes
in the drag coefficient (Figure 12 (b)) are approximated by a second-order polynomial. Tether drag is independent of $\alpha$ and

therefore added to the zero-lift drag coefficient. The resultant aerodynamic force coefficient $c_R$ is represented as

$$c_R = \sqrt{c_L^2 + c_{D,\text{total}}^2}. \tag{3}$$

The drag polar in Figure 12 (d) depicts the relationship between the kite's lift coefficient $c_L$ and total drag coefficient $c_{D,\text{total}}$ for the tethered aircraft. The maximum values of the glide ratio $c_L/c_{D,\text{total}}$ (Figure 12 (e)) and the ratio $c_R^3/c_{D,\text{total}}^2$ (Figure 12

(f)) which is one of the main determining factors of AWES power (Loyd, 1980; Schmehl et al., 2013), decrease significantly with tether length and shift towards higher angles of attack. The impact of tether drag on the total drag coefficient is less significant for larger kites because its impact decreases with the size of the aircraft.

## 3.3 Aircraft mass model

The aircraft dynamics are described by a single rigid body of mass $m_{\text{kite}}$ and moment of inertia $\mathbf{J}$, subject to aerodynamic forces

and moments. The inertial properties $m_{\text{kite}}$ and $\mathbf{J}$ are determined by upscaling the AP2 reference kite from $A_{\text{AP2}} = 3 \text{ m}^2$ to $A = 20 \text{ m}^2$. The mass $m_{\text{scaled}}$ and moment of inertia $\mathbf{J}_{\text{scaled}}$ of a fixed wing aircraft scale as functions of the wing span $b$ and aspect ratio AR, which is kept constant and its impact on scaling is neglected here, with a mass-scaling exponent $\kappa$ (Noth and Siegwart, 2006)

$$m_{\text{scaled}} = m_{\text{ref}} \left( \frac{b}{b_{\text{ref}}} \right)^\kappa, \tag{4}$$


$$\mathbf{J}_{\text{scaled}} = \mathbf{J}_{\text{ref}} \left( \frac{b}{b_{\text{ref}}} \right)^{\kappa+2}. \tag{5}$$

Pure geometric scaling of solid bodies, in contrast to aircraft structures that use a lightweight structural frame, corresponds to Galileo's square-cube law with $\kappa = 3$. In reality, as has been seen for the development of conventional WTs, design and material improvements occur over time. An appropriate mass-scaling factor was determined based on a review of the available

literature. Figure 13 depicts actual (circle) and anticipated (square) aircraft mass scaling provided by Makani (red color scheme) and Ampyx Power (blue color scheme). The diamond shaped data points (green color scheme) are scaled up versions of Ampyx Power prototypes used in several research papers (Haas et al., 2019; Eijkelhof et al., 2020; van Hagen et al., 2023). The gray area encompasses most of the data points with $\kappa = 2.2 - 2.6$. We chose $\kappa = 2.4$ based on a curve fit of the available published sizing study data. This appears to be an ambitious goal for rigid kites, but attainable for flexible ones. The mass of these hollow

tensile structures filled with air mostly scales the wing surface area, leading to significantly lower mass-scaling exponents and more beneficial mass-scaling. Sommerfeld et al. (2022) examined the impact of various size, mass, and aerodynamic scaling factors on performance.

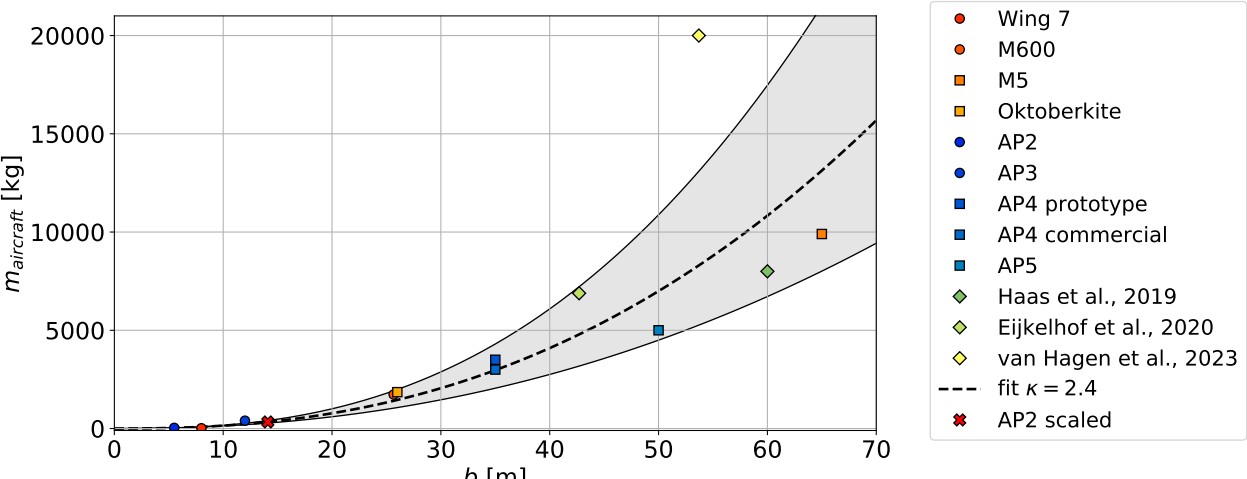

**Figure 13.** Published actual (circle) and anticipated (square) aircraft mass scaling provided by Makani (red color scheme) (Echeverri et al., 2020) and Ampyx Power (blue color scheme) (Ampyx Power BV, 2020; Kruijff and Ruiterkamp, 2018). Diamond shaped data (green color scheme) depict the mass of scaled up versions of Ampyx Power prototypes used in research papers (Haas et al., 2019; Eijkelhof et al., 2020; van Hagen et al., 2023). For most data, mass scales with a scaling exponent between $\kappa = 2.2 - 2.6$ (gray area). The chosen mass-scaling exponent of $\kappa = 2.4$ is represented by a dashed line and the investigated scaled AP2 design is highlighted by a red $X$.

## 4 Optimal control model

AWES need to dynamically adapt to changing wind conditions to maximize power generation and ensure save operation. Subsection 4.1 introduces the dynamic trajectory optimization toolbox `awebox` (De Schutter et al., 2020). We describe the most important boundary conditions in Subsection 4.2 and initial conditions in Subsection 4.3. Subsection 4.4 explains the implementation of the previously described wind profiles (Section 2).

### 4.1 AWES model overview

Maximizing the average cycle power can be formulated as an trajectory optimization problem, which takes into account the interaction between the tether, kite, and ground station. This study analyzes the mechanical power produced by a single aircraft tethered with a straight line throughout one production cycle, including reeling in and out, while disregarding take-off and landing. Power production is intrinsically linked to the aircraft's flight dynamics, as the AWES never reaches a steady state over the course of a power cycle. Generating dynamically feasible and power-optimal flight trajectories is nontrivial, given the nonlinear and unstable system dynamics and the presence of various flight envelope constraints. Optimal control methods are a natural candidate to tackle such problems, given their inherent ability to deal with nonlinear, constrained multiple-input-multiple-output systems (De Schutter et al., 2019; Leuthold et al., 2018). This trajectory optimization is a highly nonlinear and

non-convex problem which can have multiple local optima. The initial and final states of each trajectory must be equal to ensure periodic operation but are freely chosen by the optimizer. In periodic optimal control, an optimization problem is solved by computing periodic system states and control inputs that maximize a performance index (here average power output $\overline{P}$) while satisfying the system's dynamic equations and constraints. We use this approach to generate a variety of realistic trajectories from WRF-simulated wind velocity profiles. Any wind data sets, such as wind atlas data, LiDAR or met mast measurements can be implemented into the optimization model via a two-times differentiable function, depending on the scope and purpose of the investigation.

## 4.2 Constraints

Several important constraints define the operational envelop. The most important constraints such as tether length, tether reeling speed and tether force are summarized in Table 3. The following constraints define a representative and not design-optimized AWES. The power of ground-generation AWES is limited by the tether force, which is defined by the tensile strength $\sigma_{\mathrm{max}}^{\mathrm{tether}}$ and tether diameter $d$, and the tether reeling speed $\dot{l}$. The tether diameter is chosen such that the maximum tether tension is approximately $F_{\mathrm{tether}}^{\mathrm{max}} = 50$ kN with an additional safety factor of SF = 3. This produces a peak power of $P_{\mathrm{peak}} \approx 500$ kW, with a maximum reel-out speed of $\dot{l} = 10$ ms$^{-1}$. This corresponds to a rated average cycle power of approximately $P_{\mathrm{rated}} \approx 260 - 300$ kW. We assume a reel-out to reel-in tether reeling speed ratio of $\frac{2}{3}$ to be within winch design limitations. The tether length constraint is relatively lenient, to allow the optimizer to investigate a wide range of possible operating heights $z_{\mathrm{oper}}$. The flight envelope is constrained by limitations on the aircraft's acceleration, roll and pitch angles (to prevent collision with the tether), as well as the angle of attack $\alpha$ and side slip angle $\beta$. A minimal operating height of $z_{\mathrm{min}} = 50\ m + \frac{b}{2}$ is imposed for safety reasons.

## 4.3 Initialization

The trajectory optimization process is highly nonlinear and non-convex, resulting in multiple local optima. These solutions depend on the chosen initial conditions. Some of the locally optimal solutions may be feasible and within the constraints, but may have undesirable characteristics such as looping maneuvers during reel-in or excessively high operating altitudes. As a result, it is necessary to evaluate the quality of all solutions. To solve the complex optimization problem, initial guesses are generated using a homotopy technique similar to Gros et al. (2013). This technique initially fully relaxes the dynamic constraints using fictitious forces and moments to reduce model nonlinearity and coupling, which improves the convergence of Newton-type optimization techniques. The constraints are then gradually re-introduced until the relaxed problem matches the original problem. The trajectory optimization is initialized with a circular trajectory in downwind direction (positive $x$ direction) with a fixed number of $n_{\mathrm{loop}}$= 5 loop maneuvers at a 30° elevation angle, an initial tether length $l_{\mathrm{init}} = 500$ m and an estimated aircraft speed of $v_{\mathrm{init}} = 10$ ms$^{-1}$ along entire initial trajectory. This initialization is kept constant for all vertical wind velocity profiles. The number of loop maneuvers is not part of the objective function and remains unchanged during all optimization runs. Further investigation is needed to determine the impact of the number of loops. However, previous analyses have shown that the average cycle power estimated by `awebox` is relatively unaffected by the number of loops. It is likely

**Table 3.** Selected AWES design parameters for the original AP2 reference system (Malz et al., 2019) and the scaled up $A = 20$ m$^2$ design, analyzed in this study. Values in square brackets represent the upper and lower bounds, which are implemented as inequality constraints.

| | Parameter | AP2 | design 1 |
|---|---|---|---|
| | $A$ [m$^2$] | 3 | 20 |
| | $c_{\text{kite}}$ [m] | 0.55 | 1.42 |
| | $b_{\text{kite}}$ [m] | 5.5 | 14.1 |
| Aircraft | AR [-] | 10 | 10 |
| | $m_{\text{kite}}$ [kg] | 36.8 | 355 |
| | $\alpha$ [°] | | [-10 : 30] |
| | $\beta$ [°] | | [-15 : 15] |
| | $l$ [m] | | [1 : 2000] |
| | $\dot{l}$ [ms$^{-1}$] | | [-15 : 10] |
| | $\ddot{l}^{\text{max}}$ [ms$^{-2}$] | | [-10 : 10] |
| Tether | $d$ [mm] | | 7.3 |
| | $\sigma_{\text{max}}^{\text{tether}}$ [Pa] | | $3.6 \times 10^9$ |
| | SF [-] | | 3 |
| | $z_{\text{min}}$ [m] | | 60 |
| Operational | $\alpha$ [°] | | [-10 : 20] |
| | $\beta$ [°] | | [-5 : 5] |

beneficial to reduce the number of loops with wind speeds because the system can reel out faster at higher wind speeds and reach maximum tether length faster.

### 4.4 Wind profile implementation

To reduce the computational cost while maintaining an adequate representation, only implement three wind velocity profiles from each cluster into the trajectory optimization toolbox. More profiles could be chosen for an in-depth analysis. The power for

a total number of 60 wind profiles, three profiles for each of the k = 20 clusters (Section 2.3), for each location are maximized. The three selected profiles correspond to the 5th, 50th and 95th percentiles of average wind speed $\overline{U}(z_{\text{ref}} = 100 - 400 \text{ m})$ within each cluster. We assume that these profiles represent the cluster's spectrum of wind conditions at operating height $z_{\text{oper}}$. The awebox includes a simplified atmospheric model based on international standard atmosphere to account for air density variation.

The vertical wind velocity profiles $\mathbf{U}'$ are rotated such that the main wind direction, which is defined as the average direction between 100 and 400 m , points in positive $x$ direction (Figure 4). As a result the wind velocity components at every height consist of a main component $u$ in $x$ direction and transverse component $v$ in $y$ direction. The results can be seen in hodographs of Figure 14 (c) and Figure 15 (c).

The `awebox` toolbox uses the gradient-based MA57 solver (HSL, 2020) in IPOPT (Waechter and Laird, 2016) to solve the nonlinear control problem. Therefore, it is necessary to interpolate the vertical wind velocity profiles with a twice continuously differentiable functions. We chose to use Lagrangian polynomials (Abramowitz and Stegun, 1965) because the resulting polynomials pass through the input data points. To avoid over fitting a limited number of data points are implemented. These data points are chosen based on the anticipated $z_{oper}$, to best represent the wind conditions at relevant heights.

For comparison, logarithmic wind speed profiles,

$$U_{\log} = U_{10} \left( \frac{\log_{10}(z/z_0)}{\log_{10}(z_{10}/z_0)} \right) \tag{6}$$

with a roughness length of $z_0^{\mathrm{onshore}} = 0.1$ and $z_0^{\mathrm{offshore}} = 0.001$, are implemented into the trajectory optimization toolbox.

The reference wind speed $U_{10}$ at reference height $z_{10} = 10$ m varies from 3 to $20\,\mathrm{ms}^{-1}$ with a step size of $\Delta U_{10} = 1\,\mathrm{ms}^{-1}$.

## 5 Reference models

This section establishes a simplified quasi-steady-state AWES reference model (QSM) (Subsection 5.1) and a steady-state WT model (Subsection 5.2) to compare the results of the optimization with analytical solutions.

### 5.1 AWES reference model

The QSM estimates the mechanical power of ground-generation AWES based on the assumption that the trajectory of the tethered aircraft can be approximated by a progression through steady equilibrium states where tether tension and total aerodynamic force are aligned. We simplify the QSM by approximating the reel-out and reel-in trajectory with a single state and neglecting the effects of gravity. The QSM, based on Argatov et al. (2009) and generalized by Schmehl et al. (2013) and van der Vlugt et al. (2019), approximates the aircraft as a point mass. Its position is described in terms of spherical coordinates, i.e. the radial distance from the ground station, the elevation angle $\varepsilon$ and azimuth angle $\phi$ relative to the direction of the mean wind velocity vector $\mathbf{U}$. For lightweight soft-wing kites, this is a reasonably good approximation because the low mass of the kite leads to very short acceleration times. The model includes losses caused by the misalignment of the tether and wind velocity vector. The same design parameters, system constraints and wind conditions (Section 2( apply to the optimization model (Subsection 4.2) as well as the QSM reference model. presented in We maximize the cycle average power $P_{\mathrm{QSM}}$ by varying $l$, $\dot{l}$ and $z$ and assuming an optimal ratio $c_{\mathrm{R}}^3/c_{\mathrm{D,total}}^2$.

The average cycle power $P_{\mathrm{QSM}}$

$$P_{\mathrm{QSM}} = \frac{P_{\mathrm{out}}\, t_{\mathrm{out}} - P_{\mathrm{in}}\, t_{\mathrm{in}}}{t_{\mathrm{total}}} = P_{\mathrm{out}}\, \frac{\dot{l}_{in}}{\dot{l}_{out} + \dot{l}_{in}} - P_{\mathrm{in}}\, \frac{\dot{l}_{out}}{\dot{l}_{out} + \dot{l}_{in}}. \tag{7}$$

can be estimated from the reel-out power $P_{\mathrm{out}}$, the power losses during reel-in $P_{\mathrm{in}}$, the reel-in time $t_{\mathrm{in}}$ and reel-out time $t_{\mathrm{out}}$. We assume reel-in power losses $P_{\mathrm{in}}$ to be zero because optimal reel-in tether tension is negligible. This reduces the average cycle power by up to 30%, depending on wind speed. Due to the cyclic nature of the trajectory, we can determine the ratio of

the reel-in time $t_{\text{in}}$ and reel-out time $t_{\text{out}}$ to the total cycle time $t_{\text{total}}$ from the reel-in speed $\dot{l}_{in}$ and reel-out speed $\dot{l}_{out}$. $\dot{l}_{out}$ depends on the wind speed, while the $\dot{l}_{in} = -15\text{ms}^{-1}$ is assumed to be the maximum reel-in speed.

During the reel-in and reel-out phases, we assume that the tether force $F_{\text{tether}}$ and reeling speed remain constant. The time it takes to transition between these two phases is not taken into account. $P_{\text{out}}$ is calculated from the product of tether reeling speed $\dot{l}$ and tether tension $F_{\text{tether}}$:

$$P_{\text{out}} = F_{\text{tether}} \dot{l}_{\text{out}} = \frac{\rho}{2} A U_{\text{app}}^2 c_{\text{R}} \left( \frac{c_{\text{R}}}{c_{\text{D,total}}} \right)^2 \dot{l}_{\text{out}}. \tag{8}$$

Tether tension is a function of the apparent wind speed $U_{\text{app}}$, air density $\rho$ and the resultant aerodynamic force coefficient

$c_{\text{R}}$ Equation (3). The apparent wind speed can be nondimensionalized by

$$\frac{U_{\text{app}}}{U(z)} = (\cos \varepsilon \cos \phi - f) \sqrt{1 + \left( \frac{L}{D} \right)^2}. \tag{9}$$

The tether reeling speed $\dot{l}$ is nondimensionalized by defining of the reeling factor

$$f = \frac{\dot{l}}{U(z)}. \tag{10}$$

The elevation $\varepsilon$ and azimuth angle $\phi$ constrain $f \leq \cos \varepsilon \cos \phi$ because the magnitude of the apparent wind speed cannot be

negative. Combining Equations (8) and (10) results in:

$$P_{\text{out}} = \frac{\rho}{2} A U(z)^3 c_{\text{R}} \left( \frac{c_{\text{R}}}{c_{\text{D,total}}} \right)^2 f \left( \cos \varepsilon \cos \phi - f \right)^2. \tag{11}$$

The optimal reeling factor $f_{\text{opt}} = \frac{1}{3} \cos \varepsilon \cos \phi$ can be obtained from Equation (11) through an extreme value analysis. We assume an average reel-out trajectory represented by a single crosswind state instead of tracking the actual trajectory. The trajectory center is aligned with the main wind direction ($\phi = 0°$). The elevation angle $\varepsilon$ is determined using the tether length $l$

and operating height $z_{\text{oper}}$. $F_{\text{tether}}$ is constrained by the tether diameter $d$, the tensile strength $\sigma_{\text{max}}^{\text{tether}}$ and the safety factor SF

$$F_{\text{tether}} \leq \frac{d^2}{4} \pi \sigma_{\text{max}}^{\text{tether}}. \tag{12}$$

The power-harvesting factor $\zeta$ (Diehl, 2013) is an AWES performance metric.

$$\zeta = \frac{P}{P_{\text{area}}} = \frac{P}{\frac{1}{2} \rho A \overline{U}_{\text{ref}}^3} \tag{13}$$

The harvested power $P$ is expressed relative to the kinetic wind energy flow rate $P_{\text{area}}$, $P_{\text{area}}$ is a mathematical concept rather

than a physical power flux, through an area equivalent to the wing surface area $A$ to nondimensionalize the power. $\zeta$ can be derived from Equation (8) by setting the elevation angle $\varepsilon$ and the azimuth angle $\phi$ to zero.

## 5.2  WT reference model

This section introduces a simplified steady-state reference WT model model that calculates power as

$$P_{\text{WT}} = \frac{1}{2} \rho c_{\text{p}}^{\text{WT}} A_{\text{WT}} U^3(z_{\text{WT}}) \tag{14}$$

with a hub height of $z_{\mathrm{WT}} = 100$ m for both onshore and offshore conditions. The rotor diameter $D_{\mathrm{WT}} \approx 26.9$ m is sized such that an equivalent rated power of $P_{\mathrm{rated}} = 260$ kW is reached at a rated wind speed of $U_{\mathrm{rated}}(z_{\mathrm{WT}} = 100 \text{ m}) = 12 \text{ ms}^{-1}$, assuming a constant power coefficient of $c_{\mathrm{p}}^{\mathrm{WT}} = 0.45$. The power is kept constant above the rated wind speed. The performance of the WT model, dynamic optimization toolbox, and QSM is estimated using the same sampled WRF-simulated wind conditions (Section 2).

## 6    Results and discussion

This section analyses the optimization results and compares them to the reference models. Subsection 6.1 investigates power-optimal trajectories and the time series of operational parameters. Subsection 6.2 examines operating height statistics, tether length and elevation angle trends. Subsection 6.3 visualizes the impact of wind speeds at different reference heights on the power curve. We compare three different wind speeds: the wind speed at a reference height of 100 m $\overline{U}_{\mathrm{ref}} = U(z_{\mathrm{ref}} = 100 \text{ m})$, the average wind speed at pattern trajectory height $\overline{U}_{\mathrm{ref}} = \overline{U}(z_{\mathrm{ref}} = z_{\mathrm{PTH}})$, and the average wind speed at an a priori guess of pattern trajectory height $\overline{U}_{\mathrm{ref}} = \overline{U}(100 \text{ m} \leq z_{\mathrm{ref}} \leq 400 \text{ m})$. The investigated power curves do not represent design-optimal performance. Subsection 6.4 examines the variation in average cycle power cased by realistic wind profiles simulated by WRF and compares them to reference power estimates based on logarithmic profiles. All results are subject to the same constraints and design parameters introduced in Sections 3 and 4.

### 6.1    Flight trajectory and time series results

Figure 14 compares representative onshore and offshore power-optimal flight trajectories. These results have been chosen to visualize typical performance optimized trajectories for realistic wind conditions determined with the `awebox`. The reference wind speed $\overline{U}_{\mathrm{ref}}$ in the legend is the average wind speed at the a priori guess of the pattern trajectory height $z_{\mathrm{ref}}(100 \text{ m} \leq z_{\mathrm{ref}} \leq 400 \text{ m})$.

Figure 14 (a) shows the magnitude of the vertical wind velocity profile $U$. Figure 14 (c) shows the corresponding top view of the wind velocity profile, rotated such that the main wind component (average wind direction between 100 m and 400 m) $u$ points in positive $x$ direction. The WRF-simulated wind profiles are shown in gray. The highlighted segments depict the Lagrangian polynomial fit (Abramowitz and Stegun, 1965) at operating heights. These polynomials that have been incorporated into the optimization toolbox provide a sufficient fit for the wind data. Figures 14 (b) and (d) show a side ($x - z$ plane) and top view ($x - y$ plane) of the optimized trajectories. The optimization predicts an increase in tether length, operating height and stroke length with wind speed. Figure 15 shows similar results for the offshore location. Figure 16 illustrates the corresponding temporal development of important operational parameters.

The optimizer maximizes tether tension by adjusting the reel-out speed and angle of attack (Figure 16 (a)) during reel-out even for lower wind speed and adjusts the reel-out speed (Figure 16 (c)) to maximize average cycle power. This causes the reeling factor to exceed its optimal value of $f_{\mathrm{opt}} = \frac{1}{3} \cos\varepsilon \cos\phi$ at high wind speeds, resulting in an increase in power (Figure 16 (e)) even when the maximum tether force has been reached. The low wind speed example $\overline{U}_{\mathrm{ref}} = 5.4 \text{ ms}^{-1}$ (blue) seems to

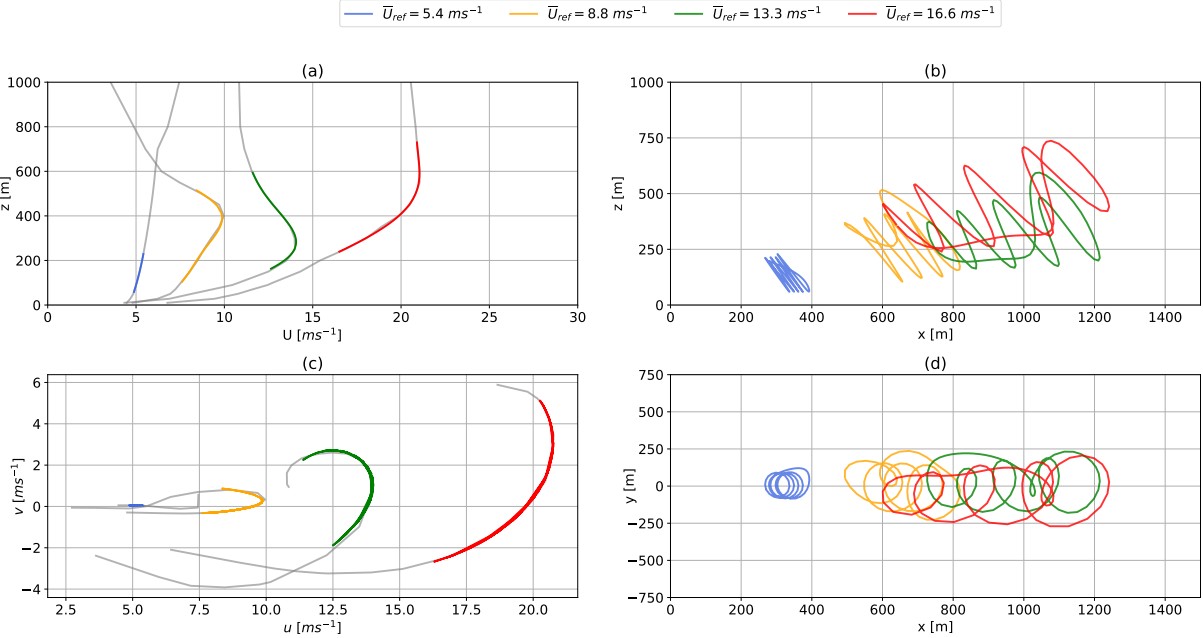

**Figure 14.** Representative WRF-simulated vertical onshore wind speed profiles $U$ (a), and hodograph (top view) up to 1000 m (c). The highlighted sections indicate Lagrangian polynomial fit of the wind velocity at operating height. Panel (b) and panel (d) show the side and top view of the corresponding `awebox`-optimized trajectories. The reference wind speed in the legend is $\overline{U}_{\mathrm{ref}} = \overline{U}(100\,\mathrm{m} \leq z_{\mathrm{ref}} \leq 400\,\mathrm{m})$. The results correspond to the time series shown in Figure 16.

be just above cut-in wind speeds. The tether reeling speed decreases to zero for a prolonged period during the reel-out phase in order to generate enough lift to keep the aircraft aloft. The production period remains almost constant ($t \approx 60$ sec) for the moderate and high wind speed trajectories (orange, green and red), while the reel-in period increases with wind speed, due to the increased reel-out length caused by a higher average reel-out speed. There are significant power losses during the transition between the production and retraction phases when the tether is being reeled in and the tension remains high because the aircraft is unable to depower quickly enough. During the reel-in phase, the tether reeling speed reaches its limit while the tether tension decreases to zero as the aircraft reduces its angle of attack and lift (Figure 16 (d)). At higher wind speeds the optimizer increases the elevation angle and reduces angle of attack to stay within the constraints. This can result in odd or unexpected trajectories, even though these local minima are feasible solutions within the system constraints. Tether length (Figure 16 (f)) generally increases with wind speed as the system reels out faster, increases its elevation angle and operates at higher altitude. Similar results for the offshore location are shown in Figure 17.

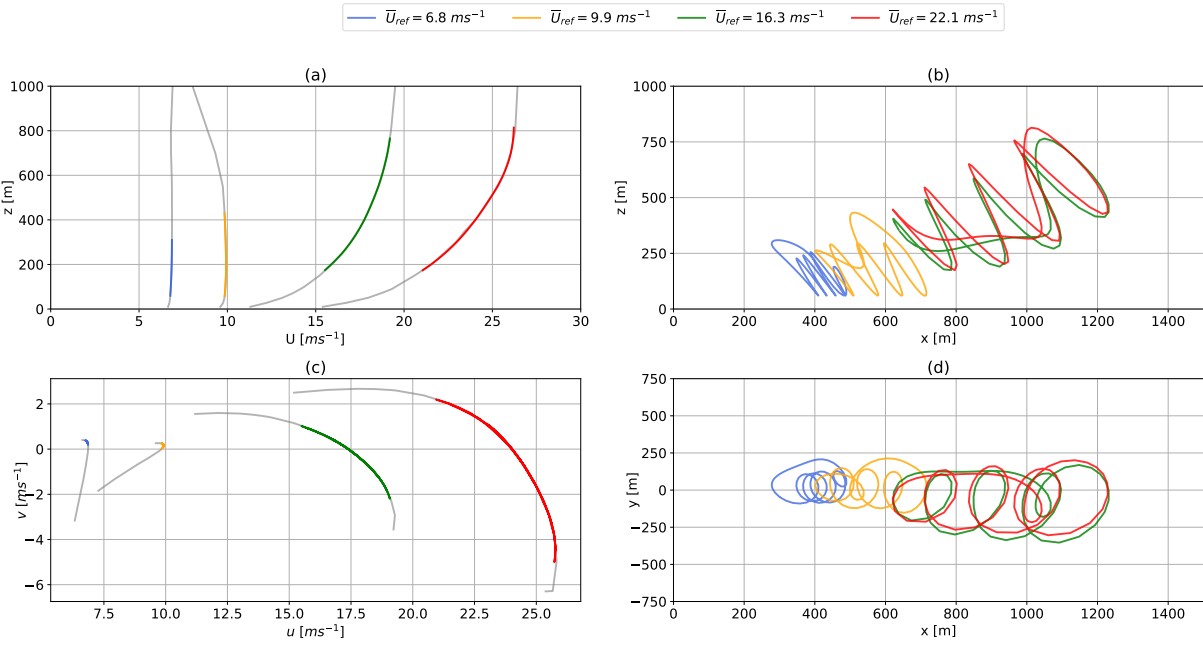

**Figure 15.** Representative WRF-simulated vertical offshore wind speed profiles $U$ (a), and hodograph (top view) up to 1000 m (c). The highlighted sections indicate Lagrangian polynomial fit of the wind velocity at operating height. Panel (b) and panel (d) show the side and top view of the corresponding `awebox`-optimized trajectories. The reference wind speed in the legend is $\overline{U}_{\mathrm{ref}} = \overline{U}(100\,\mathrm{m} \leq z_{\mathrm{ref}} \leq 400\,\mathrm{m})$. The results correspond to the time series shown in Figure 17.

## 6.2 Tether length, elevation angle and operating altitude

Figure 18 (a) illustrates the range of tether lengths $l$ for each of the 60 onshore (blue) and offshore (orange) wind velocity profiles. The maxima and minima are highlighted by circles and plotted over reference wind speed $\overline{U}(z_{\mathrm{ref}} = 100 - 400\,\mathrm{m})$.

None of the optimizations reach the maximum tether length constraint of $l_{\mathrm{max}} = 2000\,\mathrm{m}$ because a longer tether would not be advantageous due to the added drag and weight, which would decrease performance. Both locations show a trend towards longer tethers up to rated wind speed, where the reel-out speed and tension are almost constant and close to their respective constraint (Figure 16). The maximum tether length remains almost constant above rated wind speed while the minimum tether length increases slightly, reducing the total stroke length. The elevation angle (Figure 18 (b)) decreases as the tether length increases. The optimizer tries to keep the elevation angle low in order to reduce misalignment (cosine) losses between the tether and the wind velocity vector. The onshore elevation angle is slightly higher because of the increased wind shear, which makes higher operating altitudes more justifiable. This can also be seen in Figure 18 (c) which shows the frequency distribution of operating altitude. Of the optimal operating heights, 78.6% onshore and 74.7% offshore are below 400 meters, confirming

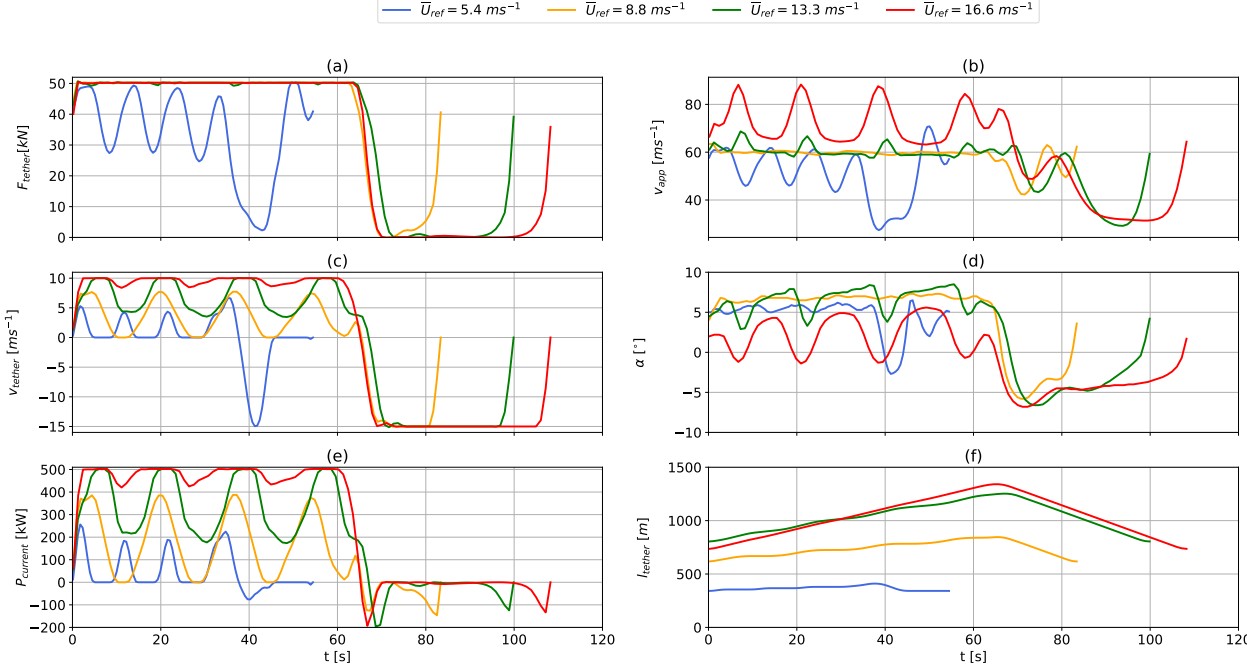

**Figure 16.** Time series of instantaneous tether tension (a), apparent wind speed (b), tether-reeling speed (c), angle of attack (d), power output (e) and tether length (f) over one pumping based on representative onshore WRF-simulated wind data. The results correspond to trajectories shown in Figure 14.

the findings in Sommerfeld et al. (2019a, b). Larger or multikite AWES could benefit from higher operating altitudes due to their higher lift to tether drag ratio and lift to tether weight ratio, but more detailed analyses are required.

### 6.3 Impact of reference height on power curve

The power curve of wind energy converters quantifies the power that can be harvested at a given reference wind speed. For conventional WTs the wind speed at hub-height is commonly used as reference wind speed. Whether this is appropriate for
ever growing towers and longer rotor blades is debatable (Van Sark et al., 2019; Wharton and Lundquist, 2012; Association et al., 2012). Defining a reference wind speed for AWES is not trivial, as the operating height dependents on the shape and magnitude of the vertical wind speed profile. The choice of reference wind speed impacts the power curve representation. The AWE Glossary (Airborne Wind Europe, 2021) recommends to use the wind speed at pattern trajectory height $z_{\mathrm{PTH}}$, which is the expected or logged time-averaged height during the power production phase, as reference wind speed. We estimate
$100\,\mathrm{m} \leq z_{\mathrm{ref}} \leq 400\,\mathrm{m}$ as an a priori guess of the pattern trajectory height. Figure 19 compares onshore (a) and offshore (b) average cycle power over $U(z_{\mathrm{ref}} = 100\,\mathrm{m})$ (blue), $\overline{U}(z_{\mathrm{ref}} = z_{\mathrm{PTH}})$ (green) and an a priori guess of the wind speed at pattern trajectory height $\overline{U}(100\,\mathrm{m} \leq z_{\mathrm{ref}} \leq 400\,\mathrm{m})$ (orange).

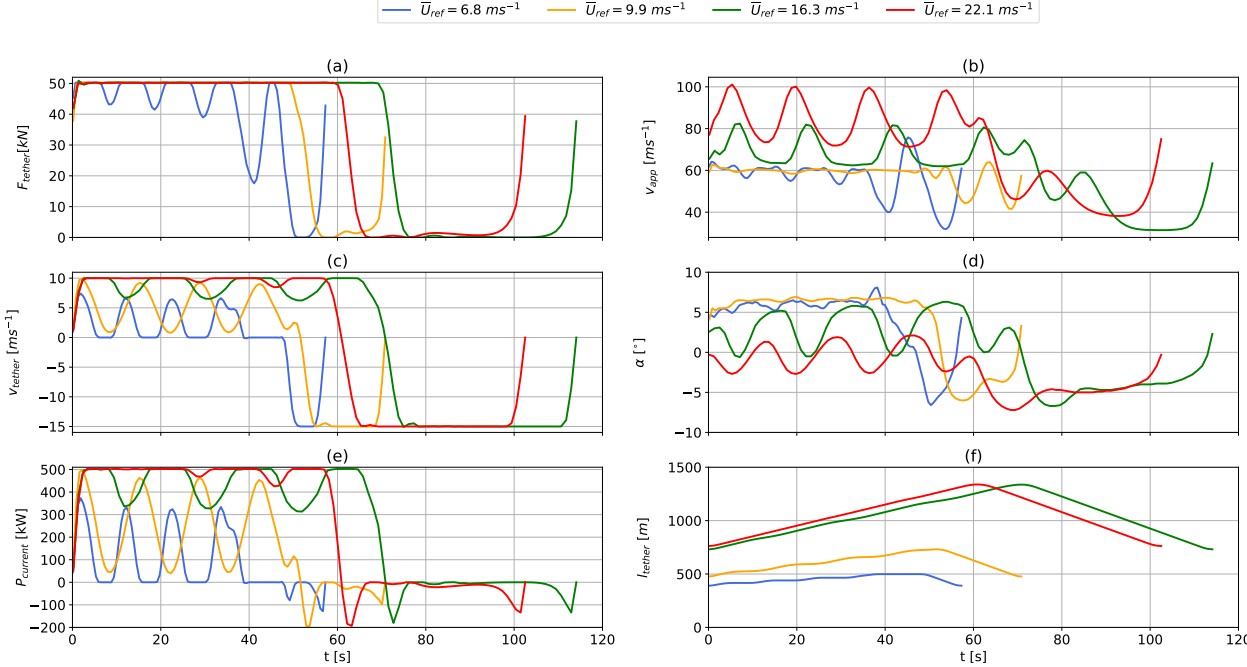

**Figure 17.** Time series of instantaneous tether tension (a), apparent wind speed (b), tether-reeling speed (c), angle of attack (d), power output (e) and tether length (f) over one pumping based on representative offshore offshore WRF-simulated wind data. The results correspond to trajectories shown in Figure 15.

Each data point corresponds to one of the sampled WRF-simulated wind velocity profiles $\mathbf{U}$. The dashed lines, which are only added as visual aid, are a least-square spline interpolation of the approximately 60 data points. Based on these results, we can conclude that the selection of the reference height is more important for onshore conditions. The onshore wind conditions with their higher number of non-monotonic wind profiles and higher wind shear lead to larger deviations from the typical power curve shape described in (Licitra et al., 2019; Airborne Wind Europe, 2021) and others. The higher wind shear onshore leads to a shift towards lower wind speeds for a reference height of $z_{\mathrm{ref}} = 100$ m. The a priori pattern trajectory height guess of $100$ m $\leq z_{\mathrm{ref}} \leq 400$ m is relatively close to the actual $z_{\mathrm{PTH}}$, especially for lower wind speeds. At very high wind speeds above $\overline{U}_{\mathrm{ref}} \geq 20$ ms$^{-1}$ the $z_{\mathrm{PTH}}$ power shifts towards higher wind speeds indicating an increased operating height.

The more homogeneous offshore wind conditions result in less power variation. The three different reference heights have almost no impact on the offshore power curve up to the rated wind speed. Above $\overline{U}_{\mathrm{ref}} \geq 20$ ms$^{-1}$ the power curves diverge and the average cycle power decreases. This seems to be a result of the `awebox` optimization and its initialization with a fixed number of loop maneuvers. As the wind speed and reel-out speed increase, the aircraft cannot complete all the loop maneuvers before reaching the maximum tether length and transitioning into reel-in. Therefore, one of the loop maneuvers is performed

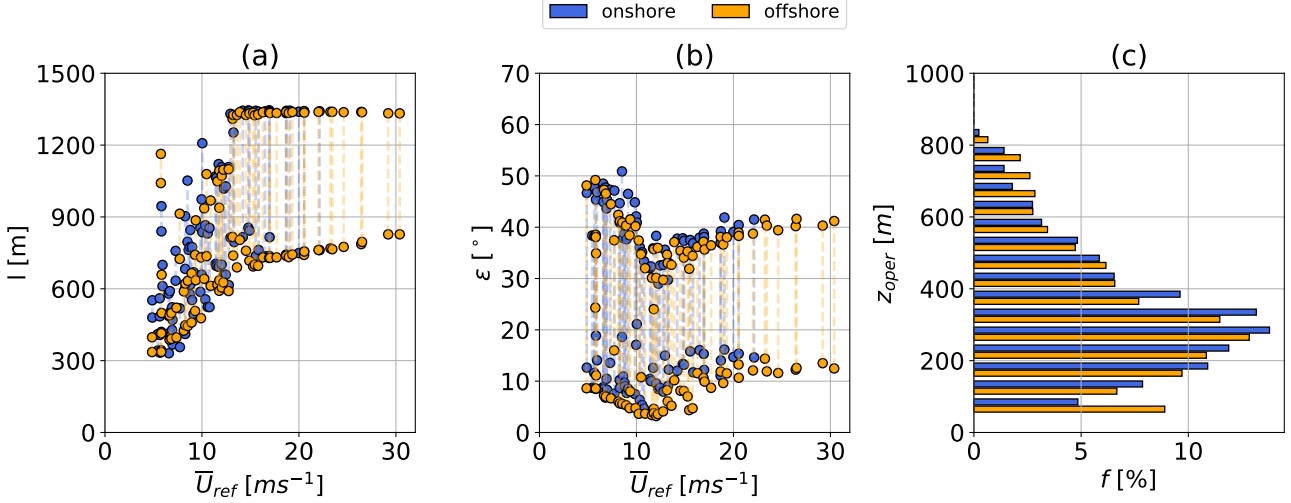

**Figure 18.** Tether length range (a) and frequency distribution of operating height $z_{\mathrm{oper}}$ (b) over reference wind speed $U(z_{\mathrm{ref}} = 100 - 400\,\mathrm{m})$ based on `awebox` trajectory optimizations of k = 20 onshore (blue) and offshore (orange) clusters.

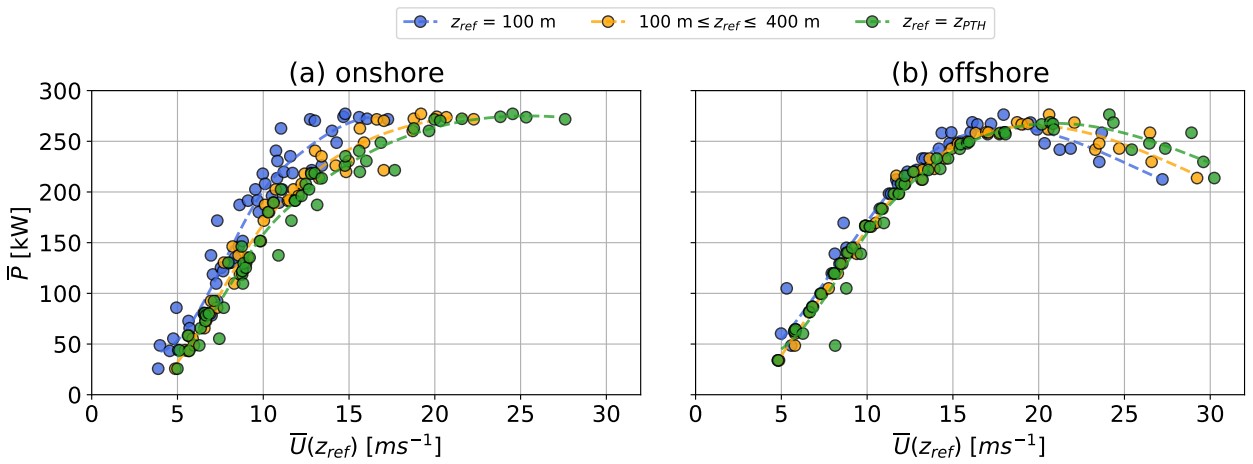

**Figure 19.** Onshore (a) and offshore (b) AWES power curve approximations with wind speeds at $z_{\mathrm{ref}} = 100\,\mathrm{m}$ (blue), $100\,\mathrm{m} \leq z_{\mathrm{ref}} \leq 400\,\mathrm{m}$ (orange) and $z_{\mathrm{ref}} = z_{\mathrm{PTH}}$ (green) reference heights. The dashed lines represent least-square spline interpolations that have been added to aid in visualization.

when already reeling in, leading to an increase in tether tension (Figure A1 (a)) and additional losses during the reel-in period (Figure A1 (e)). The corresponding trajectories are shown in Figure A2 in the appendix.

## 6.4 Reference model power comparison

Figure 20 compares the variation in the power curve for a refernce wind speed of $\overline{U}_{\text{ref}}(100 \text{ m} \leq z_{\text{ref}} \leq 400 \text{ m})$ based on
sampled WRF-simulated wind data (blue) and power estimates based on standard logarithmic wind speed profiles (red). These
results are verified against the QSM (Subsection 5.1, orange) and WT reference models (Subsection 5.2, green).

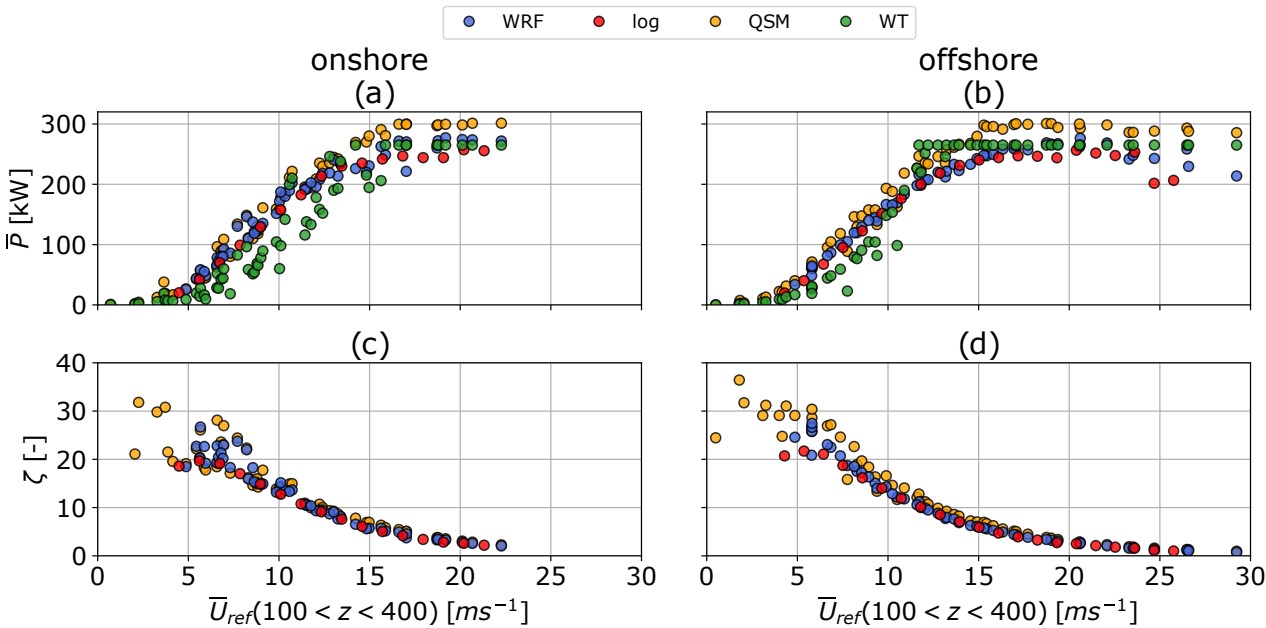

**Figure 20.** Average cycle power $\overline{P}$ and power-harvesting factor $\zeta$ for the onshore (a, c) and offshore (b, d) location as a function of average
wind speed $\overline{U}_{\text{ref}}$ between 100 and 400 meters. The data points obtained from `awebox` (blue), QSM (orange), and the WT model (green) are
based on WRF wind data and are compared to `awebox` data derived from standard logarithmic wind speed profiles (red).

The QSM and WT reference model use the same sampled WRF-simulated wind data. No cut-out wind speed is defined. The
cut-in wind speed of $\overline{U}_{\text{ref}} \approx 5 \text{ ms}^{-1}$ is the result of unconverged optimizations. The optimization algorithm was not able to
find a feasible trajectory for these low wind speeds, indicating that the wind is insufficient to keep the AWES aloft and produce
power. The QSM and WT model estimate power for these wind speeds. Rated power is achieved around $U_{\text{rated}} \approx 12-15 \text{ ms}^{-1}$,
depending on the wind speed profile shape. At these wind speeds the reel-out speed is almost constant while a constant reel-out
tension is already achieved at lower wind speeds (Figure 16).

The logarithmic wind speed profiles (Equation (6)) use roughness lengths of $z_0^{\text{onshore}} = 0.1$ and $z_0^{\text{offshore}} = 0.001$. Onshore,
the power predicted based on WRF wind data is often higher than the power predicted using logarithmic profiles (Figure 20
(a)). This is likely due to higher than predicted wind shear and the presence of LLJs that are not represented by logarithmic
profiles. Offshore, the logarithmic and WRF data are in close agreement with the logarithmic results because most of the
simulated wind profiles are more monotonic.

The $\zeta$ trends for both onshore (Figure 20 (c)) and offshore (Figure 20 (d)) conditions show a decrease with increasing wind speed and are consistent with the QSM. WT power fluctuates due to the choice of reference height. AWESs outperform WTs up to rated wind speed, particularly onshore where AWESs can take advantage of higher wind speeds aloft. Lower wind shear offshore reduces the need to operate at higher altitudes, reducing the benefit of AWESs. The QSM predicts the highest power, as anticipated, due to its simplified assumptions such as constant power during reel-out and reel-in and neglected mass.

## 7 Conclusions and outlook

This research outlines the optimal performance of single-aircraft, ground-generation AWES using sampled mesoscale WRF-simulated wind data and compares it to the average cycle power calculated using standard logarithmic wind profiles. It also describes trajectories, instantaneous performance, and trends in tether length and operating height. These analyses use one year of onshore wind data at Pritzwalk in northern Germany and one year of offshore wind data at the FINO3 research platform in the North Sea to drive the `awebox` optimization, which determines dynamically feasible, power-optimal trajectories. The annual wind data is categorized into k = 20 clusters of vertical wind velocity profiles U using a k-means clustering algorithm. To decrease the computational expense, three profiles based on the 5th, 50th, and 95th percentile of wind speed between $\overline{U}_\mathrm{ref}(100 \leq z_\mathrm{ref} \leq 400\,\mathrm{m})$ for each cluster are incorporated into the performance optimization model. The performance model uses a scaled Ampyx Power AP2 aircraft with a wing surface area of $A = 20\,\mathrm{m}^2$ and is subject to realistic tether and operational constraints. Our investigation into the impact of wind speed at reference height found that the a priori guess of $100 \leq z \leq 400\,\mathrm{m}$ is a good guess for the investigated AWES design and size. Optimal average cycle power is compared to a quasi-steady-state AWES model and a steady-state WT model.

The optimization model is able to determine power-optimal trajectories for complex, non-monotonic wind velocity profiles. The optimized results are only marginally lower than those obtained using the simplified QSM, which neglects the effects of gravity and only simulates a single reel-in and reel-out state instead of the entire trajectory. The predicted onshore AWES power exceeds the WT reference model. This is because AWES can adapt their operating altitude to benefit from higher wind shear or LLJs. Offshore wind velocity profiles are generally more monotonic and exhibit higher wind speeds, with less turbulence and wind shear. As a result, offshore winds produce average power that is similar to their logarithmic approximation. Due to the initialization of the `awebox` with a fixed number of loop maneuvers, which is not a variable in the objective function, high wind speed trajectories show loops during the reel-in period which reduces the average cycle power. This can lead to a deterioration of the trajectory at high wind speeds, as the optimizer struggles to stay within the tether tension and tether reeling speed constraint.

An investigation of the time series data show that the optimizer first maximizes tether tension by adjusting reel-out speed and angle of attack. With increasing wind speed the tether reel-out speed approaches the maximum reel-out speed limit and steadier. Up to rated wind speed, when average tether tension and tether reeling speed are maximized, the optimizer increases the deployed tether length and reduces the elevation angle to operate at optimal height. At higher wind speeds, the elevation angle increases to de-power the system and stay within design constraints. As a result, approximately 75% of the optimal

onshore and offshore operating heights are below 400 m. The offshore power curve appears to be independent of the reference height due to the lower number of non-monotonic wind speed profiles and lower wind shear. In contrast, the choice of reference height is more important for the onshore power curve.

The mesoscale wind simulations, which include a year's worth of wind data with a temporal resolution of 10 minutes, are analyzed and categorized for both onshore and offshore locations. The annual wind roses for heights of 100 and 500 m confirm the expected wind speed increase and clockwise rotation at both locations. Offshore shows a lower wind shear and veer than onshore. Annual wind speed statistics reveal that low wind speeds still occur at a fairly high probability up to 1000 m at both locations. The k-means clustering algorithm is able to categorize the wind regime and identify LLJs as well as various non-logarithmic and non-monotonic wind profiles. The primary factor in assigning a profile to a cluster appears to be wind speed, while the shape of the profile seems to have a lesser impact. Individual clusters produce coherent groups of similar wind profiles whose probability correlates with seasonal, diurnal and atmospheric stability variation. The k-means clustering method provides good insight into the wind regime, especially for higher altitudes where classification by Obukhov length is inadequate.

As a continuation of this study, the power curves and realistic wind conditions described here could be utilized to calculate AEP estimations. Further research is required into AWES power curves and their reference wind speed, which could be accomplished by deriving shape-specific power curves from normalized wind speed profiles or by considering the correlation between wind speeds at different reference heights. Future work should include a variable number of loop maneuvers as a variable in the optimization objective function. Using the same data and model, it is possible to investigate the annual and diurnal AWES power variation in comparison to WT performance. A parallel sizing study (Sommerfeld et al., 2022) using the same wind clustered wind data investigated the impact of mass and aerodynamic efficiency on AWES performance. Adding a design optimization to the `awebox` model could enable location-specific aircraft and tether investigation.

### 7.1 Acknowledgments and funding sources

The authors thank the BMWi for the funding of the "OnKites I" and "OnKites II" projects [grant numbers 0325394 and 0325394A] on the basis of a decision by the German Bundestag and project management Projektträger Jülich. We thank the PICS, NSERC and the DAAD for their funding.

`awebox` has been developed in collaboration with the company Kiteswarms Ltd. The company has also supported the `awebox` project through research funding. The `awebox` project has received funding from the European Union's Horizon 2020 research and innovation program under the Marie Sklodowska-Curie grant agreement no. 642682 (AWESCO).

We thank the Carl von Ossietzky University of Oldenburg and the Energy Meteorology research group for providing access to their high performance computing cluster *EDDY* and ongoing support.

We further acknowledge Rachel Leuthold (University of Freiburg, SYSCOP) and Thilo Bronnenmeyer (Kiteswarms Ltd.) for their help in writing this article, great, technical support and continued work on the `awebox` toolbox.

## 7.2 Author contribution

Markus Sommerfeld evaluated the data and wrote the manuscript in consultation with and under the supervision of Curran
Crawford. Martin Dörenkämper set up the numerical offshore simulation and contributed to the meteorological evaluation of
the data and reviewed the manuscript. Jochem De Schutter co-developed the optimization model and helped to write and review
this manuscript.

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

# Appendix A: Figures

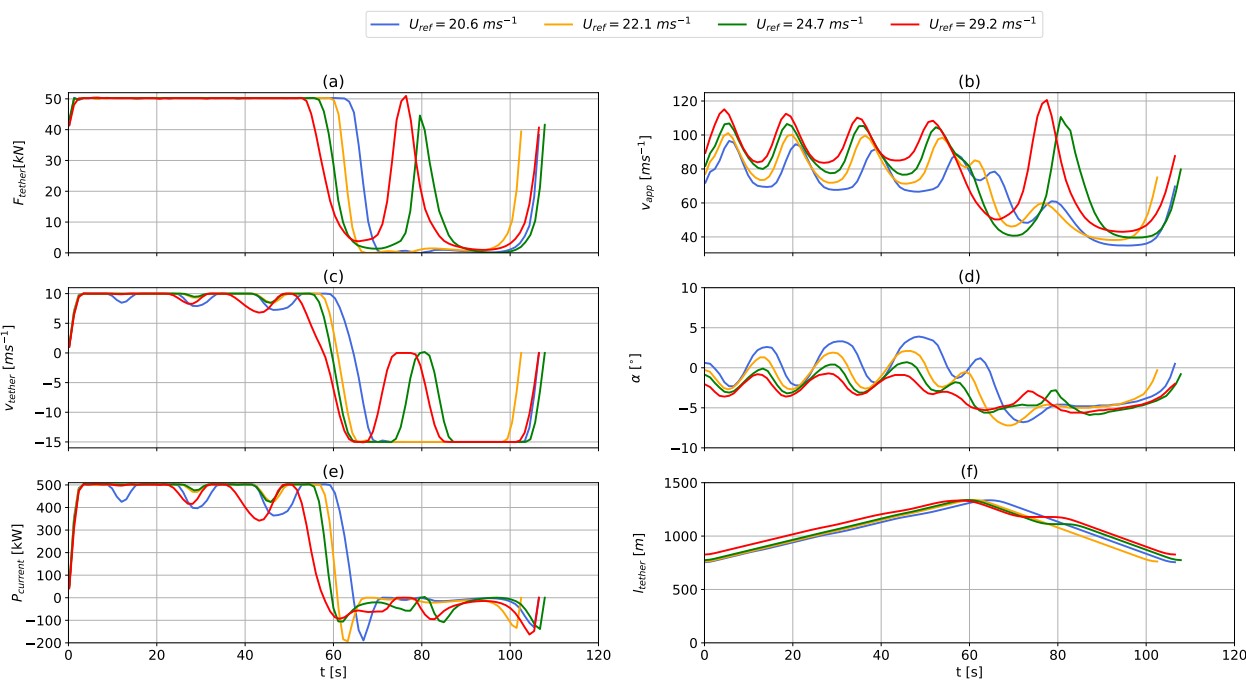

**Figure A1.** Time series of instantaneous tether tension (a), apparent wind speed (b), tether-reeling speed (c), angle of attack (d), power output (e) and tether length (f) over one pumping based on high wind speed offshore WRF-simulated wind data. The results correspond to the trajectories shown in Figure A2.

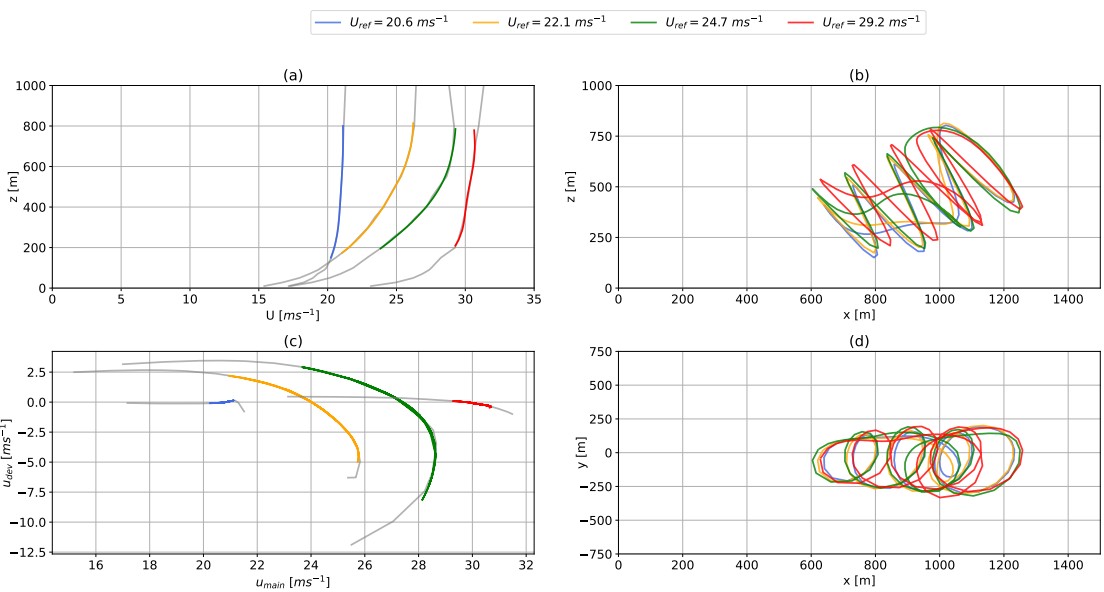

**Figure A2.** High speed WRF-simulated vertical offshore wind speed profiles (a), and hodograph (top view) up to 1000 m (c). The highlighted sections indicate Lagrangian polynomial fit of the wind velocity at operating height. Panel (b) and panel (d) show the side and top view of the corresponding `awebox`-optimized trajectories. The reference wind speed in the legend is $\overline{U}_{\mathrm{ref}} = \overline{U}(100\ \mathrm{m} \leq z_{\mathrm{ref}} \leq 400\ \mathrm{m})$. The results correspond to the time series shown in Figure A1.