# Peer review of "Impact of wind profiles on ground-generation airborne wind energy system performance"

_Wind Energy Science, 2020_

## Referee Comment (RC1) · Mark Schelbergen (Referee) · 22 Dec 2020

[wes, manuscript]copernicus

[1]MarkSchelbergen

[1]Faculty of Aerospace Engineering, Delft University of Technology, Kluyverweg 1, 2629 HS Delft, The Netherlands

Clustering wind profile shapes to estimate airborne wind energy production - Referee Comment

Mark Schelbergen

Mark Schelbergen (m.schelbergen@tudelft.nl)

Interactive
comment

[Figure]

**Offshore and onshore ground-generation airborne wind energy power curve characterization - Referee Comment**

December 22, 2020

**1    General comments**

A lot of the content reminded me of my own work (Schelbergen 2020), but with higher fidelity modelling and an alternative clustering approach. I feel that too little attention is put on the innovative part of your study, which is not so much the wind/performance modelling or clustering, but the way you calculate the AEP. The calculation is not described with enough detail and misses a strong foundation. There is no proof provided to draw conclusions about what AEP is realistic. Also the part on the AWES power coefficient is not very rigid and I expect the results to be specific to your methodology (initial guess optimizations). The $c_p$ that we know from conventional wind energy technology is derived from physical laws, the presented coefficient is lacking this strong foundation (Buckingham's Pi theorem).

Writing: The paper still feels like a draft version. It could benefit a lot from more precise writing and better choice of words. A lot of suggestions are provided in the technical

comments. There are quite a few bold statements in text, which either need more proof to back it up or could be formulated a bit more conservatively. E.g., at line 192 (page 10) it is mentioned that the cluster mean profiles that show decreasing wind speed above a certain height could be the result from choosing too many clusters. As these type of profiles are not covered by log/power law profiles, I would think that the added value of the clustering approach (and thus your methodology) is that you can identify such profiles from the data. Also there are a few statements that are not part of the scope of this paper. E.g., at line 372 (page 20) you briefly discuss the offshore WT tower design. This is a bit distracting and not adding a lot of value.

Figures, tables, and equations: The readability would improve a lot by including subla-bels (a, b, c, etc.) and referring to them in the text instead of e.g. "(right, 3rd from top)" for figure 10.

The legends of some figures are a bit confusing, e.g., the legend of figure 14 has different entries for line colors and styles and some combinations are missing (e.g. no round blue marker is included, while a square blue marker is and there is no separate general entry for a solid line). It would be good to be a bit more consistent.

The captions are often a bit long and sometimes include too much detail, e.g., the caption of figure 2 also states observations.

The size of the figures and tables are not very consistent, e.g., table 3 is huge for no clear reason.

Figures are placed in the appendix, while they are still covered in the text. In that case, in my opinion it's good to have them in the body of the paper.

Place the equations at the related text and introduce all variables in the text.

**2 Specific comments**

Introduction could be better structured, e.g., objective/contributions of the paper are not clear. The introduction is not very informative about closely related literature on AWE yield assessment. Also clustering is a separate section, but its roll in this study is not clear from the introduction.

I am missing a critical discussion about what data is suited for your methodology. In the conclusion you mention that the WRF data has a higher resolution than re-analysis data, but why would you need that? Also, doesn't the data assimilation used in the reanalysis lead to more trustworthy results? One of your conclusions is that evaluating the power output for 30 wind profiles (k=10 & p5/p50/p95 profiles) gives you the best estimate of AEP. So you discard most of the WRF simulated wind profiles for calculating the AEP. I would even expect that the data of a coarser model would be better suited for this context, as especially the p5 and p95 profiles are less prone to being eccentric, outlying wind profiles and thereby more representative.

Presenting the results with wind profiles could benefit from better structuring. Discuss choice for k at section 3. Based on my own work, I don't expect an elbow/kink in the inertia line. Be more specific about why you choose the number of clusters. Section 3 uses 10 representative profiles, then figure 4 only displays 4 profiles, figure 11 uses the p5, p50, p95 profiles for k=20, and figure 14 evaluates different k, but for only k=20 evaluates p5, p50, p95 (why not k=10?). Consider introducing your approach for these profiles at the start of each section. Also in the conclusions you state conclusions about the k=10 - p5, p50, p95 analysis which is not covered in the body of the paper.

In general the p5, p50, and p95 need to be better introduced. They are first mentioned at the end of section 4.5. After reading more I inferred myself that these are percentiles. I think their introduction needs to be accompanied by a figure for clarity. Also p50 is the median: how much does it differ w.r.t. the cluster mean?

[Figure]

I would not discuss this pre-optimization step for the tether sizing (section 4.3). Currently it is a bit confusing and does not add much value. It would be fine to consider the tether diameter as a given as this paper is about assessing performance metrics and not system design.

The non-linear grid on the x-axis of e.g. figure 12 needs to be justified. This part of your paper is where your work differs most from other studies: you should introduce it before the results section. part of your methodology I would expect the p5 and p95 causing a lot irregularities in the power curves and wind distributions, which vary depending on the metric which you put on the x-axis. As a result, the precision of the AEP which follows from integration over this axis will be low. In the figure differentiate between the p5, p50, and p95 data points.

My biggest concern about the AEP methodology is the mapping of the high dimensional space describing the wind profiles (60*2=120 dimensions?) to a 1D space (see figure 12). How would you justify integrating over only this 1D space? How accurate would this be? Do you also use p5, p50 and p95 wind profiles to the construct frequency distributions or only for the power curves?

How do you assess what AEP is realistic? For a very large k (i.e. k=number of WRF data points), the clustering output is the same as the input wind profiles. Applying your framework with a very large k is thus the same as doing the optimizations for every WRF wind profile, which would give you the best assessment of what is a realistic AEP. The trends in figure 14 don't converge to the diamond-marked value (k=20, p5,p50,p95-profiles) for very large k. This suggests that eccentric wind profiles are over-represented in the AEP calculation using the p5,p50,p95-profiles.

Mind that the $c_p$ of a wind turbine is not constant, but is lowered after the rated wind speed is reached. You compare the AWES power curve with one of a conventional WT. However, you don't justify your choice for the rotor area of the WT. From what I understand, you conclude that $c_p = 0.3$ gives a good agreement with the AWES power

curve, but it's not clear why you change the $c_p$ and not the area. As a result, the comparison is not fair.

How would you use $c_p$/chord in practice for power estimation? I think it's still a rather challenging task as you still need to determine the relation between $l_{path}$ and the wind speed. More importantly, for WTs $c_p$ is often used to quantify the efficiency of the energy conversion. I don't think the given formulation is a good metric for AWES efficiency. Take for example the (hypothetical) situation where we have a uniform wind field and let's neglect the variation of the air density. As you implied earlier (line 356): the mean cycle power is not very sensitive to the flight path length. So for the same kite size, a shorter and longer pumping cycle with the same mean cycle power will give me completely different values for $c_p$ while the AWES efficiency did not change.

Dividing cp by c is the same as having the area of the kite multiplied by the path length in the denominator of equation 5 - why not use the wing area instead of the swept area in the first place and leave out the path length?

**3 Technical corrections**

**3.1 Abstract**

page 1

line 5: "A universal" instead of "An ..."

line 5: What is the problem with power curves for log profile/power law wind conditions?

line 6: "complex tether and drag losses" - why does this lead to a more problematic power curve description? Also WTs occasionally operate in wind

conditions that are not covered by the assumptions made for determining their power curve e.g. low level jets.

line 7: The role of "rotor area normalization" to the power curve description is not clear. One can compare the harvesting efficiency of a WTs with different scales by normalization, but normally the power curve just characterizes the (non-normalized) power.

line 7: Not clear where "Therefore" refers to

line 14: put "with wind speed" after "decreases"

**3.2 Introduction**

page 1

line 20: WTs reach above 100 m

line 22: Use acronym for wind turbines

line 24: The list items are a bit random, suggestion: "3-bladed HAWT with conical tower" - as to my knowledge you don't find commercial HAWTs without nacelle and generator.

page 2

line 27: Replace "route" by "concept"

line 29: Reel-in description is a bit simplistic: flexible kite is really pulled back in, whereas a rigid wing utilizes its gliding capabilities.

line 34: State that power curves are in general only used for a preliminary analysis.

line 35: Not clear why the reference to Malz is needed here, does it belong to the previous sentence?

line 36: "wind speed magnitude" implies that the wind field in which an AWES operates can be described with one magnitude - needs some more explanation.

line 37: Ground-gen does not operate at a single altitude: "optimal trajectory" includes information about altitude.

line 37: Suggestion: split sentence: "Simple .."

line 41: "most ... studies": it would be relevant to know which studies use an alternative approach.

line 47: Discuss directly using measurements/LiDAR data for assessing wind resource (no weather modelling).

line 48: Discuss your methodology (starting from "Results in ...") in a separate paragraph. Also include here what exactly the contribution is of this paper. You already touch upon this in the paragraph starting at line 32, however it is not very concrete (probably you want to the content of this paragraph down here).

line 48: As I understand, you previously corrected WRF with LiDAR for the Pritzwalk location. Therefore none of the mentioned reasons for not using apply LiDAR apply here. So how do you justify using purely WRF data?

line 54: Suggestion: Section 2 introduces the WRF model set-up and compares the onshore and offshore wind resource that follow from the WRF simulations.

line 55: Would be good to mention clustering before when introducing methodology.

line 59: replace "derive" by "produce", replace "This includes" by "These include"

page 3

line 61: As I read it: the coefficient **definition** directly follows from the results, however I don't think this is what is meant.

**3.3 Wind data**

page 3

line 64: Representative for what?

line 67: AWE might be promising for other type of locations where it does not have to compete with WTs.

line 71: Why use different periods?

line 75: replace "in" with "with"

line 79: replace "on" with "in"

line 78: replace "The focus of this study is not on the detailed comparison between mesoscale models, but on AWES performance subject to representative onshore and offshore wind conditions determined based on clustered wind profiles (described on section 3). To that end" by "For the assessment of AWES performance"

line 80: What is adequate?

line 80: remove "data" in between sentences

line 82: Why use different data sources for boundary conditions.

line 88: replace "h" by "hours"

line 88: Why use different approaches to simulate 1 year?

page 4

figure 1: It's hard to assess the topography from this figure, leave out "topography" in caption, or add information on color scale.

line 91: "High-Performance Computing" without capitals?

line 94: Wind directions without capitals

line 94: Dominant wind direction offshore is southwest for both 100 and 500 m.

line 94: Using "rotating" is a bit strange here: the wind direction changes or the wind turns.

line 96: Start sentence with "The"

line 97: "Offshore conditions .." is a bit vague. How does the 10 degrees relate to the 5 degrees of the preceding sentence? If the wind would always veer 10 degrees than so would the average wind direction.

page 5

line 98: "the same westerly wind direction at high altitude" - what do you mean? Figure 2 shows that offshore the wind is predominantly southwest.

line 99: Add point at end of sentence

line 99: Replace "The relative wind speed increase of" by "The wind shear at", and remove "and the already high wind speeds at lower heights"

line 102: replace "distribution" by "distributions at each individual height level"

line 102: replace "These statistics give an insight into the overall wind conditions, but the actual profile shapes" by "These distributions give insight into the wind speed statistics at the individual heights, but not onto the statistics of the wind profile shapes."

line 105: replace "have a fairly narrow range" by "are relatively low"

line 106: replace "up to high altitudes" by "for the full height range"

line 106: replace "This leads to the development of" by "The distributions show"

line 109: "Such multimodal distributions at higher altitudes are better described by the sum of two or more probability distributions" - isn't this the definition of multimodal?

line 113: "As mentioned above, the relative wind speed increase with height is less pronounced offshore than onshore." - why mention it twice?

line 114: suggestion "Conventional WTs benefit from low wind shear offshore .."

line 115: replace "However, offshore AWES will also benefit from higher offshore winds and move offshore for other reasons such as safety or land use regulations" by "Nevertheless, also AWES benefit from low wind shear. Among the reasons for placing AWES offshore are safety and land use regulations."

page 6

figure 2: State that these results reflect single locations. Leave out "On average wind direction onshore rotates about 14 åŮę while offshore winds rotate about 5 åŮę between 100 and 500 m. Onshore shows a higher wind shear due to higher surface roughness and relatively high wind speeds offshore." - this belongs in the text.

line 116: "Another benefit of offshore AWES in comparison to conventional WT is the smaller and cheaper support structure." - Statement is a bit misplaced in the wind resource discussion and needs a reference. Also the arguments are one-sided, there are many reasons why not to put AWES offshore.

line 118: replace "categorized" by "characterized"

page 7

figure 3: Clarify that this is not a 2D histogram, but independent histograms/distributions per height. State that these results reflect single locations.

line 126: Not clear if this is the classification as presented in the table.

line 129: Is the ocean always warmer? Remove "likely"

line 131: Not sure what the use is of this paragraph: only covers literature without touching upon the results.

**3.4 Clustering of wind conditions**

page 8

table 2: State that these results reflect single locations.

line 136: Statement is not very informative, rephrase to emphasize that it's not just time- and space-averaged wind velocities that effect the power output of wind energy systems, but also the variation in time and space.

line 138: Elaborate on excessive averaging.

line 141: replace "proxy" by "metric", replace "a metric that" by ", which"

line 143: Only one study is listed

line 143: Not clear what diverge with height means: large wind speed spread at high altitudes?

line 144: Do you mean to say that there is no correlation between the wind speed profile over a high elevated layer and the surface-based stability?

line 146: Also grouping based on stability is "based on data similarity": be more precise.

line 146: What does the reader need to see in the appendix? - Don't refer to results before the explanation of your methodology is finished.

line 148: Mathematical clustering is a vague term: explain. You use clustering, classifying, categorizing, and binning interchangeably: be more consistent.

line 154: The u-component is per definition along the x-direction. Explanation is not clear: also $u_{main}$ and $u_{deviation}$ are not throughout this paper anymore - so why introduce them here?

line 158: Explain what a data point consists of. Also not clear that each data point is assigned to the closest centroid. Replace "defined" by "represented"

line 159: Suggestion: "The clustering finds the centroid positions that minimize .."

line 160: Distances between?

line 161: Rephrase: the centroid will at best coincide with a data point by chance

page 9

line 162: I misunderstood at first: I thought you were talking about the labels assigned to each data point. Leave out mentioning the initialization as it is confusing. Just mention that the label number that each cluster gets is rather random and does not have any mathematical meaning.

line 164: replace "Later evaluation uses clusters sorted by average wind speed up to 500 m." by "As presented, the resulting clusters sorted by average wind speed up to 500 m."

figure 4: Use the same color scheme for the clusters as for figure 5. It's unclear from the right column of figure 4 what is plotted on the y-axis as there is no grid/ticks.

line 168: "inertia reduction becomes marginally small with increasing number of clusters" - this is not completely true: above all the elbow method says that kinks in the inertia trend indicate sensible choices for k. It's hard to observe them with evaluating so little values of k. Best to just use a step size of 1.

line 169: that doesn't make inertia meaningless

line 170: add space after hyphen

page 10

line 171: table 1 doesn't show any difference between the vertical levels (grid) of the two analyses. How different are they? I would not expect the vertical grid to have a large effect. More probable is a larger spread of the centroids.

line 172: replace membership by a more precise explanation

line 176: also if you would use a fixed (not random) initialization the cluster would not be arranged in a logical order, suggested: "Note that the order of the clusters is random and does not follow any logic."

line 176: It's still on the y-axis.

line 178: Too many I don't find very likely. At least the mean silhouette score always decreases with k. Do you have a reference?

line 179: Not clear what you mean with "the continuous nature of wind which results in a high cluster proximity" and how this effects the silhouette scores - please expand.

line 180: For readability I would suggest not to jump to the conclusions of the next section.

line 181: Replace "intersect .." by "are grouped together with monotonic wind profiles"

line 182: closing parenthesis missing

line 184: Is there a reason for using first wind speed and after velocity? If not, rephrase.

line 185: replace "comprising .." by "WRF-simulated wind speed profiles that are input for the clustering"

line 185: suggestion: "Within a cluster, the wind speed profiles span.."

line 189: False statement: this explains the higher inertia, not the silhouette score. Also the onshore clusters are distinct.

line 191: I don't see why "directional differences" result in a decrease of wind speed with height - explain.

line 192: What about small-scale weather phenomena?

line 192: Why too many clusters (in which context)? I think it is very insightful to find these type of cluster mean wind profiles. Them having a lower probability makes it even more interesting to evaluate why the clustering gives these clusters as a result as normally k-means clustering tends to produce equally sized clusters.

line 195: Is the wind speed inversed or the wind shear?

line 197: replace "determining" by "dominant" and replace "stacked" by "ordered"

line 200: Suggestion: add reference to Schelbergen

page 11

line 204: For which application/analysis are long term averaged wind speed profiles considered? Using a averaged profile **shape** is commonly done for AEP calculations, but that's a somewhat different approach.

figure 5: The wrf-simulated wind profiles only show as a filled area, so not very informative - but I suppose it doesn't hurt either.

page 13

line 220: The name of the "other" category suggest to me that these conditions are exceptional/not often occurring, however your analysis shows the opposite. How should I physically interpret this category?

line 220: What do you mean with wind power assessment? Why is the impact of low wind conditions low? I don't understand the comparison.

line 221: Basically what you're saying is that how often wind conditions occur (the wind speed distribution) has a large effect on the AEP? - I would say that the reader is familiar with this.

**3.5 AWES trajectory optimization**

page 13

  line 231: Very technical start of the section - readability would benefit from further introduction.

  line 232: Which unstable dynamics are you referring to?

  line 233: Which multiple inputs, multiple outputs?

page 14

  line 238: Which simulated profiles? Aren't you using the cluster means?

  line 239: replaced "during" by "of"

  line 247: What is the main wind direction?

  line 252: Suggestion: "We use the aerodynamic model of the Ampyx AP2 (Malz et al., 2019; Ampyx)"

  line 253: The footnote is confusing as you mention that no other data is available while you are referring to a source.

  line 259: Introduce kappa and bring equation 2 forward.

  line 260: "This results in an overestimation of output power and lower cut-in speed in comparison to a heavier aircraft." - unclear how you come to this conclusion.

  line 263: Mentioning focus of paper is a bit misplaced, better save that for the outlook.

page 15

  figure 9: Why mentioning Loyd in caption?
line 264: rephrase

line 265: Leave out or mention how you would estimate it.

equation 2: Position at related text and introduce properly.

line 270: mention conditions for which this is valid.

line 270: Last part of sentence is confusing. Suggestion: ".. , thereby the reel-out speed is expected to remain below 10 m/s as the wind speed hardly exceeds 20 m/s.

line 271: Implies that the ratio is fixed, however I don't think that's what's meant. - example values 10/15 is superfluous.

line 272: I would not discuss this pre-optimization step. As you mentioned yourself: this paper is about assessing performance metrics and not system design. Consider the tether diameter as a given.

page 16

line 280: No need to state subsection number, table number suffices.

line 281: why "or"?

line 282: add "coefficient" - why do you use a linear relation?

line 285: replace "prevailing" by "can be adapted to", replace "dynamically" by "continuously"

line 286: replace "AWES" by "AWESs"

line 290: Isn't 19 m/s a bit low? - this is equivalent to a mild storm

line 293: Explain what a p-value is: add a figure to explain.

line 294: Why average wind speed up to 500 m?

line 294: On line 176 you mention that a representative k is 10. Better to choose k already at section 3, instead of quickly mentioning it here.

page 17

> line 296: Rewrite "cluster centroids.. implemented"
>
> line 297: Not clear when you use the p5,50,95 and when you don't and how you use them if you do. Please rewrite paragraph. So you only use these p-values for determining the power curve? I would say this is the most important part of this study so it should get more explanation.

equation 3: Move to the related text.

> line 304: Would the analysis benefit from running the optimization using multiple starting points?
>
> line 306: remove hyphen: "inequality"
>
> line 306: Which equality constraints? You only mentioned inequality constraints.
>
> line 307: As far as I know, an optimal control problem always uses discretized control intervals, irrespective of if direct collocation is used or not.
>
> line 307: missing space after dot
>
> line 307: Is this the input you give to awebox for generating an initial guess? Explain estimated aircraft speed - this will vary between reel-in and reel-out and also within the reel-out phase over one loop.

**3.6 Results**

page 17

> line 317: Shouldn't it be "average wind speeds for different height ranges"?
>
> line 320: "Rayleigh distributed log-profiles" is not very precise
>
> line 323: "representative" does not fit the context, list which profiles you have chosen.

page 18

line 325: missing closing parenthesis, replace "figures" by "figure"

figure 10: In caption list which profiles are plotted. Also describe what is high-lighted using the colors in the left column. Remove "The deviation of the colored lines is caused by the approximation of discrete data points with Lagrange polynomials." - too much detail for caption.

line 328: replace "optimization" by "modelling"

line 328: Wind speed vs wind velocity profiles is a bit confusing. Just mention that only the wind speed is plotted and that changes in wind direction with height can be seen in the lower left plot.

line 330: Didn't you introduce the "rotated u and v component" as $u_{main}$ and $u_{deviation}$? - then also use them here

line 330: replace "wind velocity components as experienced by the AWES in color" by "with the part of the profiles corresponding to the height range swept by the kite in color"

line 332: not clear which are the onshore and which the offshore profiles.

line 333: first should be x-z plane

page 18

line 335: perpendicular to x is both y and z-direction

line 335: What is unrealistic about them?

line 338: Can you identify whether the system is de-powering or not? If so than you can differentiate between the two.

line 338: What about de-powering by increasing the reelout speed?

line 342: So for an elevated path, you don't want to fly at maximum $cl^3/cd^2$? This is not what we observe in figure 10: the most elevated path has the lowest angles of attack during reelout.

line 347: So you're relating this observation in the reel-in phase (reaching max reel-in speed) to an observation in the reel-out phase (de-powered loop)? Isn't it more likely that the de-powering is triggered by reaching max tether force during reel-out?

line 355: Which previous analyses?

line 356: Not completely: zero would probably not give you the best performance.

page 19

line 363: Clarify that you're not just using the 4 profiles that were treated in the previous subsection.

line 363: What is $z_{operation}$? I would expect a height range, not a single value.

line 364: Replace "emerge"

line 365: add comma after onshore, correct for this in the rest of the paper

line 365: remove "and typically higher winds offshore" from end of sentence and mention in a different sentence that generally tether length increases with wind speed

page 20

line 369: "on the other hand" - comparison is not clear

line 372: "This also has implications for tower-based.." - Leave the WT discussion out. It's well known that offshore turbines have lower towers than onshore.

line 376: The distributions for 25 and 50m2 are not that much different and is worthwhile mentioning.

line 378: Figure 11 shows only onshore 1% exceeding 600 m for 20m2 and offshore 0%

figure 11: Differentiate between p5, p50 and p95 using for example different markers. Put occurrences on the horizontal axis. If I understand correctly your only plotting the results of 60 optimizations: this would be more clear by plotting occurrences.

page 21

line 384: replace "unanimously accepted" by "consensus"

line 385: replace "wind speed probability distribution" by "wind resource model"

line 387: I would just mention that AWESs generally operate in a larger height range

line 389: So this leaves you with a non-linear grid on the x-axis? Also the order is different for every reference height range? - If so, mention this. I would expect the p5 and p95 causing a lot irregularities in the power curves and wind distributions. Would it be possible to show data points represent p5, p50 and p95 powers? About using the power curves for integrating: basically you're trying to map this high dimensional space describing the wind profiles (60*2=120 dimensions?) to a 1D space. By doing so you loose a lot of information on the wind profiles. How would you justify integrating over only this 1D space? How accurate would this be? Do you also use p5, p50 and p95 wind profiles to the frequency distributions? Is the area underneath your frequency distributions 100% in that case?

line 393: "respective AWES operating altitude" - why using 60 - 629 m and 59 -551 m? Not all trajectories fly over this full range.

line 394: "black" instead of "red"

line 395: The power curve shows a varying $c_p^{WT}$

line 397: what is the value of the rotor disc area?

equation 4: $U^3$

line 400: What do you mean with limit specific designs?

line 405: Isn't this just due to your choice for the rotor disc area?

line 406: Explain more precisely: if wind speed is monotonically increasing with height $u_{100m} < u_{100-400m}$. So the data points move to the when plotted against $u_{100-400m}$ with respect to when they are plotted against $u_{100m}$.

line 409: replace "overlap" by "lie on top"

line 409: Not clear why 200m ref would be better, since you mention in the previous sentence that every reference height gives roughly the same power curve.

line 410: with divergence you mean large difference?

line 411: Also $c_p^{WT}$ is lowered after the rated wind speed.

page 22

figure 12: On y-axis is the energy contribution: not E average

line 414: That the flight trajectory influence mean cycle power should be clear from the literature study.

line 415: An overkill to use a separate plot when only adjusting the cp. What if you change the rotor disc area?

line 416: Leave out "A better .. future study.". Mention in the outlook at the end of paper.

line 418: Rephrase "integral multiplication"

line 419: replace "Its total" by "The area underneath the distribution"

page 23

line 420: Rewrite "shifts towards higher wind speeds due to .. higher wind speeds"

line 421: How is that similar?

line 426: Which effect scales with size? - Isn't this specific to your methodology, i.e. the integration of the equivalent 1D wind profile space.

line 428: Leave out "This indicates that onshore wind conditions favor higher operating altitude due to higher wind shear."

line 429: What reduction? The AEP goes up.

line 432: "This main difference" - which difference is explained exactly?

line 433: The cp is very much dependent on the area you chose, which you haven't justified. Also, the onshore 20m2 curve for reference height 100m lies on top of that of the wind turbine.

line 434: "wind speed along the actual AWES trajectory": this is a varying property - how would you use it as a reference?

line 436: Figure 11 shows that the operational height range of 100 m are common. Why would a 500m height range be a good reference in that case?

page 25

line 439: How would you use this power coefficient in practice for power estimation? I think it's still a rather challenging task as you still need to determine the relation between $l_{path}$ and the wind speed. More importantly, for WTs cp is often used to quantify the efficiency of the energy conversion. I don't think the given formulation is a good metric for AWES efficiency. Take for

example the (hypothetical) situation where we have a uniform wind field and let's neglect the variation of the air density. As you implied earlier (line 356): the mean cycle power is not very sensitive to the flight path length. So for the same kite size, a shorter and longer pumping cycle with the same mean cycle power will give me completely different values for cp while the AWES efficiency did not change.

line 441: You mean average wind speed?

line 442: Air density at which height?

line 444: $z_{operation}$ suggests a single height, but probably needs to be a height range? Change throughout the paper.

line 451: cp is not truly "velocity profile independent": especially at low wind speeds you see a vertical spread for fixed wind speed. How sensitive is power output to the wind profile shape?

line 452: sentence does not explain any difference

line 452: rewrite "The difference .. remains almost constant (see sub-section 5.1 and 5.2)" - unclear what you're saying.

line 457: Dividing cp by c is the same as having the area of the kite multiplied by the path length in the denominator of equation 5 - why not use the wing area instead of the swept area in the first place and leave out the path length?

line 459: Fit looks poor for larger wind speeds. Why do you only plot the line up to 18 m/s?

line 462: "contrasts" replace by "compares"

page 26

figure 13: The markers are not connected by the lines. Why is this?

page 27

equation 7: leave out h/year

    line 472: remove "static"

    line 473: use "probability distributions"

    line 475: replace "justified" by "justify"

    line 480: I think it is a bit bold to say that they are averaged out. You could reason that they are underrepresented. Also show with a figure similar to 12 how both computations are different.

**3.7 Conclusions and outlook**

page 27

    line 490: replace "characterized" by "evaluated"

    line 492: replace "deduced" by "uses"

    line 493: replace "Representative wind velocity profiles based on k-means clustering were chosen to reduce computational cost." by "A representative wind resource model is deduced using the results of clustering wind profiles" - and place after discussing WRF

    line 494: Do you need this high resolution if you drastically simplify the wind model in the end?

    line 497: "acceleration"?

    line 511: replace "implemented" by "given as input"

    line 514: replace "fast" by "high"

    line 517: See earlier comments on WT power coefficient.

    line 519: remove part about social acceptance: not touched upon in body of paper

line 522: replace "collapse .." by "yields a location and size independent metric"

line 526: You did not evaluate the AEP for 10 clusters and the p5, p50, p95 profiles if I'm not mistaken. So how do you get to this conclusion? If the AEP is higher than for all other calculations I wouldn't expect it to be the most realistic.

line 529: I don't see this back in Figure 14, it shows quite a large bias between the log and cluster AEP.

line 536: Why do you expect this?

line 538: Be more general in this paragraph.

---

## Referee Comment (RC2) · Anonymous Referee #2 · 26 Jan 2021

5: An universal -> A universal

8: annual energy prediction (AEP) -> production

249: I'm curious about how pressure & density vary with stable vs. unstable conditions and how much that affects power.

271: Why is a reel-out to reel-in ratio used? Is this a combination of a motor torque constraint and the lift during reel-in and reel-out?

279: Assumed lift and drag on reel-in and reel-out should be included here.

280: Was a power constraint used? It's implied in other places.

357: I'd address elevation angle here; based on figure 10, it looks like the optimizer found a common optimal elevation angle for several of the cases, which links tether length and altitude. Vander Lind 2013 calculated an optimal elevation angle for fly-gen systems assuming an exponential wind profile; I'm curious how close this elevation angle is.

398: Missing a $U^3$?

440: l_path and A_swept aren't in table 3

459: The fit for cp is a function of c_wing (and because AR is constant, a function of Aswept) so it's not non-dimensional and it's not clear how generalizable it is (changes in AR or L/D). I'm curious about whether another definition of cp may also be comparable to conventional wind turbines but work better. The Loyd paper (see eqs. 1 and 16) shows a limit on a cp (4/27 $CL^3/CD^2$) defined by wing area. What does your data show for a cp defined by Awing? Or if you express cp as a function of L/D or $CL^3/CD^2$?
* * *

---

## Author Comment (AC2) · 9 Jun 2021

**Response to referee 2 wes-2020-120**

Markus Sommerfeld

January 2021

**1   Author response**

Dear referee 2, Thank you very much for your helpful comments to our manuscript, "Offshore and onshore ground-generation airborne wind energy power curve characterization", wes-2020-120. Please accept my apologies for the delayed response.

A lot of time was spend on the revision of this paper including re-clustering WRF wind data, re-running optimizations, re-evaluating results and re-writing major sections of this manuscript. We added a reference section which compares optimization results to quasi steady-state (QSS) AWES and WT reference models. We agree with the criticism to the AWES power coefficient and removed it. Instead, we implemented a brief description and investigation using the harvesting factor **?**. Please find detailed responses below. I am looking forward to your comments to further improve this paper.

Sincerely, Markus Sommerfeld

**2 Specific comments**

**Line 5** "A universal" instead of "An ..."

- – implemented

**Line 8** annual energy prediction (AEP) → production

- – implemented

**Line 249** I'm curious about how pressure & density vary with stable vs. unstable conditions and how much that affects power.

- – That would be interesting to investigate, but was deemed out of scope for this analysis. I would expect the impact on power to be rather small.

**Line 271** Why is a reel-out to reel-in ratio used? Is this a combination of a motor torque constraint and the lift during reel-in and reel-out?

- – This was a design choice based on conversations with a ground station developer. Motor torque is limited by tether tension.

**Line 279** Assumed lift and drag on reel-in and reel-out should be included here

- – It is hard to a priori estimate lift and drag as it highly depends on angle of attack, side slip angle and tether drag. Therefore, I would refer to figure 9 which summarizes representative lift, drag and pitch moment coefficients.

**Line 280** Was a power constraint used? It's implied in other places.

- – Power was indirectly limited by tether speed and tension constraints.

Line 357 I'd address elevation angle here; based on figure 10, it looks like the optimizer found a common optimal elevation angle for several of the cases, which links tether length and altitude. Vander Lind 2013 calculated an optimal elevation angle for flygensystems assuming an exponential wind profile; I'm curious how close this elevation angle is.

– Brief elevation angle analysis added.

Line 398 Missing $U^3$?

– implemented

Line 440 $l_{path}$ and $A_{swept}$ aren't in table 3

– section removed

Line 459 The fit for cp is a function of $c_{wing}$ (and because AR is constant, a function of $A_{swept}$) so it's not non-dimensional and it's not clear how generalizable it is (changes in $AR$ or $L/D$). I'm curious about whether another definition of cp may also be comparable to conventional wind turbines but work better. The Loyd paper (see eqs. 1 and 16) shows a limit on a cp ($4/27 CL^3/CD^2$) defined by wing area. What does your data show for a cp defined by $A_{wing}$? Or if you express cp as a function of $L/D$ or $CL^3/CD^2$?

– section removed

---

## Author Response (AR1)

**Response to referee 1 wes-2020-123**

Markus Sommerfeld

January 2021

**1 Author response**

Dear referee 1,

Thank you very much for your helpful comments to our manuscript, "Offshore and onshore ground-generation airborne wind energy power curve characterization", wes-2020-120. Please accept my apologies for the delayed response.

A lot of time was spend on the revision of this paper including re-clustering WRF wind data, re-running optimizations, re-evaluating results and re-writing major sections of this manuscript. We added a reference section which compares optimization results to quasi steady-state (QSS) AWES and WT reference models. We agree with the criticism to the AWES power coefficient and removed it. Instead, we implemented a brief description and investigation using the harvesting factor [1]. Please find detailed responses below. I am looking forward to your comments to further improve this paper.

Sincerely, Markus Sommerfeld

**2 Specific comments**

- It depends on the purpose of your investigation. I would generally argue that higher resolution data results in better power and AEP predictions.

- WRF uses re-analysis data as boundary conditions and therefore also assimilates measurements.

- It depends on the purpose of your investigation. If you are not interested in the optimal trajectories an AWES would fly during non-monotonic / eccentric wind conditions than you would not simulate them. p5 and p95 are not necessarily more eccentric.

- The main contribution of this paper is to investigate AWES trajectories, power and AEP predictions based on more realistic wind conditions.

- Mapping high dimensional wind data onto 1D space is the standard approach for WT as well, even more as vertical changes in wind speed are just represented by a single average speed. We are trying to derive a simple, easy to understand representation for AWES power which can be communicated with industry, wind park developers and AWES designers. This is a complex problem which needs more investigation, but hopefully is a step into the right direction.

**2.1 Abstract**

**Line 5** "A universal" instead of "An ..."

  – implemented

**Line 5** What is the problem with power curves for log profile/power law wind conditions?

  – Only log might not be enough to properly represent the complex wind resource and describe realistic AWES trajectories and operating conditions.

**Line 6** "complex tether and drag losses" - why does this lead to a more problematic power curve description? Also WTs occasionally operate in wind conditions that are not covered by the assumptions made for determining their power curve e.g. low level jets

  – rewritten

**Line 7** The role of "rotor area normalization" to the power curve description isnot clear. One can compare the harvesting efficiency of a WTs with differentscales by normalization, but normally the power curve just characterizes the(non-normalized) power.

  – rewritten

**Line 7** Not clear where "Therefore" refers to

  – rewritten

**Line 14** put "with wind speed" after "decreases"

  – rewritten

**2.2 Introduction**

**Line 20** WTs reach above 100 m

  – Rewritten for clarity.

**Line 22** Use acronym for wind turbines

  – implemented

**Lines 24** The list items are a bit random, suggestion: "3-bladed HAWT with con-ical tower" - as to my knowledge you don't find commercial HAWTs without nacelle and generator.

  – removed nacelle and generator. Focus on main attributes of HAWT & it's singular concept

**Line 27** Replace "route" by "concept"

  – implemented

**Line 29** Reel-in description is a bit simplistic: flexible kite is really pulled back in,whereas a rigid wing utilizes its gliding capabilities.

– Rewritten for clarity. This sentence is kept general for the introduction section and because it is not the focus of this paper.

**Lines 34** State that power curves are in general only used for a preliminary analysis.

– Not implemented. It is expected that the reader already knows that.

**Line 35** Not clear why the reference to Malz is needed here, does it belong to the previous sentence?

– implemented

**Line 36** "wind speed magnitude" implies that the wind field in which an AWES operates can be described with one magnitude - needs some more explanation.

– I disagree. Both magnitude & profile shape including direction variation with height are clearly stated in the sentence.

**Line 37** Ground-gen does not operate at a single altitude: "optimal trajectory" includes information about altitude

– Kept as is. Right, trajectory includes altitude information, but the point is that both the shape of the trajectory as well as altitude change.

**Line 37** Suggestion: split sentence: "Simple .."

– Kept as is

**Line 41** "most ... studies": it would be relevant to know which studies use an alternative approach.

– Basically every study I found uses simple log or exponential or global reanalysis model. Added an additional sentence and cited your paper.

**Line 47** Discuss directly using measurements/LiDAR data for assessing wind resource (no weather modelling).

– Added an explanatory sentence

**Line 48** Discuss your methodology (starting from "Results in ...") in a separate paragraph. Also include here what exactly the contribution is of this paper. You already touch upon this in the paragraph starting at line 32, however it is not very concrete (probably you want to the content of this paragraph down here).

– Added paragraph and contributions through summary of main findings

**Line 48** As I understand, you previously corrected WRF with LiDAR for the Pritzwalk location. Therefore none of the mentioned reasons for not using apply LiDAR apply here. So how do you justify using purely WRF data?

– This comment is unrelated to this work. I did not use LiDAR because 1. only 6 mon of data available and that is not enough to generate annual statistics; 2. Data availability decreased significantly with height, making the combination of simulation and measurement necessary; Therefore, using WRF is the better approach here. Also, can be found in my previous papers.

**Figure 54** Suggestion: Section 2 introduces the WRF model set-up and compares the onshore and offshore wind resource that follow from the WRF simulations.

– implemented

**Line 55** Would be good to mention clustering before when introducing methodology

– added a sentence above

**Line 59** Replace "derive" by "produce", replace "This includes" by "These include

– implemented

**Line 61** As I read it: the coefficient definition directly follows from the results, however I don't think this is what is meant.

– Removed the power coefficient section and replaced with harvesting factor defined in "Airborne Wind Energy: Basic Concepts and Physical Foundations" (DOI: 10.1007/978-3-642-39965-7_1)

**2.3 Wind data**

**Figure 64** Representative for what?

– added "onshore and offshore"

**Line 67** AWE might be promising for other type of locations where it does not have to compete with WTs.

– No statement was made about where AWES might be deployed or are promising. This is rather a statement on the quality of the selected site, i.e. the location has good wind conditions.

**Line 71** Why use different periods?

– For this study we make several generalizing assumptions, given the that this is not a site or time specific analysis. We assume that the periods are not important and the wind data are representative of not only that location, but also on- and offshore wind conditions in general. Generating these mesoscale data is costly. The simulated time for onshore location coincides with the measurement campaign. The offshore data was part of a different project.

**Line 75** replace "in" with "with"

– implemented

**Line 79** replace "on" with "in"

– implemented

**Line 78** replace "The focus of this study is not on the detailed comparison between mesoscale models, but on AWES performance subject to representative onshore and offshore wind conditions determined based on clustered wind profiles (described on section 3). To that end" by "For the assessment of AWES performance"

– Not sure what you mean

**Line 80** What is adequate?

– it is adequate for our application: preliminary AWES performance assessment. It his not a specific site assessment for a particular AWES design. It is not important that both WRF models are slightly different. It is not important that we chose different time frames.

**Line 80** remove "data" in between sentences

– implemented

**Line 82** Why use different data sources for boundary conditions

– Onshore WRF simulation was performed while ERA-Interim was still supported, i.e. before 31 August 2019.

**Line 88** replace "h" by "hours"

– implemented

**Line 88** Why use different approaches to simulate 1 year?

– We are aware that this raises some questions. However, we believe that both WRF models provide adequate wind data for our purposes (As mentioned in line 75). This is solely the result of ongoing research, costly & time consuming simulations, collaboration with the awebox team and work on other projects.

**Figure 1** It's hard to assess the topography from this figure, leave out "topography" in caption, or add information on color scale.

– removed "topography"

**Line 91** "High-Performance Computing" without capitals?

– implemented

**Line 94** Wind directions without capitals

– implemented

**Line 94** Dominant wind direction offshore is southwest for both 100 and 500 m.

– clarified

**Line 94** Using "rotating" is a bit strange here: the wind direction changes or the wind turns.

– implemented

Line 96 Start sentence with "The..."

   – implemented

Line 97 "Offshore conditions .." is a bit vague. How does the 10 degrees relate to the 5 degrees of the preceding sentence? If the wind would always veer 10 degrees than so would the average wind direction.

   – These 10 degrees relate specifically to wind direction above 500m as stated in the sentence. Clarified

Line 98 "the same westerly wind direction at high altitude" - what do you mean? Figure 2 shows that offshore the wind is predominantly southwest.

   – As mentioned in the sentence: "at high altitudes.". Clarified

Line 99 Add point at end of sentence

   – implemented

Line 99 Replace "The relative wind speed increase of" by "The wind shear at", and remove "and the already high wind speeds at lower heights"

   – implemented

Line 102 replace "distribution" by "distributions at each individual height level"

   – implemented

Line 102 replace "These statistics give an insight into the overall wind conditions, but the actual profile shapes" by "These distributions give insight into the wind speed statistics at the individual heights, but not onto the statistics of the wind profile shapes."

   – implemented

Line 105 105: replace "have a fairly narrow range" by "are relatively low"

   – added "are relatively low" and kept "narrow range" as this necessitates the nonlinear color map

Line 106 replace "up to high altitudes" by "for the full height range".

   – implemented

Line 106 replace "This leads to the development of" by "The distributions show"

   – implemented

Line 109 "Such multimodal distributions at higher altitudes are better described by the sum of two or more probability distributions" - isn't this the definition of multimodal?

   – According to Wikipedia: In statistics, a bimodal distribution is a probability distribution with two different modes (The mode is the value that appears most often in a set of data values.) (...).

**Line 113** "As mentioned above, the relative wind speed increase with height is less pronounced offshore than onshore." - why mention it twice?

- removed

**Line 114** suggestion "Conventional WTs benefit from low wind shear offshore .."

- Is the benefit that the shear is lower or the speeds are higher? kept as is

**Line 115** replace "However, offshore AWES will also benefit from higher offshore winds and move offshore for other reasons such as safety or land use regulations" by "Nevertheless, also AWES benefit from low wind shear. Among the reasons for placing AWES offshore are safety and land use regulations."

- implemented

**Figure 2** State that these results reflect single locations. Leave out "On average wind direction onshore rotates about 14 (symbol missing) while offshore winds rotate (symbol missing) about 5(symbol missing)between 100 and 500 m. Onshore shows a higher wind shear (symbol missing) due to higher surface roughness and relatively high wind speeds offshore." - this belongs in the text.

- implemented

**Line 116** "Another benefit of offshore AWES in comparison to conventional WT is the smaller and cheaper support structure." - Statement is a bit misplaced in the wind resource discussion and needs a reference. Also the arguments are one-sided, there are many reasons why not to put AWES offshore.

- moved up to previous sentence together with " safety or land use regulations". I do not have a specific reference for this, but this argument (replacing tower with tether) has been made before.

**Line 118** replace "categorized" by "characterized"

- implemented

**Figure 3** Clarify that this is not a 2D histogram, but independent histograms/distributions per height. State that these results reflect single locations.

- implemented

**Line 126** Not clear if this is the classification as presented in the table.

- clarified

**Line 129** Is the ocean always warmer? Remove "likely"

- I'd say generally warmer, yes. I would keep likely, as I did not specifically compare air and water temperature at the investigated location.

**Line 131** Not sure what the use is of this paragraph: only covers literature without touching upon the results

– Point is to describe the wind conditions at the specific location and to justify the choice of location as well. Therefore, atmospheric stability and Obukhov length are introduced to the reader as they are later discussed in figure 8.

**2.4 Clustering of wind conditions**

Table 2 State that these results reflect single locations.

– implemented

Line 136 Statement is not very informative, rephrase to emphasize that it's not just time- and space-averaged wind velocities that effect the power output of wind energy systems, but also the variation in time and space.

– implemented

Line 138 Elaborate on excessive averaging

– rewwritten

Line 141 replace "proxy" by "metric", replace "a metric that" by ", which"

– implemented

Line 143 Only one study is listed

– implemented

Line 143 Not clear what diverge with height means: large wind speed spread at high altitudes?

– yes, within each OL classification, while low altitudes did not show that spread

Line 144 Do you mean to say that there is no correlation between the wind speed profile over a high elevated layer and the surface-based stability?

– No, there still is some correlation. I am saying it is not enough to use surface data to describe wind conditions at hundreds or thousands of meter altitude. Atmospheric stability close to the surface might be different from the one aloft.

Line 146 Also grouping based on stability is "based on data similarity": be more precise.

– added "their" similarity to clarify it is the similarity of "wind speed or velocity profiles"

Line 146 What does the reader need to see in the appendix? - Don't refer to results before the explanation of your methodology is finished.

– It is not a result. This figure visually shows the coherent / non-diverging clusters with all the comprising profiles. It is not a necessary result, but rather a visual addition. Moved to body of text

Line 148 Mathematical clustering is a vague term: explain. You use clustering, classifying, categorizing, and binning interchangeably: be more consistent.

– Point is that the clustering algorithm does not take physical conditions into account, but only relies on profile similarity. clarified

Line 154 The u-component is per definition along the x-direction. Explanation is not clear: also $u_{main}$ and $u_{deviation}$ are not throughout this paper anymore - so why introduce them here?

– Which definition? You can define u & v or x & y as you wish! Such as along lat and lon lines or something else. Why anymore? It is important because this way it is easier to compare trajectories. Otherwise the wind speed profiles would just go in any direction.

Line 159 Replace "defined" by "represented"

– implemented

Line 159 Suggestion: "The clustering finds the centroid positions that minimize ..."

– rewritten

Line 160 Distances between?

– "These centroids are arranged such that they minimize the sum of the Euclidean distances, to every data point within each cluster."

Line 161 Rephrase: the centroid will at best coincide with a data point by chance

– added

Line 161 Explain what a data point consists of. Also not clear that each data point is assigned to the closest centroid.

– added

Line 162 I misunderstood at first: I thought you were talking about the labels assigned to each data point. Leave out mentioning the initialization as it is confusing. Just mention that the label number that each cluster gets is rather random and does not have any mathematical meaning.

– implemented

Line 164 replace "Later evaluation uses clusters sorted by average wind speed up to 500 m." by "As presented, the resulting clusters sorted by average wind speed up to 500 m."

– implemented

Figure 4 Use the same color scheme for the clusters as for figure 5. It's unclear from the right column of figure 4 what is plotted on the y-axis as there is no grid/ticks.

– improved the clustering and implemented consistent coloring scheme

Line 168 "inertia reduction becomes marginally small with increasing number of clusters" - this is not completely true: above all the elbow method says that kinks in the inertia trend indicate sensible choices for k. It's hard to observe them with evaluating so little values of k. Best to just use a step size of 1

– That is one definition and approach.

**Line 169** that doesn't make inertia meaningless

– removed

**Line 170** add space after hyphen

– implemented

**Line 171** table 1 doesn't show any difference between the vertical levels (grid) of the two analyses. How different are they? I would not expect the vertical grid to have a large effect. More probable is a larger spread of the centroids.

– removed

**Line 172** replace membership by a more precise explanation

– added explanation

**Line 176** also if you would use a fixed (not random) initialization the cluster would not be arranged in a logical order, suggested: "Note that the order of the clusters is random and does not follow any logic."

– implemented

**Line 176** It's still on the y-axis.

– clarified

**Line 178** Too many I don't find very likely. At least the mean silhouette score always decreases with k. Do you have a reference?

– Added citations, also see: `https://www.mathworks.com/help/stats/silhouette.html` and `https://medium.com/analytics-vidhya/how-to-determine-the-optimal-k-for-k-means-708505d204eb`.

**Line 179** Not clear what you mean with "the continuous nature of wind which results in a high cluster proximity" and how this effects the silhouette scores - please expand.

– removed

**Line 180** For readability I would suggest not to jump to the conclusions of the next section.

– removed

**Line 181** Replace "intersect .." by "are grouped together with monotonic wind profiles"

– Clarified. I want to say that these non-monotonic profiles are closer to one cluster at low altitudes and then closer to another at higher altitude.

**Line 182** closing parenthesis missing

– added

**Line 184** Is there a reason for using first wind speed and after velocity? If not, rephrase.

– I only used wind speed profiles (magnitude over height) for visualization purposes in this paper. The actual code uses 2D wind velocity profiles ($u_{main}$ & $U_{deviation}$ component over height)

Line 185 replace "comprising .." by "WRF-simulated wind speed profiles that are input for the clustering"

  – clarified

Line 185 suggestion: "Within a cluster, the wind speed profiles span.."

  – implemented

Line 189 False statement: this explains the higher inertia, not the silhouette score. Also the onshore clusters are distinct.

  – removed these parts of the sentence

Line 191 I don't see why "directional differences" result in a decrease of wind speed with height - explain.

  – removed

Line 192 What about small-scale weather phenomena?

  – added

Line 192 Why too many clusters (in which context)? I think it is very insightful to find these type of cluster mean wind profiles. Them having a lower probability makes it even more interesting to evaluate why the clustering gives these clusters as a result as normally k-means clustering tends to produce equally sized clusters.

  – removed

Line 195 Is the wind speed inversed or the wind shear?

  – implemented

Line 197 replace "determining" by "dominant" and replace "stacked" by "ordered"

  – implemented

Line 200 Suggestion: add reference to Schelbergen

  – added

Line 204 For which application/analysis are long term averaged wind speed profiles considered? Using a averaged profile **shape** is commonly done for AEP calculations, but that's a somewhat different approach.

  – An error would occur if using a simple log-profile and low reference wind speed at low heights. Extrapolating this to higher altitudes might over-predict wind speed.

**Figure 5** The wrf-simulated wind profiles only show as a filled area, so not very informative - but I suppose it doesn't hurt either.

– It's hard to display $\sim 52.000$ profiles. Just supposed to give a feel for the range of profiles

**Line 220** The name of the "other" category suggest to me that these conditions are exceptional/not often occurring, however your analysis shows the opposite. How should I physically interpret this category?

– These clusters exhibit Obukhov lengths close to zero (likely caused by very low friction velocity $u_*$) and are classified as "other" because they do not fall within one of the other atmospheric stability classes according to Floors et al. (2011) (see table 2).

**Line 220** What do you mean with wind power assessment? Why is the impact of low wind conditions low? I don't understand the comparison.

– Removed. Meant that they do not contribute a lot to the power curve as they are just around cut-in wind speed.

**Line 221** Basically what you're saying is that how often wind conditions occur (the wind speed distribution) has a large effect on the AEP? - I would say that the reader is familiar with this.

– Removed.

**2.5 AWES trajectory optimization**

**Line 231** Very technical start of the section - readability would benefit from further introduction.

– added a short paragraph

**Line 232** Which unstable dynamics are you referring to?

– The tethered AWES dynamics are unstable.

**Line 233** Which multiple inputs, multiple outputs?

– input: tether & aircraft control inputs, control surface deflections etc; outputs: tether speed, tension, aircraft position, speed etc

**Line 238** Which simulated profiles? Aren't you using the cluster means?

– No, I am not using cluster means, but actual (5th, 50th, 95th percentile) wind velocity profiles

**Line 239** replaced "during" by "of"

– rewritten

**Line 247** What is the main wind direction?

– clarified. $u_{main}$ as defined by the average wind direction up to 500 m, compare section 3

**Line 252** Suggestion: "We use the aerodynamic model of the Ampyx AP2 (Malz et al., 2019; Ampyx)"

– implemented

**Line 252** The footnote is confusing as you mention that no other data is available while you are referring to a source.

– clarified

**Line 259** Introduce kappa and bring equation 2 forward.

– moved and rewritten.

**Line 260** "This results in an overestimation of output power and lower cut-in speed in comparison to a heavier aircraft." - unclear how you come to this conclusion.

– Aircraft mass delays cut-in and reduces average power.

– $\kappa$ was changed to 2.4.

**Line 263** Mentioning focus of paper is a bit misplaced, better save that for the outlook.

– This is more to justify the unrealistic mass scaling.

**Figure 9** Why mentioning Loyd in caption?

– Because that is where $c_L^3/c_D^2$ comes from

**Line 264** rephrase

– How and why?

**Line 265** Leave out or mention how you would estimate it.

– Included a reference calculation on AWES power and AEP estimation using Loyd's formula including tether drag.

**Equation 2** Position at related text and introduce properly.

– implemented

**Line 270** mention conditions for which this is valid.

– iplemented

**Line 270** Last part of sentence is confusing. Suggestion: ".. , thereby the reelout speed is expected to remain below 10 m/s as the wind speed hardly exceeds 20 m/s.

– implemented

**Line 271** Implies that the ratio is fixed, however I don't think that's what's meant. - example values 10/15 is superfluous.

– clarified

**Line 272** I would not discuss this pre-optimization step. As you mentioned yourself: this paper is about assessing performance metrics and not system design. Consider the tether diameter as a given.

– removed

Line 280  No need to state subsection number, table number suffices.

 – removed

Line 281  why "or"?

 – replaced by and

Line 282  add "coefficient" - why do you use a linear relation?

 – added and clarified

Line 285  replace "prevailing" by "can be adapted to", replace "dynamically" by "continuously"

 – than there would be 2 "adapt" in the same sentence. "continuously" implemented

Line 286  replace "AWES" by "AWESs"

 – implemented

Line 290  Isn't 19 m/s a bit low? - this is equivalent to a mild storm

 – 19 m/s at 10 m height would be quite extreme.

Line 293  Explain what a p-value is: add a figure to explain.

 – p-value stands for percentile. Profiles are chosen based on percentile of average wind speed distribution. Sentenced added, no figure added

Line 294  Why average wind speed up to 500 m?

 – Added an additional sentence. Previous analyzes showed that AWES mostly operate within this range

Line 294  On line 176 you mention that a representative k is 10. Better to choose k already at section 3, instead of quickly mentioning it here.

 – These are actually 2 different things. Line 176 is just showing silhouette score for k=10 (could also be 20,30, 1000). Added sentences in section 3.

Line 296  Rewrite "cluster centroids.. implemented"

 – not sure what you mean, but "to estimate AEP" was moved to the end of sentence

Line 297  Not clear when you use the p5,50,95 and when you don't and how you use them if you do. Please rewrite paragraph. So you only use these p-values for determining the power curve? I would say this is the most important part of this study so it should get more explanation

 – implemented

Equation 3  Move to the related text.

 – implemented

**Line 304** Would the analysis benefit from running the optimization using multiple starting points?

– Yes, ideally you want to compare multiple initialization (tether length, radius, elevation angle, etc.), but that is extremely costly. We initialized different tether length depending on system size and reference wind speed

**Line 306** remove hyphen: "inequality"

– implemented

**Line 306** Which equality constraints? You only mentioned inequality constraints.

– e.g. tether diameter, wind speed profile for each optimization

**Line 307** As far as I know, an optimal control problem always uses discretized control intervals, irrespective of if direct collocation is used or not.

– yes, but there are different methods and we use direct collocation

**Line 307** missing space after dot

– fixed

**Line 307** Is this the input you give to awebox for generating an initial guess? Explain estimated aircraft speed - this will vary between reel-in and reel-out and also within the reel-out phase over one loop.

– Aircraft speed is estimated from the circumference (2 pi radius) divided by estimated time of 1 loop. Clarified in text

**2.6   Results**

**Line 317** Shouldn't it be "average wind speeds for different height ranges"?

– I also compare average between height ranges to a single reference height. I therefore believe that my formulation includes these as well and is formulated more generally.

**Line 320** "Rayleigh distributed log-profiles" is not very precise

– What would you add? I understand that this is how wind conditions are defined in the IEC standards and in sub-section 'Wind boundary conditions'

**Line 323** "representative" does not fit the context, list which profiles you have chosen.

– The point here is that these are not specific, special trajectories. They are supposed to give the reader an impression of typical optimal trajectories based on representative onshore and offshore wind conditions that have been chosen as written in brackets: 'chosen because of different wind speeds and profile shape'. Removed brackets and clarified in text

**Line 325** missing closing parenthesis, replace "figures" by "figure"

– removed

Figure 10  In caption list which profiles are plotted. Also describe what is highlighted using the colors in the left column. Remove "The deviation of the colored lines is caused by the approximation of discrete data points with Lagrange polynomials." - too much detail for caption.

– Wind profiles chosen to represent typical low (blue, orange), medium (red) and high wind speeds (green).Removed

Line 328  replace "optimization" by "modelling"

– it's for optimization though because we need 2 times differntiable functions for the used solver

Line 328  Wind speed vs wind velocity profiles is a bit confusing. Just mention that only the wind speed is plotted and that changes in wind direction with height can be seen in the lower left plot.

– removed

Line 330  Didn't you introduce the "rotated u and v component" as $u_{main}$ and $u_{deviation}$? - then also use them here

– changed labels

Line 330  replace "wind velocity components as experienced by the AWES in color" by "with the part of the profiles corresponding to the height range swept by the kite in color

– changed

Line 332  not clear which are the onshore and which the offshore profiles.

– removed

Line 333  first should be x-z plane

– implemented

Line 335  perpendicular to x is both y and z-direction

– removed

Line 335  What is unrealistic about them?

– probably unrealistic than anyone will want to fly such crazy trajectories and will probably take a less optimal, but safer / easier to fly trajectory. removed

Line 338  Can you identify whether the system is de-powering or not? If so than you can differentiate between the two.

– I don't think that is possible. The system de-powers when it would exceed any constraint, e.g. tether force constraint

Line 338  What about de-powering by increasing the reelout speed?

– that is possible, but the optimizer does not chose to do so when looking at figure 10. Possible that that would increase the tether length too much and increase reel-in losses.

Line 342 So for an elevated path, you don't want to fly at maximum cl3/cd2? This is not what we observe in figure 10: the most elevated path has the lowest angles of attack during reelout

- But lowest $\alpha$ does not mean optimal $\alpha$. $\alpha$ here was (wrongfully) estimated without tether drag. Including tether drag optimal $\alpha$ would be higher. It can be seen that the aircraft reduces $\alpha$ to stay within force constraints and optimal $\alpha$ is higher as all other $\alpha$ have similar values during reel-out. Section removed as $c_L^2/c_D^2$ did not include tether drag and the $\alpha$ value was off.

Line 347 So you're relating this observation in the reel-in phase (reaching max reel-in speed) to an observation in the reel-out phase (de-powered loop)? Isn't it more likely that the de-powering is triggered by reaching max tether force during reel-out?

- I believe the phrase 'in these cases' was not clear. rewritten

Line 355 Which previous analyses?

- unpublished analyses done by the awebox developers

Line 356 Not completely: zero would probably not give you the best performance.

- agreed, hence 'almost' zero

Line 363 Clarify that you're not just using the 4 profiles that were treated in the previous subsection.

- see line 359: 'The data is based on the p5, p50, p95-th wind profiles of k=20 onshore and offshore clusters...'

Line 363 What is $z_{operation}$? I would expect a height range, not a single value.

- Not sure what you mean: 'frequency distribution of operating altitude $z_{\mathrm{operating}}$'

Line 364 Replace "emerge"

- with what? I want to keep the sentence in active language, don't want to write: become visible or can be seen.

Line 365 add comma after onshore, correct for this in the rest of the paper

- added

Line 365 remove "and typically higher winds offshore" from end of sentence and mention in a different sentence that generally tether length increases with wind speed

- Yes, tether length increases with wind speed, but comparing the the tether lengths for the same reference wind speed at both locations show different tether lengths with must be due to wind speed profile shape. Clarified

Line 369 "on the other hand" - comparison is not clear

- removed

Line 372 "This also has implications for tower-based.." - Leave the WT discussion out. It's well known that offshore turbines have lower towers than onshore.

– removed

Line 376 The distributions for 25 and 50m2 are not that much different and is worthwhile mentioning.

– That is also a finding that needs to be communicated.

Line 378 Figure 11 shows only onshore 1% exceeding 600 m for 20m2 and offshore 0%

– This sentence refers to $A_{wing} = 50m^2$. Clarified

Figure 11 Differentiate between p5, p50 and p95 using for example different markers. Put occurrences on the horizontal axis. If I understand correctly your only plotting the results of 60 optimizations: this would be more clear by plotting occurrences.

– These are the results of 100 time steps (added information in text) for each of the 60 optimizations. The percentile of each profile does not add any information, but would just be a distraction. These are are 60 representative wind conditions.

Line 384 replace "unanimously accepted" by "consensus"

– replaced

Line 385 replace "wind speed probability distribution" by "wind resource model"

– implemented

Line 387 I would just mention that AWESs generally operate in a larger height range

– implemented

Line 389 So this leaves you with a non-linear grid on the x-axis? Also the order is different for every reference height range? - If so, mention this. I would expect the p5 and p95 causing a lot irregularities in the power curves and wind distributions. Would it be possible to show data points represent p5, p50 and p95 powers? About using the power curves for integrating: basically you're trying to map this high dimensional space describing the wind profiles (60*2=120 dimensions?) to a 1D space. By doing so you loose a lot of information on the wind profiles. How would you justify integrating over only this 1D space? How accurate would this be? Do you also use p5, p50 and p95 wind profiles to the frequency distributions? Is the area underneath your frequency distributions 100% in that case?

– Recreated this figure with new data . The point of using p5, p50 and p95 within each cluster is to better represent the entire spectrum of simulated wind conditions and not to isolate their effect. No, this does not result in a non-linear grid on the x-axis. The x-axis is linear, but the points are scattered at a different location on the abscissa. The purpose of this analysis is to derive a SIMPLE power curve (+ investigating a good reference height x-axis) that can easily be understood and communicated. Therefore, it it necessary to simplify the wind profiles to a single value. Yes, the frequency of wind speeds adds up to 100%.

Line 393 "respective AWES operating altitude" - why using 60 - 629 m and 59 -551 m? Not all trajectories fly over this full range.

– That's right. Added: The height ranges displayed in the legend represent the minimum and maximum operating altitude.

Line 394 "black" instead of "red"

– implemented

Line 395 'The power curve shows a varying $c_p^{WT}$

– added 'up to rated wind speed'

Line 397 what is the value of the rotor disc area?

– added

Equation 4 $U^3$

– implemented

Line 400 What do you mean with limit specific designs?

– removed

Line 405 Isn't this just due to your choice for the rotor disc area?

– The point here is to show that reference wind speed shifts the power curve and AWES power curves don't always align with typical WT power curves.

Line 406 Explain more precisely: if wind speed is monotonically increasing with height $u_{100m} \leq u_{100-400m}$. So the data points move to the when plotted against $u_{100-400m}$ with respect to when they are plotted against $u_{100m}$.

– Implemented

Line 409 replace "overlap" by "lie on top"

– implemented

Line 409 Not clear why 200m ref would be better, since you mention in the previous sentence that every reference height gives roughly the same power curve.

– removed

Line 410 with divergence you mean large difference?

– I mean that the power curves diverge from each other with increasing reference wind speed.

Line 411 Also $c_p^{WT}$ is lowered after the rated wind speed

– sentence removed

Figure 12 On y-axis is the energy contribution: not E average

– changed to E. Also change legend of operating height

**Line 414** That the flight trajectory influence mean cycle power should be clear from the literature study.

– That is true, but these results nonetheless highlight this fact.

**Line 415** An overkill to use a separate plot when only adjusting the cp. What if you change the rotor disc area?

– Which is why the plot is only in the appendix. Added a comment, assuming the same rotor diameter.

**Line 416** Leave out "A better .. future study.". Mention in the outlook at the end of paper.

– Left in "A better...", but removed "This however ..."

**Line 418** Rephrase "integral multiplication"

– removed "integral"

**Line 419** replace "Its total" by "The area underneath the distribution "

– implemented

**Line 420** Rewrite "shifts towards higher wind speeds due to .. higher wind speeds"

– rewritten

**Line 421** How is that similar?

– removed

**Line 426** Which effect scales with size? - Isn't this specific to your methodology, i.e. the integration of the equivalent 1D wind profile space.

– removed. I don't expect that this is because my method, but rather due to the more realistic wind conditions which result in a wider range of profiles

**Line 428** Leave out "This indicates that onshore wind conditions favor higher operating altitude due to higher wind shear.

– removed

**Line 429** What reduction? The AEP goes up

– it's the energy relative to system size.

**Line 432** "This main difference" - which difference is explained exactly?

– removed

**Line 433** The cp is very much dependent on the area you chose, which you haven't justified. Also, the onshore 20m2 curve for reference height 100m lies on top of that of the wind turbine.

– removed

**Line 434** "wind speed along the actual AWES trajectory": this is a varying property - how would you use it as a reference?

    – added 'average'

**Line 436** Figure 11 shows that the operational height range of 100 m are common. Why would a 500m height range be a good reference in that case?

    – It is the average wind speed between 100 and 600m not a single height.

**Line 439** How would you use this power coefficient in practice for power estimation? I think it's still a rather challenging task as you still need to determine the relation between lpath and the wind speed. More importantly, for WTs cp is often used to quantify the efficiency of the energy conversion. I don't think the given formulation is a good metric for AWES efficiency. Take for example the (hypothetical) situation where we have a uniform wind field and let's neglect the variation of the air density. As you implied earlier (line 356): the mean cycle power is not very sensitive to the flight path length. So for the same kite size, a shorter and longer pumping cycle with the same mean cycle power will give me completely different values for cp while the AWES efficiency did not change.

    – Agreed, removed sub-section

**Line 441** You mean average wind speed?

    – removed

**Line 442** Air density at which height?

    – removed

**Line 444** $z_{operation}$ suggests a single height, but probably needs to be a height range? Change throughout the paper.

    – removed

**Line 451** cp is not truly "velocity profile independent": especially at low wind speeds you see a vertical spread for fixed wind speed. How sensitive is power output to the wind profile shape?

    – removed

**Line 452** sentence does not explain any difference. Rewrite "The difference .. remains almost constant (see sub-section 5.1 and 5.2)" - unclear what you're saying.

    – removed

**Line 457** Dividing cp by c is the same as having the area of the kite multiplied by the path length in the denominator of equation 5 - why not use the wing area instead of the swept area in the first place and leave out the path length?

    – removed

**Line 459** Fit looks poor for larger wind speeds. Why do you only plot the line up to 18 m/s?

– removed

**Line 462** "contrasts" replace by "compares"

– implemented

**Figure 13** The markers are not connected by the lines. Why is this?

– because these are individual data points and connecting them would result in zig zag lines. Instead, the lines are the curve fit. Figure removed

**Equation 7** leave out h/year

– removed

**Line 472** remove "static"

– removed

**Line 473** use "probability distributions"

– but it is the cluster frequency and not the probability distribution derived from a model

**Line 475** replace "justified" by "justify"

– implemented

**Line 480** I think it is a bit bold to say that they are averaged out. You could reason that they are underrepresented. Also show with a figure similar to 12 how both computations are different

– I think my point here is that high wind speeds contribute more to power than low speeds because of the cubic relation between power and wind speed. If you instead just use the average wind speed you lose this non-linear relationship.

**2.7 Conclusions and outlook**

**Line 490** replace "characterized" by "evaluated"

– implemented

**Line 492** replace "deduced" by "uses"

– rewritten

**Line 493** replace "Representative wind velocity profiles based on k-means clustering were chosen to reduce computational cost." by "A representative wind resource model is deduced using the results of clustering wind profiles" - and place after discussing WRF

– implemented

**Line 494** Do you need this high resolution if you drastically simplify the wind model in the end?

    – I still used actual profiles rather than cluster-means. I guess it depends on the purpose of the analysis. For the purpose of analysing trajectories and considering the non-linear relation between wind speed and power, I would argue yes.

**Line 497** "acceleration"?

    – 'increase'

**Line 511** replace "implemented" by "given as input"

    – implemented

**Line 514** replace "fast" by "high"

    – implemented

**Line 517** See earlier comments on WT power coefficient

    – removed

**Line 519** remove part about social acceptance: not touched upon in body of paper

    – True, but kept in to motivate moving offshore for other reasons than beneficial wind conditions

**Line 522** replace "collapse .." by "yields a location and size independent metric"

    – removed

**Line 526** You did not evaluate the AEP for 10 clusters and the p5, p50, p95 profiles if I'm not mistaken. So how do you get to this conclusion? If the AEP is higher than for all other calculations I wouldn't expect it to be the most realistic.

    – rewritten

**Line 529** I don't see this back in Figure 14, it shows quite a large bias between the log and cluster AEP.

    – rewritten

**Line 536** Why do you expect this?

    – removed

**Line 538** Be more general in this paragraph

    – implemented

**References**

[1] Moritz Diehl. Airborne wind energy: Basic concepts and physical foundations. In Uwe Ahrens, Moritz Diehl, and Roland Schmehl, editors, *Airborne Wind Energy*, pages 3–22. Springer Berlin Heidelberg, Berlin, Heidelberg, 2013.

---

## Referee Report (RR1)

**Offshore and onshore ground-generation airborne wind energy power curve characterization - Referee Comment**

Mark Schelbergen[1]

[1]Faculty of Aerospace Engineering, Delft University of Technology, Kluyverweg 1, 2629 HS Delft, The Netherlands

**Correspondence:** Mark Schelbergen (m.schelbergen@tudelft.nl)

**1   General comments**

The first part of the paper up to section 3 was a pleasure to read: clear structure and good language. The goals of these sections are clear and it was clear how these contribute to the bigger picture: summarizing the WRF results (section 3) and obtaining a wind resource model from it (section 4). On the other hand, from section 5 it's not clear how the presented material will be used. The reader can be helped a lot if the story line/different analyses are introduced early in the paper, preferably using a diagram. Only while reading section 5 it becomes more clear how the material in section 4 is used. Still a lot of details are missing for the reader to be able to reproduce the presented results. I find it unsatisfying that there is no justification/discussion on the degree of simplifications needed to map the high-fidelity model output to wind statistics and power curve as single argument functions (height-range-averaged wind speed). I would expect that the accuracy benefits of the relatively high fidelity (and computational costly) models in the first computational steps, cancel out when the author simplifies them before calculating AEP. Why not use a more detailed numerical integration that regards the detailed information of the clustering output/power optimizations.

I understand that the goal is to get a simple characterization of the power output/AEP similar to that of a WT. However, how much is this worth when you loose precious details in the process given that you went through all the effort of setting up the suggested high-fidelity tool chain?

Writing: The paper still contains numerous typos. It could benefit from a better structure and more precise writing. Citing and referring to figures is done inconsistently and not in line with WES standards. Also referring to figures/sections that are presented later on in the paper is distracting. Minimize inconclusive language, e.g.: "which could indicate that these are more realistic estimates" (p27, l493), however more results might be required to give definite statements.

We have different understanding of how to use "However" (pointing out a contradiction) and "Non-trivial" (not unimportant).

Figures, tables, and equations: Figures are placed in the appendix, while they are covered in the body of the paper.

**2    Specific comments**

Section 2 misses a discussion on why WRF data is used. It is clear that WRF can provide an accurate basis for getting to a wind resource model. However, this study only considers 1 year of simulation data, which is far too little for giving a good representation of the wind climate and thus for AEP calculations. Typically 30 years of data is needed for this purpose. The

5   need for correcting WRF using lidar is stated in the introduction. However, it is not clear if this is done for this study. Also it is not justified why going through all the effot of doing WRF simulations should be an integral part of the methodology. There are a few wind atlases available that cover a wide area (including the investigated sites) for a much longer period, which might be a better source for the wind data? In conclusion, what would the added value be compared to alternative approaches such as that of Malz and Schelbergen?

10     Section 4 could gain a lot of clarity by more clear structuring. First introducing the system models. Therefore a clear distinction between the dynamic model and OCP should be made. At the moment they are presented as one. The QSS model is a very simplistic. I would suggest also including the reel-in in the model, similar to what is done by Luchsinger. The current model does not give a good estimation for a pumping mode system. It's not clear why the upscaling model is needed. I don't see results for multiple system sizes in the section 5. I only found some plots in the appendices. The contribution to the paper

15   of including a scaled up system is unclear in the current format. Split up 4.3 in the dynamic model and OCP and add 4.5, 4.6, and 4.8 to the latter as these are part of the OCP formulation/solving it. Split section 4.7 up in the wind profile model and wind resources model. Add the polynomial description of the wind profiles. After reading this section, I did not have a good understanding of how the p5/p50/p95 cluster profiles are used. I would argue that this is the most important part of the paper and therefore the approach taken there should be presented unambiguously. E.g., I'm puzzled by the last sentences - do you

20   sum up 10 min energy production of every WRF data point, or do you integrate the product of power curve and probability function? Also you "interpolate within each cluster linearly between p5,p50 and p95" - do you do this to get the power output of every 10 min data point? Why do you use 3 points? Isn't 3 points very little? It would help me if you would try visualize this process, or at least add some equations. Finish the section with an overview of what analyses are performed to get to the results in section 5 in e.g. a table. Clearly state which combination of system/wind models are analyzed.

25     Section 5.1 and 5.2 nicely introduce the OCP results. However it is not clear how the results of the remaining sections are obtained. This could be improved, by improving section 4. Don't mix methodology and results too much, e.g. equation 10 would nicely fit in section 4. Also, it would also help to take the reader a bit more by the hand when introducing the results. I had to go back and forth between section 4.7 and 5.3 to get a taste of what's been done, however I couldn't figure it out completely. Some open questions:

30   – which profiles are presented in fig 11?

   – which of the data points represent p5/50/95 profiles in fig 12?

   – how do you get to the dashed curve in fig 13a/b? a curve fit of some kind? - describe how

   – how do you come up with the relation between Uref(z operating) and U(z W=100 m)? - provide it in a plot

- how exactly is the log profile used? - I suppose you also use it in the OCP?

- how do you get the input for constructing the QSS curve?

- how do you calculate the AEP distribution exactly?

If I understand correctly, you did is doing some kind of curve fit to the markers in fig 13a to get an expression as a function of a height-range-averaged wind speed. What kind of curve fit do you use? How sensitive is the AEP to the type of curve fit? By doing the fitting, you loose precious information about how the power output relates to the wind profile that you acquired with the computational costly optimizations. Also by expressing the probability function of the wind resource as a single argument function (solely a function of height-range-averaged wind speed) you loose much information that was acquired with a lot of computation effort by the WRF and clustering. I think a more detailed numerical integration would be in place here instead of going through all this effort of detailed system/wind modelling to end up with simplistic models for power output and wind statistics. However, I might have misinterpreted the methodology used here (see comments section 4). Also, since the relation between Uref(z operating) and U(z W=100 m) is non-linear, you'll get a differently shaped power curve/probability distribution and thus AEP, depending on which of the two properties you use to express these functions. This tells me that this approach is mathematically unsound. I would say more proof is needed to justify your methodology.

Does the approach suggested in eq 10 act as a benchmark for the other AEP calculations? W.r.t. which property do you integrate? If I understand correctly, it comes down to sampling 3 points per cluster (p5/p50/p95) to quantify the power output within this cluster. Why just 3? I would suggest you use a sampling technique that picks a higher number of samples. Only 3 does not give a trustworthy benchmark.

---

## Referee Report (RR2)

**Offshore and onshore ground-generation airborne wind energy power curve characterization - Referee Comment**

Mark Schelbergen[1]

[1]Faculty of Aerospace Engineering, Delft University of Technology, Kluyverweg 1, 2629 HS Delft, The Netherlands

**Correspondence:** Mark Schelbergen (m.schelbergen@tudelft.nl)

The manuscript attempts to capture the detailed WRF results with a single curve describing the probability of the wind conditions at variable operational height and the associated power curve of an airborne wind energy system. The difference in estimated AEP ( 10%) depending on the choice of the curve's variable suggests that the methodology fails to adequately describe the WRF profiles and power output with a single curve. The work lacks validation of the calculated AEP with a detailed numerical integration. The presented power curve expressions are ambiguous, rely on site-specific training, and in my opinion conflict with the author's 'easily understandable' requirement. Lastly, the work would benefit from more focus to present the research output.

Please find my response to the author's response below with the bullets in blue indicating the more urgent topics.

[RR] - Referee report

[AR] - Author response

[R2A] - Response to author by referee

[RR] I find it unsatisfying that there is no justification/discussion on the degree of simplifications needed to map the high-fidelity model output to wind statistics and power curve as single argument functions (height-range-averaged wind speed).

[AR] The same simplification is done for every application of the Weibull distribution or simple power curve derivation for conventional wind turbines. They also simplify complex wind conditions to a simple distribution and ignore the variation of wind speed along the rotor diameter.

[R2A] Can you cite any work on conventional WTs that use mesoscale simulations to find a Weibull distribution to finally arrive at the AEP calculation?

[RR] I would expect that the accuracy benefits of the relatively high fidelity (and computational costly) models in the first computational steps, cancel out when the author simplifies them before calculating AEP.

[AR] See previous answer. These simplifications are done every day when applying wind statistics. They have been made to generate easily understandable, comparable to conventional power curves and AEP estimates.

[R2A] I understand your motivation. What I question is the justification/validation. I believe more proof is needed to increase confidence in the results.

[RR] I understand that the goal is to get a simple characterization of the power output/AEP similar to that of a WT. However, how much is this worth when you loose precious details in the process given that you went through all the effort of setting up the suggested high-fidelity tool chain?

[AR] – This is a different application of the data. I do not think that I lost these details. We can still investigate the time series data or statistics if we want to, what we did in Sub-section 5.1 and 5.2. Other analysis are possible, but beyond the scope of this paper. – The description of power curve and AEP are supposed to be simple to be similar to that of conventional WT.

[R2A] - I meant details on the connection between the wind profiles and AEP. E.g.: how much of the AEP is attributed to certain wind conditions? This information is lost when casting the WRF results in the probability distributions. - That would be convenient. However, I would argue that more importantly they should provide a reasonable AEP prediction.

[RR] Section 2 misses a discussion on why WRF data is used. This study only considers 1 year of simulation data, which is far too little for giving a good representation of the wind climate and thus for AEP calculations. Typically 30 years of data is needed for this purpose

[AR] – Added comment in introduction and AEP section. – The investigation only make statements regarding the investigated years.

[R2A] You mention only 1-year of data is used "to simply the analyses" - what is more simple about it? Would you also recommend others to use the same approach? Wouldn't using wind atlas data simplify the analysis more and at the same time allow you to use more years of data?

[RR] There are a few wind atlases available that cover a wide area (including the investigated sites) for a much longer period, which might be a better source for the wind data?

[AR] As mentioned in the paper, ideally we could use long-term measurements, but this is not possible. Therefore, WRF provides a compromise between computational cost, spatial and temporal resolution and accuracy of the wind data. WRF and wind atlases do have their justification and application!

[R2A] So what is the trade-off between WRF and wind atlases? Why did you opt for WRF?

[RR] What would the added value be compared to alternative approaches such as that of Malz and Schelbergen?

[AR] - I don't understand the question. We are using different approaches, models and model fidelity, locations and analyses. I believe there is enough justification for both our research. - I am using higher resolution wind data and a dynamic optimization model to estimate AWES performance.

[R2A] - I'm not saying that there isn't. But make it explicit in the Introduction. With this you can make your key contribution mentioned in the introduction more precise. - I don't find in the text why these are need to answer your main research question/hypothesis. What is you main research question?

[RR] The QSS model is a very simplistic. I would suggest also including the reel-in in the model, similar to what is done by Luchsinger. The current model does not give a good estimation for a pumping mode system.

[AR] – Investigating the performance using a detailed QSS model is beyond the scope of this paper. The focus is the investigation of AWES performance using the awebox dynamic optimization model. The QSS model merely contextualizes the results. – Which specific model are you referring to? I am only aware of Pumping Cycle Kite Power from Luchsinger and this publication does not include a reel-in model for pumping mode systems

[R2A] - The quasi-steady state calculation you're referring to is not made for estimating the mean power of a cycle, solely instantaneous power. So the results can not be fairly compared with cycle results and thus do not help contextualizing the results. - It does, see sec 3.2.

[RR] After reading this section, I did not have a good understanding of how the p5/p50/p95 cluster profiles are used. I would argue that this is the most important part of the paper and therefore the approach taken there should be presented unambiguously.

[AR] Implemented a clarification in sub-section Wind profile model

[R2A] "From these sorted wind profiles, the 5th, 50th and 95th percentile profile are chosen and assumed to be representative of the spectrum of wind conditions within this cluster" - The clarification is still missing important details: Why 3 profiles? Representative how? Provide more arguments.

[RR] I'm puzzled by the last sentences - do you sum up 10 min energy production of every WRF data point, or do you integrate the product of power curve and probability 20 function? Also you "interpolate within each cluster linearly between p5,p50 and p95" - do you do this to get the power output of every 10 min data point? Why do you use 3 points? Isn't 3 points very little?

[AR] – rewrote sub-section and added additional explanation to AEP sub-section – It's 3 points per cluster and therefore 20x3 = 60 data points which are used to derive the power curve and estimate AEP!

[R2A] - Still unclear in Sec. 6.3. Why use this approach against e.g. using 60 clusters and only use the cluster means? Why using linear interpolation?

[RR] which profiles are presented in fig 11?

[AR] These profiles were solely chosen to visualize the range of wind conditions and the resulting trajectories. Giving the specific cluster number and p-value is meaningless.

[R2A] It would give a better understanding of how it fits in the methodology. Illustrative examples help understanding the methodology.

[RR] Which of the data points represent p5/50/95 profiles in fig 12?

[AR] Figure 12 shows operating heights for 3(3 profiles within each cluster)x20(number of clusters)=60 profiles . Therefore, it is not possible to indicate the percentile of each profile. p5 of a high wind speed cluster might be close to p95 of a low wind speed cluster.

[R2A] You could distinguish with different markers. Same for the power points.

[RR] how do you get to the dashed curve in fig 13a/b? a curve fit of some kind? - describe how

[AR] Yes, least square fit, interpolated data points. Added clarification.

[R2A] The clarification is still incomplete. What fitting function do you use?

[RR] By doing the fitting, you loose precious information about how the power output relates to the wind profile that you acquired with the computational costly optimizations.

[AR] I guess that is true to some degree, but this is done to be comparable to conventional WT power curve / AEP derivation.

[R2A] It depends on the size of the error that is introduced if this argument justifies the error, i.e., it is important to quantify what the penalty is.

[RR] Also by expressing the probability function of the wind resource as a single argument function (solely a function of height-range-averaged wind speed) you loose much information that was acquired with a lot of computation effort by the WRF and clustering. I think a more detailed numerical integration would be in place here instead of going through all this effort of detailed system/wind modelling to end up with simplistic models for power output and wind statistics. However, I might have misinterpreted the methodology used here (see comments section 4)

[AR] I understand your point, but we decided to use an already established, easily understood method to describe the wind resource as an annual wind speed probability distributions.

[R2A] It's common practice in preliminary AEP calculations for WTs though it relies on quite big assumptions. Also I would say your method is quite different; in particular how you derive the probability functions. How do you come to the probability distributions? Do you use curve fitting again? Is the area underneath even 1?

[RR] Also, since the relation between Uref(z operating) and U(z W=100 m) is non-linear, you'll get a differently shaped power curve/probability distribution and thus AEP, depending on which of the two properties you use to express these functions. This tells me that this approach is mathematically unsound. I would say more proof is needed to justify your methodology.

[AR] – Yes, this is correct. Using different reference heights does lead to differently shaped power curves with different probability distributions and AEP estimates (Figure 15 shows AEP for different reference heights). The point of this investigation is to show that the choice of reference height is important and needs to be investigated – Added clarification in the introduction section – In the conclusion it is mentioned that the choice of reference height is significant, particularly onshore

[R2A] Why would the AEP be different when all come from the same WRF/cluster results? I can understand that the curves appear to be different depending on the quantity on the x-axis. However, the discrepancies in AEP suggest that the methodology is faulty.

[RR] Does the approach suggested in eq 10 act as a benchmark for the other AEP calculations? W.r.t. which property do you integrate? If I understand correctly, it comes down to sampling 3 points per cluster (p5/p50/p95) to quantify the power output within this cluster. Why just 3? I would suggest you use a sampling technique that picks a higher number of samples. Only 3 does not give a trustworthy benchmark.

[AR] – I assume you mean interpolate?! ([R2A] yes) The profiles are sorted by wind speed up to 500 m, which is used as an a priori proxy for operating height. – Added a sentence to AEP section. – Using more profiles per cluster would probably improve the accuracy and trustworthiness of this approach. Ideally one would perform a trajectory optimization for every profile, but this is not feasible and comes at a very high computational cost.

[R2A] Malz did such a computational costly analysis, which shows that evaluating more profiles per cluster should be feasible.

---

## Referee Report (RR3)

[revised manuscript text omitted]

50    and Houle (2013) used exponential wind speed profiles with a wind shear exponent of 0.15 and a standard Rayleigh distribution with $7 \text{ ms}^{-1}$ to estimate performance and analyze cost. Leuthold et al. (2018) investigated the power-optimal trajectories and performance of a ground-generation multiple-kite layout over a range of logarithmic wind speed profiles, with different reference wind speeds. Licitra et al. (2019) estimated the performance and power curve of a ground-generation, fixed-wing AWES by generating power-optimal trajectories and validating them against Ampyx AP2 data (Licitra, 2018; Malz et al., 2019;

55    Ampyx, 2020), which is also used in this research. The optimal, single-loop trajectory was defined by a simple power law approximation of the wind speed profile. Because of the the up-scaling drawbacks of single-kite AWES, De Schutter et al. (2019) analyzed the performance of utility-scale, stacked multi-kite systems, using the same optimization framework as this research. Two logarithmic wind speed profiles, one onshore and one offshore, provided boundary conditions for the non-linear optimization problem. Malz et al. (2020b) ptimized performance, based on the model described in (Malz et al., 2019), for clustered

wind speed profiles, similar to this research. To reduce computation time, wind data were clustered into characteristic profile shapes and sorted by average wind speed. This allowed for the initial guesses of every subsequent optimization to be based on the previous results. Aull et al. (2020) explored the design and sizing of fly-gen rigid wing system based on a steady-state model with simple aerodynamic and mass-scaling approximations. The wind resource was described by an exponential wind shear model with Weibull distribution. Bechtle et al. (2019) used ERA5 data to assess the wind resource at higher altitudes for entire Europe. The authors describe the potential energy yield without accounting for a specific power conversion mechanism . The investigation includes a description of wind speed and probability for several heights. Schelbergen et al. (2020) compares energy production based on this data set to performance based on the Dutch Offshore Wind Atlas (DOWA) and light detection and ranging (LiDAR) data. The authors used principal component analysis and $k$-means clustering to determine representative wind speed profiles for a part of the Netherlands and the North Sea. They derived power curves and estimated AEP from wind statistics for several locations. Faggiani and Schmehl (2018) investigated aspects of joint operation, such as spacial stacking of the systems and phase-shifted operation of several 100 kW soft wing pumping kite systems arranged in a wind park. Performance was estimated by a quasi steady-state model (QSM) (Schmehl et al., 2013; van der Vlugt et al., 2019), similar to the one used for in this research, subject to a standard logarithmic wind profile.

[revised manuscript text omitted]

---

## Editor Decision (ED2)

**Impact of offshore and onshore wind profiles on ground generation airborne wind energy system performance**

Markus Sommerfeld[1], Martin Dörenkämper[2], Jochem De Schutter[3], and Curran Crawford[1]

[1]Institute for Integrated Energy Systems, University of Victoria, British Columbia, Canada
[2]Fraunhofer Institute for Wind Energy Systems (IWES), Oldenburg, Germany
[3]Systems Control and Optimization Laboratory IMTEK, University of Freiburg, Germany

**Correspondence:** Markus Sommerfeld (msommerf@uvic.ca)

**Abstract.** Airborne wind energy systems (AWESs) aim to operate at altitudes above conventional wind turbines (WTs) and harvest energy from stronger winds aloft. This study investigates these claims by determining dynamic, power-optimal flight trajectories, operating heights subject to realistic offshore and onshore wind conditions. The utilized wind speed profiles are based on simulated offshore and onshore 10-minute mesoscale wind conditions which are analyzed and categorized using $k$-means clustering. To reduce computational cost of the trajectory optimization, representative wind speed profiles from each cluster are implemented into the awebox optimal control model to determine feasible, power-optimal trajectories. The results describe the influence of wind speed magnitude and profile shape on optimal trajectories, tether speed, tether length and tension. Optimal operating heights are generally below 400 m with most AWES operating at around 200 m. This study compares power curve visualizations for a constant reference height of 100 m to an apriori operating altitude guess of 100 - 400 m to the pattern trajectory height. Power curves are estimated based on average cycle power and compared to wind turbine (WT) and quasi-steady-state AWES reference model (QSM) performance. A power curve comparison between mesoscale-simulated wind conditions and logarithmic wind speed profiles shows that the offshore location is reasonably well approximated by logarithmic wind speed profiles. Realistic wind data onshore often outperform the logarithmic reference due to the higher number of non-monotonic wind speed profiles.

**1 Introduction**

Airborne wind energy systems (AWESs) aspire to harvest stronger and less turbulent winds at mid-altitude, here defined as heights above 100 m and below 1000 m, presumably beyond what is achievable with conventional wind turbines (WTs). The prospects of higher energy yield combined with reduced capital cost motivate the development of this novel class of renewable energy technology (Lunney et al., 2017; Fagiano and Milanese, 2012). Unlike conventional WTs, which over the last decades have converged to a single concept with three blades and a conical tower, several different concepts and designs are still under investigation by numerous companies and research institutes (Cherubini et al., 2015). These kite-inspired systems consist of three main components: one or more flying wings or kites, one or more ground stations and one or more tethers to connect them. This study focuses on the two-phase, ground-generation concept, also referred to as pumping mode. During the reel-out phase the wing pulls a tether from a drum on the ground which is connected to a generator, thereby producing electricity. This

is followed by the reel-in phase during which the kite adjusts its angle of attack to reduce the aerodynamic forces and returns to its initial position. Various other concepts such as fly-gen, aerostat or rotary lift are not considered in this study (Cherubini et al., 2015).

Since this technology is still at a relatively early stage of development, validation and comparison of results is difficult. A standardized power curve definition and reference design, similar to Jonkman et al. (2009) or Gaertner et al. (2020), will enable comparison between different concepts and to conventional wind turbines. It is not the goal of this study to determine such a general power curve, but rather investigate the variation in power of a specific type of machine stemming from realistic wind profiles. Recent consensus among the community defined a power curve as the maximum average electrical cycle power as a function of the wind speed at pattern height, which is the time-averaged height during the reel-out power-producing phase (Airborne Wind Europe, 2021). Together with the site-specific wind resource, power curves can be used by wind park planners and manufacturers to estimate the annual energy production (AEP), levelized cost of electricity (LCOE) and determine financial viability (Malz et al., 2020a). The glossary does not define an estimation method for these metrics. In contrast to conventional WTs, where the wind speed probability distribution at hub height is used to determine AEP, AWES continuously change their operating height, making it difficult to determine the AEP with this approach. Furthermore, the performance of AWESs is highly dependent on the shape and magnitude of the wind speed profile over a range height from which the wind energy is harvested. Simple wind profile approximations using logarithmic or exponential wind speed profiles, which are often erroneously applied beyond earth's surface layer (Optis et al., 2016), might approximate long-term average conditions, but cannot capture the broad variation of profile shapes that exist on short timescales (Emeis, 2013). They are therefore an inappropriate approximation of instantaneous wind conditions and do neither capture diurnal nor seasonal changes, which decreases the quality of the predicted power output. However, they can be employed to estimate average performance and are the standard in most AWES power estimation studies. Heilmann and Houle (2013) used exponential wind speed profiles with a wind shear exponent of 0.15 and a standard Rayleigh distribution with 7 $\mathrm{ms}^{-1}$ to estimate performance and cost. Enneberg et al. (2018) describes the performance of a soft kite pumping mode AWES with a family of power curves at different fixed altitudes, which correspond to the findings in this research. Leuthold et al. (2018) investigated power-optimal trajectories and performance of a ground generation multi kite configuration for a range of logarithmic wind speed profiles. Licitra et al. (2019) estimated the performance and power curve of a ground generation, fixed-wing AWES by generating power-optimal trajectories and validating them against Ampyx AP2 Power data (Licitra, 2018; Malz et al., 2019; Ampyx, 2020), which is also used in this research. The optimal, single-loop trajectory was defined by a simple power law approximation of the wind speed profile. Because of the the upscaling drawbacks of single-kite AWES, De Schutter et al. (2019) analyzed the performance of utility scale, stacked multi-kite systems, using the same optimization framework as the present research. Onshore and offshore logarithmic wind speed profiles served as boundary conditions for the non-linear optimization problem. Malz et al. (2020b) maximized performance, based on the model described in Malz et al. (2019), similar to the present research, wind data were clustered into characteristic profile shapes and sorted by average wind speed. This allowed for the initial guesses of every subsequent optimization to be based on the previous results to reduce the overall computation times. Aull et al. (2020) explored the design and sizing of fly-gen rigid-wing systems based on a steady-state model with simple aerodynamic and mass-scaling approximations. The wind resource was described by an exponential wind shear

model with Weibull distribution. Bechtle et al. (2019) used ERA5 data to assess the wind resource at higher altitudes for entire Europe. The authors describe the available wind energy without accounting for a specific conversion mechanism. The investigation included a description of wind speed and probability for several heights. Schelbergen et al. (2020) compared the energy production computed from this data set to the production computed from the Dutch Offshore Wind Atlas (DOWA) and light detection and ranging (LiDAR) data. Principal component analysis and $k$-means clustering was used to determine representative wind speed profiles for a part of the Netherlands and the North Sea. They derived power curves and estimated AEP from wind statistics for several locations. Faggiani and Schmehl (2018) investigated aspects of joint operation, such as spacial stacking of the systems and phase-shifted operation of several 100 kW soft wing pumping kite systems arranged in a wind park. Performance was estimated using a quasi-steady-state model (QSM) (Schmehl et al., 2013; van der Vlugt et al., 2019), similar to the one used in the present research, subject to a standard logarithmic wind profile.

Wind profiles are governed by weather phenomena, environmental and location-dependent conditions (e.g. surface roughness) on a multitude of temporal and spatial scales. The preferred means of determining wind conditions for wind energy converters are long-term, high resolution measurements, which at mid-altitudes can solely be achieved by long-range remote sensing such as LiDAR or SoDAR (sonic detection and ranging). Measuring wind conditions at mid-altitudes is costly and difficult, due to reduced data availability (Sommerfeld et al., 2019a). Additionally, publicly available measurements are scarce. Therefore, we decided to base this study on wind data derived from Weather Research and Forecasting model (WRF) mesoscale simulations (Skamarock et al., 2008). However, the described trajectory optimization methodology can be applied to any wind data set such as wind atlas data or measurements. Numerical mesoscale weather prediction models such as the WRF, which is well known for conventional WT siting applications (Salvação and Guedes Soares, 2018; Dörenkämper et al., 2020), are used to estimate wind conditions on time scales of a few minutes to years. Sommerfeld et al. (2019b) compares the simulated onshore data used in this study, located in northern Germany near the city of Pritzwalk, to LiDAR measurements and found a good, but altitude-dependent agreement between both data sets . The simulated offshore conditions used in this study can be references against data at the FINO3 research platform in the North Sea. This study investigates AWES performance subject to 10-minute average wind data, which is the standard for conventional WT, while the New European Wind Atlas (NEWA) only provides 30-minute average data (Witha et al., 2019). We use this higher resolution wind data because the higher temporal, spatial and vertical resolution reduces averaging and allows for the investigation of more realistic wind conditions.

The key contribution of this paper is the investigation of power-optimal AWES performance subject to realistic onshore and offshore wind profiles and its impact on average cycle power variation. Therefore, WRF-simulated wind data are used instead of assuming a wind profile relationship such as the logarithmic or exponential wind profile. Furthermore, this study is a continuation of previous analyses of LiDAR measurements (Sommerfeld et al., 2019a) and WRF simulations (Sommerfeld et al., 2019b) at the onshore location. To justify the realism of the data location specific characteristics are described. The data are categorized using $k$-means clustering which classifies the wind data at each location into groups of similar wind speed and vertical profile shape. From these clusters three representative profiles are sampled and implemented into the awebox optimization toolbox as boundary conditions. By selecting these profiles based on their average wind speed between 100 and

400 m, which is an apriori guess of the pattern trajectory height (Airborne Wind Europe, 2021), we use actual simulated data instead of averaged data. By choosing the 5th, 50th, 95th percentiles, we encompass the most likely operating conditions within each cluster and avoid non-representative profile extrema. This drastically reduces the computational cost as only few selected profiles are needed to represent the entire wind spectrum. This study only uses 60 out of 52560 10-minute wind speed profiles. This is sufficient for the scope of this study, which includes the analysis of representative operating conditions and the estimation of power curves.

The `awebox` optimization model allows for the investigation of dynamic performance parameters, such as aircraft trajectories, tether tension, tether speed and power which highly depend on the wind conditions. The aircraft model is based on the well investigated and published Ampyx Power AP2 prototype (Licitra, 2018; Malz et al., 2019; Ampyx, 2020), scaled to a wing area of $A = 20\ \mathrm{m}^2$. The optimized average cycle powers are referenced against optimal performance subject to a simple logarithmic wind speed profile, a quasi-steady-state reference AWES model (QSM), and a steady-state WT power estimation. The apriori guess of 100 and 400 meter reference height is verified by comparing AWESs power curves over wind speed at reference height to wind speed at pattern trajectory height.

The paper is structured as follows: Section 2 introduces the mesoscale WRF model setup, first Section 2.2 analyzing the onshore and offshore wind resource, then Section 2.3 introducing the $k$-means clustering algorithm and summarizing the results of clustered wind profiles (both longitudinal and lateral wind components). For visualization purposes data are shown for $k = 10$ clusters, while 20 clusters are used in the later analysis. Section 3 introduces the dynamic AWES model, comprising of aircraft, tether and ground-station models. Section 4 describes the `awebox` optimization framework, summarizes the aircraft parameters, system constraints and initial conditions. This is followed by a description of the wind, WT and AWES reference models in Section 5. Section 6 presents the results which include flight trajectories and time series of various performance parameters, a statistical analysis of tether length and operating altitude as well as a power curve estimation. Finally, Section 7 concludes with an outlook and motivation for future work.

**2 Wind conditions**

As of now no universally accepted mid-altitude AWES reference wind model exists. Therefore, we analyze onshore and offshore wind conditions based on the mesoscale WRF model introduced in Sub-section 2.1. Sub-section 2.2 analyses wind statistics to give an insight into the wind regime at both locations. Clustering, introduced in Sub-sections 2.3, is used to determine groups of similar vertical wind profiles from which unaveraged, representative profiles are sampled. 
[revised manuscript text omitted]
  profiles, also referred to as centroids. Their color corresponds to the average wind speed between 100 and 400 m. All WRF-simulated wind speed profiles are depicted in gray. Clusters are sorted by average centroid speed between 100 and 400 m, represented by their colors and labels ($C = 1 - 10$).

[Figure]

[Figure]

**Figure 5.** Onshore (left) and offshore (right) average annual wind speed profiles (centroids) resulting from $k$-means clustering for $k = 10$ (a,b). All comprising WRF-simulated wind speed profiles are depicted in gray. The centroids are sorted, labeled and color coded in ascending order of average wind speed between 100 and 400 m. The corresponding cluster frequency of occurrence $f$ for each cluster $C$ is shown in (c) and (d) below.

As expected, offshore (Figure 5 b) low altitude wind speeds are higher and wind shear is lower than onshore (Figure 5 a). Overall, offshore centroids  wider spread in comparison to the onshore profiles and do not show a wind speed reversal. This indicates more homogeneous wind conditions offshore and a higher likelihood of LLJs onshore. At both locations, the first two clusters exhibit very low wind shear with an almost constant wind speed above 200 m. These low wind speed clusters amount to approximately 25 % onshore (c) and 20% offshore (d), as can be seen in the corresponding cluster frequency of occurrence $f$. A standard logarithmic wind profile does not accurately describe such almost constant profiles which could lead to an overestimation of wind speeds at higher altitudes. Therefore, AWESs need to be able to either operate under such low

255 wind speeds or need to safely land and take-off. Onshore clusters 4 and 5 seem to mostly comprise of non-monotonic profiles as these centroids show a distinct LLJ nose at about 200 m and 300 m. Onshore centroids of clusters 7 and 8 also show a slight wind shear inversion at higher altitudes.

Within a cluster, the wind speed profiles span a fairly narrow range of wind speeds indicating coherent clusters. Figure 6 shows the distribution of wind speed profiles within each of the clusters.

[Figure]

**Figure 6.** Vertical onshore wind speed profiles categorized into $k = 10$ clustered using the $k$-means clustering algorithm. Later analyses use $k = 20$ clusters. The average profile (centroid) is shown in blue and the profiles associated with this cluster are shown in gray. Clusters 1 to 10 (a-j) are sorted and labeled in ascending order of average centroid wind speed between 100 m and 400 m. The corresponding cluster frequency $f$ for each cluster $C$ is shown in Figure 5. The red lines mark the wind speed profile with the 5th, 50th and 95th percentiles of average wind between 100 and 400 m within each cluster.

260 The clusters C = 1 (a) to C = 10 (j) are sorted by average centroid (blue line) wind speed between $U(z_{\mathrm{ref}} = 100 - 400 \text{ m})$. The red lines indicate the profile associated with the 5th, 50th and 95th percentiles of $U(z_{\mathrm{ref}})$ within each cluster. To reduce computational cost, only these profiles are later implemented into the awebox optimization framework. We chose these profiles because they are less likely to be an irregular outliers of their respective cluster than the cluster's extrema. Furthermore, these profiles describe the in-cluster variation with respect to wind speed and profile shape. The focus of this study is the investigation
265 of AWES performance subject to realistic wind conditions, which is why we opted against using averaged or scaled data, such as the cluster centroids or normalized wind speed profiles. The equivalent offshore clusters can be found in Figure A1 in the

appendix. Evidently, the wind speed magnitude plays a dominant role in the clustering process. This can lead to profiles with different shapes to be assigned to the same cluster due to similar average wind speed. A clearer wind profile shape distinction could be achieved by normalizing the data before clustering it (Molina-García et al., 2019; Schelbergen et al., 2020).

[Figure]

**2.5 Analysis of clustered statistics**

This subsection investigates the correlation between clusters and monthly (Figure 7), diurnal (Figure 8) and atmospheric stability (Figure 9) for the onshore (top row) and offshore (bottom row) location. This reveals patterns within the clusters, gives an insight into the wind regime and informs AWES performance for a given time and location. Here only $k = 10$ clusters are chosen for presentation purposes, but wind data from $k = 20$ will be investigated in later sections. Clusters are sorted in ascending order of average centroid wind speed $U(z_{\mathrm{ref}} = 100 - 400 \text{ m})$ and color coded accordingly. The corresponding centroids are shown in Figure 5.

[revised manuscript text omitted]

$$c_{D,\text{total}} = c_{D,\text{wing}} + \frac{1}{4}\frac{ld}{A}c_{D,\text{tether}}\qquad(2)$$

See Houska and Diehl (2007); Argatov and Silvennoinen (2013) and van der Vlugt et al. (2019) for details.

We approximate the wing's lift coefficient $c_L$ (Figure 10 a) by a quadratic function to simulate stall effects. A polynomial description for the entire range of AoA is necessary, as the optimization algorithm requires a two-times differentiable function. As a result, the implemented $c_L$ (blue) slightly exceeds the linear (orange) lift coefficient $c_L^{\text{ref}}$ between $-5 \leq \alpha \leq 10°$. The pitch moment (Figure 10 c) is assumed to behave linearly and changes in the drag coefficient (Figure 10 b) are approximated by a quadratic function. Tether drag is independent of aircraft angle of attack and therefore added to the zero-lift drag coefficient $c_{D0}$. $c_R$ is represented as the resultant aerodynamic force coefficient:

$$c_R = \sqrt{c_L^2 + c_{D,\text{total}}^2}. \tag{3}$$

The maximum values of the glide ratio $c_L/c_{D,\text{total}}$ (Figure 10 e) and the ratio $c_R^3/c_{D,\text{total}}^2$ (Figure 10 f) which is one of the main determining factors of AWES power (Schmehl et al., 2013; Loyd, 1980), decrease significantly with tether length and shift towards higher angles of attack. This effect is less pronounced for larger wings because the effect of tether drag reduces when scaling up to larger aircraft.

**3.3 Aircraft mass model**

The aircraft dynamics are described by a single rigid body of mass $m_{\text{aircraft}}$ and moment of inertia $\mathbf{J}$, subject to aerodynamic forces and moments. The inertial properties $m_{\text{aircraft}}$ and $\mathbf{J}$ are determined by upscaling the AP2 reference wing from $A_{\text{wing}}^{\text{AP2}} = 3 \text{ m}^2$ to $A = 20 \text{ m}^2$. The Mass $m_{\text{scaled}}$ and moment of inertia $\mathbf{J}_{\text{scaled}}$ of a fixed wing aircraft scale as functions of the wing span $b$ with a mass-scaling exponent $\kappa$.

$$m_{\text{scaled}} = m_{\text{ref}} \left( \frac{b}{b_{\text{ref}}} \right)^{\kappa}, \tag{4}$$

$$\mathbf{J}_{\text{scaled}} = \mathbf{J}_{\text{ref}} \left( \frac{b}{b_{\text{ref}}} \right)^{\kappa+2}. \tag{5}$$

Pure geometric scaling corresponds to Galileo's square-cube law with $\kappa = 3$. In reality, as has been seen for the development of conventional WTs, design and material improvements occur over time. A review of the available literature containing system mass details was conducted to identify an appropriate mass-scaling factor. The results are shown in Figure 11 depicting actual and anticipated AWES scale bounded by $\kappa = 2.2 - 2.6$ (gray area). We chose $\kappa = 2.4$ based on a curve fit of the available published sizing study data. This seems quite ambitious and might be achievable for soft wing kites. The mass of these hollow tensile structures filled with air mostly scales the wing area, leading to significantly lower mass-scaling exponents and more beneficial mass-scaling. (Sommerfeld et al. (2020) investigates the effect of variable mass-scaling exponents.

[Figure]

**Figure 11.** Curve fit of published sizing studies aircraft mass (Haas et al., 2019; Kruijff and Ruiterkamp, 2018; Eijkelhof et al., 2020; Ampyx, 2020; Echeverri et al., 2020). For these data mass scales within a scaling exponent range of $\kappa = 2.2 - 2.6$ (gray area). The chosen mass-scaling exponent of $\kappa = 2.4$ is represented by a dashed line and the investigated scaled AP2 design is highlighted by a black square.

**4   Optimal control Model**

AWES need to dynamically adapt to changing wind conditions to optimize power generation and ensure save operation. This section introduces the dynamic trajectory optimization awebox toolbox (De Schutter et al., 2020) (Sub-section 4.1) and describes the most important boundary (Sub-section 4.2) and initial conditions (Sub-section 4.3 . Sub-section 4.4 explains the implementation of the previously described wind profiles. A polynomial fit through the simulated data points is needed, as the gradient-based optimizer requires an at least two times differntiable function.

**4.1   AWES model overview**

Only one production cycle, including reel-in and reel-out, is optimized. Take-off and landing are not considered. Maximizing the average cycle power can be formulated as an trajectory optimization problem which combines the interaction between tether, flying wing and ground station. This study analyzes the mechanical power produced by a single tethered aircraft with a straight tether. Power production is intrinsically linked to the aircraft's flight dynamics, as the AWES never reaches a steady state over the course of a power cycle. Generating dynamically feasible and power-optimal flight trajectories is nontrivial, given the nonlinear and unstable system dynamics and the presence of nonlinear flight envelope constraints. Optimal control methods

are a natural candidate to tackle these problems, given their inherent ability to deal with nonlinear, constrained multiple-input-multiple-output systems (De Schutter et al., 2019; Leuthold et al., 2018). This trajectory optimization is a highly nonlinear
and non-convex problem which can have multiple local optima, depending on initialization. The initial and final states of each trajectory are freely chosen by the optimizer but must be equal to ensure periodic operation. In periodic optimal control, an optimization problem is solved by computing periodic system state and control trajectories that optimize a performance index (here average power output $\overline{P}$) while satisfying the system's dynamic equations. We apply this methodology to WRF-simulated wind speed profiles to generate a range of realistic trajectories. The temporal development of important operational parameters is illustrated to better understand instantaneous performance and estimate average cycle power. Any wind data sets, such as wind atlas data, LiDAR or met mast measurements can be implemented into the optimization model via a twice differentiable function, depending on the scope and purpose of the investigation.

**4.2 Constraints**

Several important constraints define the operational envelop. The most important constraints such as tether length, speed and force are summarized in Table 3. The following constraints define a representative and not optimized AWES design.

The power of ground generation AWES is limited by the tether force, which is defined by the tensile strength ($\sigma_{\max}^{\text{tether}}$) and tether diameter, and the tether speed. The tether diameter is chosen such that the maximum tether tension is about $F_{\text{tether}}^{\max} = 50$ kN with a safety factor of SF = 3. This results a peak power of $P_{\text{peak}} \approx 500$ kW, assuming a maximum reel-out speed of $\dot{l} = 10$ ms$^{-1}$. This corresponds to a rated average cycle power of approximately $P_{\text{rated}} \approx 260 - 300$ kW. The tether length constraint is very loos, to allow the optimizer to investigate a wide range of possible operating heights. We assume a reel-out to reel-in ratio of $\frac{2}{3}$ to be within winch design limitations. Flight envelope constraints include limitation of aircraft acceleration, roll and pitch angle (to avoid collision with the tether) and angle of attack. Furthermore, a minimal operating height of $z_{\min} = 50 + \frac{A_{\text{wing}}}{2}$ m is imposed for safety reasons.

**4.3 Initialization**

The results generated by the highly nonlinear and non-convex trajectory optimization can have multiple local optima. These solutions, for which only local optimality can be guarantied, depend on the chosen initialization. Some of these local optima can have unwanted characteristics, which is why the quality of all solutions needs to be evaluated a posteriori. To solve this complex problem, initial guesses are generated using a homotopy technique similar to Gros et al. (2013). The homotopy technique initially fully relaxes the dynamic constraints using fictitious forces and moments to reduce model nonlinearity and coupling, improving the convergence of Newton-type optimization techniques. The constraints are then gradually re-introduced until the relaxed problem matches the original problem. The optimization is initialized with a circular trajectory with a fixed number of $n_{\text{loop}} = $ five loops at a 30° elevation angle, an initial tether length $l_{\text{init}} = 500$ m, in positive $x$ direction and an estimated aircraft speed of $v_{\text{init}} = 10$ ms$^{-1}$. This initialization is kept constant for all wind speed profiles. The number of loops is not part of the objective function and does therefore not change with wind speed. The impact of the number of loops needs to be investigated further, but previous analyses showed that the awebox-estimated average cycle power is rather insensitive to the number of

**Table 3.** Selected AWES design parameters for the original AP2 reference system (Malz et al., 2019) and the scaled $A = 20$ m$^2$ design, analyzed in this study. Values in square brackets represent the upper and lower bounds, which are implemented as inequality constraints.

| | Parameter | AP2 | design 1 |
|---|---|---|---|
| Aircraft | $A$ [m$^2$] | 3 | 20 |
| | $c_{\text{wing}}$ [m] | 0.55 | 1.42 |
| | $b_{\text{wing}}$ [m] | 5.5 | 14.1 |
| | AR [-] | 10 | 10 |
| | $m_{\text{aircraft}}$ [kg] | 36.8 | 355 |
| | $\alpha$ [°] | | [-10 : 30] |
| | $\beta$ [°] | | [-15 : 15] |
| Tether | $l$ [m] | | [1 : 2000] |
| | $\dot{l}$ [ms$^{-1}$] | | [-15 : 10] |
| | $\ddot{l}^{\text{max}}$ [ms$^{-2}$] | | [-10 : 10] |
| | $d$ [mm] | | 7.3 |
| | $\sigma_{\text{max}}^{\text{tether}}$ [Pa] | | 3.6×10$^9$ |
| | SF [-] | | 3 |
| Operational | $z_{\text{min}}$ [m] | | 60 |
| | $\alpha$ [°] | | [-10 20] |
| | $\beta$ [°] | | [-5 5] |

loops. It is likely beneficial to reduce the number of loops with wind speeds because higher wind speeds reel out faster and reach maximum tether length faster.

**4.4 Wind profile implementation**

This study investigates WRF-simulated wind data, instead of assuming a wind profile relationship such as the logarithmic or exponential wind profile. These relationships do not appropriately represent wind conditions above earth's surface layer (Optis et al., 2016) and cannot emulate the variety of non-monotonic and non-logarithmic wind profiles which occur at both locations. This is particularly important for AWES which can benefit from and need to be able to operate in these conditions.

To reduce the computational cost while maintaining an adequate representation, we only implement three wind velocity profiles from each cluster into the trajectory optimization framework. More profiles could be chosen for an in-depth analysis. A total of 60 profiles, three profiles for each of the $k$ = 20 clusters (Section 2.3), for each location are optimized. The three selected profiles correspond to the 5th, 50th and 95th percentiles of average wind speed $U(z_{\text{ref}} = 100 - 400$ m) within each cluster. We assume that these profiles represent the cluster's spectrum of wind conditions at operating height.

The longitudinal $u$ and lateral $v$ wind components of the sampled WRF-simulated wind profiles are rotated such that the main wind direction $u_{\text{main}}$, defined as the average wind direction between 100 and 400 m, is pointing in positive $x$-direction

and the transverse component $u_{\mathrm{dev}}$ in $y$-direction. This is equivalent to assuming omnidirectional operation. The `awebox` software uses the gradient-based MA57 solver (HSL, 2020) in IPOPT (Waechter and Laird, 2016) to solve the non-linear control problem. Therefore, it is necessary to interpolate the  vertical wind speed profiles with a twice continuously differentiable function. We chose to use Lagrangian polynomials (Abramowitz and Stegun, 1965) because the resulting polynomials pass through the input data points. To avoid over fitting a limited number of data points are implemented. These data points are chosen based on the anticipated operating height, to best represent the wind conditions at relevant heights.

For comparison, logarithmic wind speed profiles, with a roughness length of $z_0^{\mathrm{onshore}} = 0.1$ and $z_0^{\mathrm{offshore}} = 0.001$, are implemented into the trajectory optimization framework

$$U_{\log} = U_{\mathrm{ref}} \left( \frac{\log_{10}(z/z_0)}{\log_{10}(z_{\mathrm{ref}}/z_0)} \right). \tag{6}$$

The reference wind speed $U_{\mathrm{ref}}$, at reference height $z_{\mathrm{ref}} = 10\,\mathrm{m}$, varies from 3 to $20\,\mathrm{ms}^{-1}$ with a step size of $\Delta U_{\mathrm{ref}} = 1\,\mathrm{ms}^{-1}$. The `awebox` includes a simplified atmospheric model based on international standard atmosphere to account for air density variation.

**5 Reference models**

This section introduces reference models  to analyze and contextualize the optimization results. To compare the optimization results to analytic solutions, we define a quasi-steady-state AWES reference model (QSM) (Sub-section 5.1) and a steady-state WT model (Sub-section 5.2).

**5.1 AWES reference model**

The QSM estimates the mechanical power of ground-generation AWES based on the assumption that the trajectory of the tethered aircraft can be approximated by a progression through steady equilibrium states where tether tension and total aerodynamic force are aligned. The QSM, based on Argatov et al. (2009) and generalized by Schmehl et al. (2013), approximates the aircraft as a point mass. Its position is described in  terms of spherical coordinates  , i.e. the radial distance from the ground station, the elevation angle $\varepsilon$ mean? and azimuth angle $\phi$ relative to the wind velocity vector $U$. For lightweight soft-wing kites, this is a reasonably good approximation because the low mass of the kite leads to very short acceleration times. The model includes losses caused by the misalignment of the tether and wind velocity vector. The same model parameters and constraints of the optimization model also apply to the QSM reference model (see Sub-section 4.2).

The average cycle power $P_{\mathrm{QSM}}$ can be estimated from the reel-out power $P_{\mathrm{opt}}$, the power losses during reel-in $P_{\mathrm{in}}$:

$$P_{\mathrm{QSM}} = P_{\mathrm{out}}\, t_{\mathrm{out}} - P_{\mathrm{in}}\, t_{\mathrm{in}} = P_{\mathrm{out}} \frac{\dot{l}_{in}}{\dot{l}_{out} + \dot{l}_{in}} - P_{\mathrm{in}} \frac{\dot{l}_{out}}{\dot{l}_{out} + \dot{l}_{in}}. \tag{7}$$

We assume reel-in power losses $P_{\mathrm{in}}$ to be zero because optimal reel-in tether tension is negligible. This reduces the average cycle power by up to 30%, depending on wind speed. Due to the cyclic nature of the trajectory, we can determine the ratio of

the reel-in time $t_{\text{in}}$ and reel-out time $t_{\text{out}}$ to the total cycle time from the reel-in speed $\dot{l}_{in}$ and reel-out speed $\dot{l}_{out}$. $\dot{l}_{out}$ depends on the wind speed, while the $\dot{l}_{in} = -15\text{ms}^{-1}$ is assumed to be the maximum reel-in speed. We assume a constant tether force $F_{\text{tether}}$ and tether speed during reel-in and reel-out. The transition time between both phases is neglected. $P_{\text{out}}$ is calculated from the product of tether speed $\dot{l}$ and tether tension $F_{\text{tether}}$:

$$P_{\text{out}} = F_{\text{tether}}\,\dot{l}_{\text{out}} = \frac{\rho_{\text{air}}}{2} A v_{\text{app}}^2 c_{\text{R}} \left(\frac{c_{\text{R}}}{c_{\text{D,total}}}\right)^2 \dot{l}_{\text{out}}. \tag{8}$$

Tether tension is a function of wind speed magnitude $U$, air density $\rho_{\text{air}}$ and the resultant aerodynamic force coefficient $c_{\text{R}}$ as defined in (Equation (3)), which is calculated from the aerodynamic lift $c_{\text{L}}$ and total drag coefficient $c_{\text{D,total}}$ (Equation (2)), including wing and tether drag. The tether speed $\dot{l}$ is non-dimensionalized in the form of by defining the reeling factor:

$$f = \frac{\dot{l}}{U}, \leq \cos\varepsilon\cos\phi. \tag{9}$$

which is constrained by the elevation $\varepsilon$ and azimuth angle $\phi$ as the magnitude of the apparent wind speed cannot be negative. Combining equations (8) and (9) results in:

$$P_{\text{out}} = \frac{\rho_{\text{air}}}{2} A U^3 c_{\text{R}} \left(\frac{c_{\text{R}}}{c_{\text{D,total}}}\right)^2 f\left(\cos\varepsilon\cos\phi - f\right)^2. \tag{10}$$

The optimal reeling factor is $f_{\text{opt}} = \frac{1}{3}\cos\varepsilon\cos\phi$ which can be derived from Equation (10) by a simple extreme value analysis. $F_{\text{tether}}$ is constrained by the tether diameter $d$, the tensile strength $\sigma_{\text{max}}^{\text{tether}}$ and the safety factor SF.

$$F_{\text{tether}} \leq \frac{d^2}{4}\pi\sigma_{\text{max}}^{\text{tether}} \tag{11}$$

The same sampled WRF-simulated wind profiles (Section 2, Sub-section 2.3) and presented in as implemented into the dynamic optimization framework are also investigated using the QSM. We maximize the cycle average power $P_{\text{QSM}}$ by varying $l$, $\dot{l}$ and $z$ and assuming an optimal ratio $\frac{c_{\text{R}}^3}{c_{\text{D,total}}^2}$. The aircraft is assumed to move directly crosswind with a zero azimuth angle $\phi$ relative to the wind direction.

**5.2 WT reference model**

This section introduces a simple steady-state WT model to contextualize the AWES performance. WT power is estimated by:

$$P_{\text{WT}} = \frac{1}{2}\rho_{\text{air}}c_{\text{p}}^{\text{WT}} A_{\text{WT}} U^3 (z_{\text{WT}} = 100\text{ m}) \tag{12}$$

with a hub height of $z_{\text{WT}} = 100\,\text{m}$ for both onshore and offshore conditions. The rotor diameter $D_{\text{WT}} \approx 26.9\,\text{m}$ is sized such that an equivalent rated power of $P_{\text{rated}} = 260\,\text{kW}$ is reached at a rated wind speed of $v_{\text{rated}}(z_{\text{WT}} = 100\text{ m}) = 12\text{ ms}^{-1}$, assuming a constant power coefficient of $c_{\text{p}}^{\text{WT}} = 0.45$. The power is kept constant above the rated wind speed. Performance is compared based on the same sample of WRF-simulated wind speed profiles.

**6 Results and discussion**

This section analyses the optimization results and compares them to the reference models. Sub-section 6.1 investigates power-optimal trajectories and the time series of important operational parameters. Sub-section 6.2 examines operating height statistics, tether length and elevation angle trends. Sub-section 6.3 visualizes the impact of differences reference heights on a power curve approximation by comparing average cycle power over $U(z_{\rm ref} = 100 \text{ m})$, $\overline{U}(z_{\rm ref} = z_{PTH})$ and an apriori guess of the wind speed at pattern  height $\overline{U}(100 \text{ m} \leq z_{\rm ref} \leq 400 \text{ m})$. Lastly, Sub-section 6.4 compares the average cycle power based on simulated WRF wind conditions to logarithmic wind speed profiles and contextualizes the data by comparing them to QSM and WT power. All results are subject to the constraints and design parameters introduced in Sections 3 and 4 and do not represent general ground-generation AWES.

**6.1 Flight trajectory and time series results**

Figure 12 compares power-optimal flight trajectories  for a range of  onshore wind conditions. The reference wind speed mentioned in the legend is the apriori guess  at the pattern trajectory height $U_{\rm ref} = \overline{U}(100 \text{ m} \leq z_{\rm ref} \leq 400 \text{ m})$. These results have been chosen to visualize typical  optimized trajectories  for realistic wind conditions. (determined with awebox)

Figure 12 (a) shows the wind speed  $U$ over altitude $z$. Figure 12 (c) shows the corresponding top view of the wind velocity profiles, rotated such that $u_{\rm main}$ points in positive $x$ direction. The WRF-simulated wind profiles are shown in gray. The highlighted segments depict the Lagrangian polynomial fit (Abramowitz and Stegun, 1965) at operating heights, which sufficiently fits the wind data. Figures 12 (b) and (d) show a side ($x - z$ plane) and top view ($x - y$ plane) of the optimized trajectories The optimization predicts an increase in tether length and stroke length with wind speed. Similar results for the offshore location can be found in Figure A2 in the appendix.

Figure 13 illustrates the corresponding temporal development of important operational parameters. [Remove line break?]

The optimizer maximizes tether tension (Figure 13 (a)) during reel-out even for lower wind speed and adjusts the reel-out speed (Figure 13 (c)) to maximize average cycle power. This increases the reeling factor beyond its optimal value of $f_{\rm opt} = \frac{1}{3}\cos\varepsilon\cos\phi$ and increases power with wind speed even though the maximum tether force is reached. The resulting instantaneous power is shown in Figure 13 (e). The low wind speed example $U_{\rm ref} = 5.4\,{\rm ms}^{-1}$ (blue) seems to be just above cut-in wind speeds. Its tether speed drops to zero for an extended amount of time during the reel-out phase to maintain sufficient lift to keep the aircraft aloft. The production period remains almost constant ($t \approx 60\,{\rm sec}$) for the moderate and high wind speed trajectories (orange, green and red), while the reel-in period increases with wind speed, due to the increased reel-out length caused by a higher average reel-out speed. Significant power losses only occur during the transition between the production and retraction phases. During the reel-in phase the tether speed is maxed out while tether tension drops to zero and the aircraft reduces its angle of attack (Figure 13 (d)) to reduce lift. At higher wind speeds the optimizer  increases the elevation angle and reduces angle of attack to stay within the constraints. This can results in odd or unexpected trajectories, even though these local minima are feasible solutions within the system constraints. Tether length

[Figure]

**Figure 12.** Exemplary onshore trajectories. Wind data are based on sampled WRF-simulated clusters  Vertical wind speed profiles (a), and hodograph (top view) of wind velocity up to 1000 m (c). The  trajectory sections indicate operating wind conditions highlighted by coloring. Panel (b) and panel (d) show the side and top view of the corresponding `awebox`-optimized trajectories. The reference wind speed in the legend is $U_{\mathrm{ref}} = \overline{U}(100\ \mathrm{m} \le z_{\mathrm{ref}} \le 400\ \mathrm{m})$. The results correspond to the time series shown in Figure 13.

(Figure 13 (f)) generally increases with wind speed as the  reels out faster, increases its elevation angle and operates at higher altitude. Similar results for the offshore location can be found in Figure A3 in the appendix.

**6.2 Tether length, elevation angle and operating altitude**

This sub-section compares tether lengths $l$, elevation angles $\varepsilon$ and operating heights $z_{\mathrm{operating}}$ resulting from the trajectory

505 optimization of 60 wind velocity profiles from $k = 20$ clusters. Figure 14 (a) illustrates the range of onshore (blue) and offshore (orange) tether lengths  for each wind velocity profile. The maxima and minima are highlighted by circles and plotted over reference wind speed $\overline{U}(z_{\mathrm{ref}} = 100 - 400\ \mathrm{m})$.

None of the optimizations max out the tether length constraint of $l^{\max} = 2000$ m. Both locations show a trend towards longer tether lengths up to rated wind speed, where the reel-out speed and tension are almost constant and close to maximum

510 (Figure 13). A longer tether is not beneficial as the AWES needs to stay within design constraints and the additional drag and weight would only reduce performance. The maximum tether length remains almost constant above rated wind speed while the minimum tether length increases slightly, reducing the total stroke length. The elevation angle (Figure 14 (b)) decreases

[Figure]

**Figure 13.** Time series of instantaneous tether tension (a), apparent wind speed (b), tether-reeling speed (c), angle of attack α (d), power (e) and tether length (f). The results correspond to the trajectories, based on sampled onshore WRF-simulated wind data, shown in Figure 12.

[Figure]

**Figure 14.** Tether length range (a) over reference wind speed $U(z_{\text{ref}} = 100 - 400\text{ m})$ and frequency distribution of operating altitude (b) based on `awebox` trajectory optimization of $k = 20$ onshore (blue) and offshore (orange) clusters.

as the tether length increases. The optimizer tries to keep the elevation angle low in order to reduce misalignment (cosine) losses between the tether and the horizontal wind velocity vector. The onshore elevation angle is slightly higher due to the higher wind shear which justifies higher operating altitudes. This can also be seen in Figure 14 (c) which shows the frequency distribution of operating altitude $z_{\mathrm{operating}}$. 78.6 % onshore and 74.7 % offshore the optimal operating heights are below 400 m, confirming the findings in Sommerfeld et al. (2019a, b). Larger or multi-kite AWES could benefit from higher operating altitudes due to their higher lift to tether drag ratio and weight ratio, but more detailed analyses are required.

**6.3  Impact of reference height on power curve**

The power curve of wind energy converters depicts the average power over reference wind speed. For conventional WT the wind speed at hub-height is commonly used as reference wind speed. Whether this is appropriate for ever growing towers an longer WT blades is debatable. Defining a reference wind speed for AWES is not trivial, as they change their operating height with wind speed, during each cycle and dependent on wind speed profile shape. The choice of reference wind speed impacts the power curve representation. The AWE Glossary (Airborne Wind Europe, 2021) recommends to use the wind speed at pattern trajectory height $z_{\mathrm{PTH}}$, which is the expected or logged time-averaged height during the power production phase, as reference wind speed. We estimate $100\,\mathrm{m} \leq z_{\mathrm{ref}} \leq 400\,\mathrm{m}$ as an apriori guess of the wind speed at pattern trajectory height. We do not claim to define a general power curve, but rather investigate the variaton of average cycle power caused by realistic wind profiles. Figure 15 compares onshore (a) and offshore (b) average cycle power over $U(z_{\mathrm{ref}} = 100\,\mathrm{m})$ (blue), $\overline{U}(z_{\mathrm{ref}} = z_{PTH})$ (green) and an apriori guess of the wind speed at pattern trajectory height $\overline{U}(100\,\mathrm{m} \leq z_{\mathrm{ref}} \leq 400\,\mathrm{m})$ (orange).

[Figure]

**Figure 15.** Onshore (a) and offshore (b) AWES power curve approximations over wind speed at $z_{\mathrm{ref}} = 100\,\mathrm{m}$ (blue), $100\,\mathrm{m} \leq z_{\mathrm{ref}} \leq 400\,\mathrm{m}$ (orange) and $z_{\mathrm{ref}} = z_{\mathrm{PTH}}$ (green) reference height. The dashed lines are least-square spline interpolation with a knot at $U_{\mathrm{ref}} = 13\,\mathrm{ms}^{-1}$.

530     The data points correspond to the clustered and sampled WRF-simulated wind speed profiles. The dashed lines, which are only added as visual aid, are a least-square spline interpolation of the approximately 60 data points with a knot at $U_{\mathrm{ref}} = 12\,\mathrm{ms}^{-1}$. This spline definition is chosen to account for the difference in power up to and above rated wind speed.

From these results we conclude that the choice of reference height is more significant onshore. The onshore wind conditions with their higher number of non-monotonic wind speed profiles and higher wind shear lead to larger deviations [] from the typical power curve shape. The higher

535 wind shear onshore leads to a shift towards lower wind speeds for a reference height of $z_{\mathrm{ref}} = 100$ m. The apriori pattern trajectory height guess of $100\,\mathrm{m} \leq z_{\mathrm{ref}} \leq 400\,\mathrm{m}$ is relatively close to the actual $z_{\mathrm{PTH}}$, especially for lower wind speeds. At very high wind speeds above $\overline{U}_{\mathrm{ref}} \geq 20\,\mathrm{ms}^{-1}$ the $z_{\mathrm{PTH}}$ power shifts towards higher wind speeds indicating an increased [] operating height [].

    The more homogeneous offshore wind conditions result in less power variation. The three different reference heights have

540 almost no impact on the offshore power curve up to the rated wind speed. Above $\overline{U}_{\mathrm{ref}} \geq 20\,\mathrm{ms}^{-1}$ the power curves diverge and the average cycle power decreases. This seems to be a result of the awebox optimization and its initialization with a fixed number of loops. As the wind speed and reel-out speed increase, the aircraft cannot complete all the loops before reaching the maximum tether length and transitioning into reel-in. Therefore, one of the loops is performed when already [] reel-in [], leading to an increase in tether tension (Figure 16 (a)) and additional losses during the reel-in period (Figure 16 (e)). The corresponding

545 trajectories are shown in Figure A4 in the appendix.

[Figure]

**Figure 16.** Time series of instantaneous tether tension (a), apparent wind speed (b), tether-reeling speed (c), angle of attack $\alpha$ (d), power (e) and tether length (f) for high speed WRF-simulated offshore wind conditions. The results correspond to the trajectories shown in Figure A4.

**6.4 Reference model power comparison**

Figure 17 presents the impact of the wind speed profile shape on optimized average cycle power $\overline{P}$ over $\overline{U}_{\mathrm{ref}}$ ($100 \mathrm{~m} \leq z_{\mathrm{ref}} \leq 400 \mathrm{~m}$), by comparing power estimates based on sampled WRF-simulated wind data (blue) to power estimates based on standard logarithmic wind speed profiles (red). These results are verified against the QSM (Sub-section 5.1, orange) and WT reference models (Sub-section 5.2, green).

[Figure]

**Figure 17.** Onshore (a) and offshore (b) average cycle power $\overline{P}$  *as a function of* average wind speed between 100 and 400 meters. WRF (blue) data, based on  *three* wind speed profiles for each of the $k = 20$ clusters, is compared to standard logarithmic wind speed profiles (red). QSM (orange) and WT (green) which use the same sampled WRF profiles are added for reference. The onshore (c) and offshore (d) power-harvesting factor $\zeta$ is added as an performance indicator.

No cut-out wind speed is defined. The cut-in wind speed of $\overline{U}_{\mathrm{ref}} \approx 5 \mathrm{~ms}^{-1}$ is the result of unconverged optimizations below this threshold, indicating that the wind is insufficient to keep the AWES aloft. The QSM and WT model estimate power for these wind speeds. Rated power is achieved around $U_{\mathrm{rated}} \approx 12 - 15 \mathrm{ms}^{-1}$, depending on the wind speed profile shape. At this wind speed the reel-out speed is almost constant while a constant reel-out tension is already achieved at lower wind speeds (Figure 13).

The logarithmic wind speed profiles (Equation (6)) use  roughness lengths  *between* $z_0^{\mathrm{onshore}} = 0.1$ and $z_0^{\mathrm{offshore}} = 0.001$ (Sub-section 4.4). As expected, logarithmic power estimates do not fluctuate as much as the WRF-simulated power. The predicted logarithmic onshore power (Figure 17 (a)) is often slightly below WRF which  *would* indicate that these WRF profiles exhibit narrow areas of higher wind speeds, such as LLJs. Offshore, the logarithmic and WRF data are in good agreement with the

logarithmic results as most of the simulated wind profiles are more monotonic. At both locations, the higher WRF power above $U_{\mathrm{ref}} \geq 15 \mathrm{~ms}^{-1}$ is likely caused by higher than logarithmic wind shear. However, another contributing factor is the `awebox` optimization and initialization with a fixed number of loops which can lead to loops being performed during the reel-in period which leads to a reduction in average cycle power. Additionally, determining a dynamically feasible and power-optimal trajectory becomes more difficult at higher wind speeds, due to tether speed and tension constraints.

The power-harvesting factor $\zeta$ (Diehl, 2013) is an AWES performance indicator. It expresses the estimated AWES power $P$ relative to the total wind power $P_{\mathrm{area}}$ through an area of the same size as the wing $A$. Here the average wind speed between $\overline{U}_{\mathrm{ref}}(100 \leq z_{\mathrm{ref}} \leq 400 \mathrm{~m})$ is use to calculate $P_{\mathrm{area}}$, which is not a physical power, but a mathematical concept to non-dimensionalize power.

$$\zeta = \frac{P}{P_{\mathrm{area}}} = \frac{P}{\frac{1}{2}\rho_{\mathrm{air}}AU_{\mathrm{ref}}^3} \tag{13}$$

$\zeta$ can be derived from (8) by setting the elevation angle $\varepsilon$ and the azimuth angle $\phi$ to zero. An extreme value analysis results in an optimal reel-out speed $\dot{l} = 1/3\ U$ (Equation (9)) and $\zeta_{\max} = \frac{4}{27}c_{\mathrm{R}}\left(\frac{c_{\mathrm{R}}}{c_{\mathrm{D}}}\right)^2$.

Both onshore (Figure 17 (c)) and offshore (Figure 17 (d)) show similar trends in agreement with the QSM. $\zeta$ decreases with wind speed because tether tension and speed constraints need to be satisfied. Both the QSM and WT reference model use the same sampled WRF-simulated wind data. WT power fluctuates significantly due to the choice of reference height. AWESs outperform WTs up to rated wind speed, particularly onshore where AWESs can take advantage of higher wind speeds aloft. Lower wind shear offshore reduces the need to operate at higher altitudes, reducing the benefit of AWESs. As expected, the QSM predicts the highest power due to the simplified model and assumptions such as constant reel-out and reel-in power and neglected mass.

**7   Conclusions and outlook**

This study describes optimal single-aircraft, ground generation AWES performance based on sampled mesoscale WRF-simulated wind data by analyzing trajectories, instantaneous performance, operating heights and average cycle power. Throughout the paper an apriori operating heights guess of $100 \leq z_{\mathrm{ref}} \leq 400 \mathrm{~m}$ is used. This guess is verified by comparing the power curve approximations over wind speed at different reference heights. These analyses use one year of onshore wind data at Pritzwalk in northern Germany and one year of offshore wind data at the FINO3 research platform in the North Sea to drive the `awebox` optimization framework, which determines dynamically feasible, power-optimal trajectories. The model uses a scaled Ampyx AP2 aircraft with a wing area of $A = 20 \mathrm{~m}^2$ and is subject to realistic constraints. The annual wind data set is categorized into $k = 20$ clusters using $k$-means clustering algorithm. To reduce the computational cost, only three wind speed profiles per cluster are implemented into the optimization model. These profiles are sampled based on the 5th, 50th and 95th percentile of wind speed between $\overline{U}_{\mathrm{ref}}(100 \leq z_{\mathrm{ref}} \leq 400 \mathrm{~m})$ to represent the in-cluster variation. Optimal average cycle power is compared to a quasi-steady-state AWES model and a steady-state WT model.

The optimization model is able to determine power-optimal trajectories for complex, non-monotonic wind speed profiles. The results are only slightly lower than the QSM predictions, which assume constant reel-out power, no reel-in power losses and neglect gravity. The predicted AWES power exceeds the WT reference model onshore because it can utilize the higher wind shear and can operate at high wind speed altitudes such as LLJ noses. These conditions are not represented by the logarithmic wind speed profiles which is why average power is generally lower than for WRF-simulated wind. Offshore wind conditions, which are more monotonic and have less wind shear, produce similar average power as their logarithmic approximation. Due to the initialization with a fixed number of maneuvers, high wind speed trajectories show loops during the reel-in period which reduces the average cycle power. The number of loops is currently not a variable in the objective function for the awebox. This can lead to a deterioration of the trajectory at high wind speeds, as the optimizer struggles to stay within the tether tension and tether speed constraint.

An investigation of the instantaneous performance shows that the optimizer first maximizes tether tension and adjusts reel-out speed and angle of attack. With increasing wind speed the tether reel-out speed becomes more constant and approaches the maximum reel-out speed constraint. Up to rated wind speed, when average tether tension and tether speed are maximized, the optimizer increases the deployed tether length and reduces the elevation angle to operate at optimal height. At higher wind speeds, the elevation angle increases to de-power the system and stay within design constraints. As a result, approximately 75 % of the optimal onshore and offshore operating heights are below 400 m. This informs airspace regulators and companies to address the airspace restriction challenge and weakens the claim in early airborne wind energy literature of the substantial energy potential of high altitudes above 500 m. The onshore power curve estimation, using the average wind speed between $100 \leq z_{\mathrm{ref}} \leq 400$ m as reference wind speed, slightly overestimates power compared to the wind speed at pattern trajectory height, which is the expected or actual time-averaged height during the reel-out phase. Offshore, the power curve seems independent of reference height due to lower number of non-monotonic wind speed profiles and lower wind shear.

The mesoscale wind simulations, which comprise one year of wind data with a temporal resolution of 10 minutes at both locations, are categorized and analyzed. The annual wind roses for heights of 100 and 500 m confirm the expected wind speed increase and clockwise rotation at both locations. Offshore shows a lower wind shear and veer than onshore. Annual wind speed statistics reveal that low wind speeds still occur at a fairly high probability up to 1000 m at both locations. The $k$-means clustering algorithm is able to categorize the wind regime and identify LLJs as well as various non-logarithmic and non-monotonic wind profiles. The main deciding factor seems to be the wind speed, while the profile shape seems to play a less important role. Individual clusters produce coherent groups of similar wind profiles whose probability correlates with seasonal, diurnal and atmospheric stability variation. $k$-means clustering provides good insight into the wind regime, especially for higher altitudes where classification by Obukhov length is inadequate.

To follow up on this research the AEP could be derived from realistic wind conditions. AWES power curves and their reference wind speed also needs to be investigated further, for example by comparing normalized wind speed profiles including the correlation between wind speeds at different reference heights. Future work could analyze the impact of different number of initialization maneuvers or include the number of loops as a variable in the objective function of the optimization. 
[revised manuscript text omitted]

[Figure]

**Figure A2.** Exemplary offshore trajectories. Wind data are based on sampled WRF-simulated clusters (Section 2). Wind speed magnitude (a), and hodograph (top view) of wind velocity up to 1000 m (c). The highlighted sections indicate operating wind conditions. Panel (b) and panel (d) shows the side and top view of the corresponding `awebox`-optimized trajectories. The reference wind speed in the legend is $U_{\mathrm{ref}} = \overline{U}(100 \text{ m} \leq z_{\mathrm{ref}} \leq 400 \text{ m})$. The results correspond to the time series shown in Figure A3.

[Figure]

**Figure A3.** Time series of instantaneous tether tension (a), apparent wind speed (b), tether-reeling speed (c), angle of attack $\alpha$ (d), power (e) and tether length (f). The results correspond to the trajectories, based on sampled offshore WRF-simulated wind data, shown in Figure 12.

[Figure]

**Figure A4.** Exemplary offshore high wind speed trajectories. Wind data are based on sampled WRF-simulated clusters (Section 2). Wind speed magnitude (a), and hodograph (top view) of wind velocity up to 1000 m (c). The highlighted sections indicate operating wind conditions. Panel (b) and panel (d) shows the side and top view of the corresponding `awebox`-optimized trajectories. The reference wind speed in the legend is $U_{\mathrm{ref}} = \overline{U}(100 \text{ m} \leq z_{\mathrm{ref}} \leq 400 \text{ m})$. The results correspond to the time series shown in Figure 16.

---

## Author Response (AR4)

**Response to reviewers of wes-2020-120**

Markus Sommerfeld

July 19, 2022

**Author response**

Dear reviewers,

Thank you very much for your very detailed and helpful comments to our manuscript, "Offshore and onshore power curve characterization for ground-generation airborne wind energy systems", wes-2020-120.

Before implementing the requested revisions, we would like to address some of the comments and clarify the purpose of the paper and the interpretation of our optimization results. We respond below to the main questions and propose changes to the manuscript. We would very much appreciate confirming if we have interpreted the comments correctly and that the proposed revisions will be acceptable before investing further effort.

The manuscript has already been through several rounds of major revisions by multiple reviewers and the editor since its initial draft more than 2 years ago. Most of the original code and result files were lost in a computer crash during my relocation to a new country. Much of the code had to be re-written and the computationally expensive and time consuming optimizations had to be re-run. As such, we would like to avoid running additional optimizations, both because of the significant afford that is associated with additional optimizations and post-processing now that the original project is complete and I have graduated, and because we feel that with proper re-framing and scoping the work your concerns can be addressed by the existing data sets. We therefore propose to focus on implementing changes to the manuscript text, the QSM model and figures, as detailed in the following specific suggested changes.

Sincerely, Markus Sommerfeld

**1   Reviewer 1**

Reviewer 1 rejects the paper after multiple rounds of major revisions by several reviewers and the editor. Most their critique focuses on the estimation of AEP from WRF-derived wind data and awebox power optimizations. They believe that the difference in estimated AEP shows that the methodology fails to adequately describe the wind profiles and power output with a single power curve. They demand more validation via numerical integration and ask for justification for using WRF data, as well as trade-offs between WRF and wind atlas data. They also state that simulating every wind speed profile in the entire data set has been done by Malz et al. and is therefore feasible and should be used as validation.

We argue that we need not justify one choice of wind data over the other in the context of this paper. Wind atlases cover a longer time period and larger geographical area, but the coarser temporal and spatial grid averages out the wind speed profiles relative to our WRF data set. The purpose of this paper is not founded on a comparison between wind data sets. In fact, wind atlas data or indeed physical measurements or any other wind data set can utilize the proposed method of clustering and simulation to derive AWES performance. The focus of this work is to investigate AWES performance subject to realistic, high resolution wind data, regardless of the data source and formulated to enable more accurate while still computationally tractable computations. Since long-term LiDAR measurements were not available, we (in cooperation with the co-author meteorologists) chose WRF because it gives us 10-min average wind speed profiles with sufficient vertical profile resolution. Our collaborating meteorologists and researchers at the University of Oldenburg confirmed that this is an appropriate tool to derive realistic, high resolution wind speed profiles in case measurements are not available. To be consistent, we then used the same wind data set to derive a wind speed probability distribution and estimated AEP from it. The purpose of WRF is not to derive a simple Weibull distribution for wind speeds from which to then to estimate AEP. The point is to use WRF directly as inputs to an analysis of realistic AWES performance. Indeed, the New European Wind Atlas (`https://map.neweuropeanwindatlas.eu/about`) is based on WRF simulations and therefore consistent with our approach.

We propose to fully clarify the main research question at the core of this manuscript, modify the paper title and justify the usage of WRF. We propose to rework the manuscript so that *Pattern Trajectory Height* (see `https://airbornewindeurope.org/resources/glossary-2/`) is the base case reference for all the other power and AEP estimates, based on different reference heights $z = 100m$ and $100 \leq z_{ref} \leq 400m$. We will modify the QSM model to include reel-in power losses in the QSM, by assuming reel-in speed and minimum tether tension (maybe $F_{tether} = 0$), to reduce the power and AEP difference between the QSM and awebox model.

We simply cannot numerically integrate simulated performance for wind speed profile in our data set to validate the results. This would entail computing 105,120 10-min wind speed profiles to get 1 year of performance data at both onshore and offshore location ($2 \times 6 \times 24 \times 365$). Our awebox simulations (including trajectory optimization) require 10 minutes computation time per profile, so it would take more than 2 years (for 1 design). We note that Malz was able to compute performance hourly as her model was significantly computationally cheaper, given the model assumptions and most importantly her focus on fly-gen systems which simply simple circular orbits, rather than the much more complex real-out and real-in pumping cycles we focused on. We chose to focus on the later given that the preponderance of commercial concepts are pumping mode, and to provide a differentiation to Malz's work. It should be noted that we did carry out a convergence study, wherein we found that increasing the number of profiles (by increasing the number of clusters), and/or taking more samples per cluster, does not significantly change the AEP. We will clarify this convergence analysis in the paper, as it provides statistical evidence of the method's efficacy.

**1.1 Specific comment 1**

Specific comments not addressed here will be implemented in the revision.
RC1: first comment by reviewer
AR1: first response by author
RC2: second comment by reviewer
AR2: (new) second response by author

RC1 I find it unsatisfying that there is no justification/discussion on the degree of simplifications needed to map the high-fidelity model output to wind statistics and power curve as single argument functions (height-range-averaged wind speed).

AR1 This comment is unclear, and could be directed at any wind energy converter, conventional or AWES. The same simplification is done for every application of the Weibull distribution or simple power curve derivation for conventional wind turbines. Wind shear and other input wind field non-uniformity, unsteady operations, controller actions and like all yield in reality a cloud of binned performance points over the wind speed range that are then described by a best-fit curve. They also simplify complex wind conditions to a simple distribution and ignore the details of the variation of wind speed along the rotor diameter. Our analysis is therefore consistent in approach to standard practice, but with the intent to embed the impacts of realistic wind profiles on performance.

RC2 Can you cite any work on conventional WTs that use mesoscale simulations to find a Weibull distribution to finally arrive at the AEP calculation?

AR2 Is your objection that we should not use mesoscale simulations to estimate a power curve or AEP at all? Is your objection against averaging out wind profiles to hub-height? Is your objection against using higher resolution wind data to derive wind speed probability distributions or use them to estimate wind power? Standard wind energy practise is to take an OEM power curve and integrated that with various sources of wind data include correlated wind speed histories, mesoscale reanalysis models, etc.

AR2 Here are some citations that use WRF [4, 8, 6, 2, 9, 5, 3]. These are using mesoscale (WRF) simulations to derive Weibull parameters; the extension to AEP prediction is available in any wind textbook, e.g. Wind Energy Handbook by [1].

AR2 To your original comment: We do not see why we need to justify mapping high-fidelity model output to wind statistics and power curve as single argument functions (height-range-averaged wind speed). Using a single wind speed (e.g. hub-height) is common for conventional WTs. Any wind data, measurements or simulation, can and are mapped to wind statistics and a single reference height for power curve description. In [7] you also simplify wind speed data from the Dutch Offshore Wind Atlas data, which is based on numerical weather model HARMONIE very similar to WRF, to a single reference height: 100m.

AR2 We do not believe that this comment needs to be addressed.

RC1 I would expect that the accuracy benefits of the relatively high fidelity (and computational costly) models in the first computational steps, cancel out when the author simplifies them before calculating AEP.

AR1 See previous answer. These simplifications are done every day when applying wind statistics. They have been made to generate easily understandable, comparable to conventional power curves and AEP estimates. The power curves for conventional WT use very expensive aeroelastic simulations, and later field test data, to produce 'simplified' binned power curves which are then used with Weibul curves to produce AEP estimates.

RC2 I understand your motivation (to simplify high fidelity and computational costly data). What I question is the justification/validation. I believe more proof is needed to increase confidence in the results.

AR2 Do we understand you correctly, that your objection is not against using WRF data to derive a wind speed distribution and power estimates, but rather against using a mesoscale simulation at all to perform this task, while other data , e.g. wind atlas is available? The New European Wind Atlas (`https://map.neweuropeanwindatlas.eu/about`) is based on WRF simulations.

AR2 Yes, information is lost in the process of deriving a power curve or AEP. But this high fidelity wind data was never the end goal of this investigation, but rather means to an end. The purpose of WRF is not to derive a simple Weibull distribution from it to estimate AEP. The point is to use it to generate realistic AWES performance which we would not get from temporally and spatially averaged data. We use WRF because we cooperated with meteorologists and researchers at the university of Oldenburg who confirmed that this is an appropriate tool to derive realistic, high resolution wind speed profiles in case measurements are not available. See also comments above on ability computationally to do a full year worth of 10 min simulations. It should also be noted that conventional wind aeroelastic analysis IEC standards requires only a limited number (6-10 per wind speed bin) worth of calculations for performance prediction. Granted they are unsteady turbulent ones, but more aimed at loads verification than power prediction. Future work in AWES will have to perform similar turbulent/unsteady calculations, which will be even more computationally hard, and our approach is a first step in that direction to use clustering to rationally down-select a subset of required simulation input conditions.

AR2 We propose to point out in the revised paper that the purpose of using WRF is to derive realistic wind conditions to investigate realistic performance of AWES. We could also rename AEP to something like "estimated annual energy of this particular year" to differentiate it from long term AEP estimations.

RC1 I understand that the goal is to get a simple characterization of the power output/AEP similar to that of a WT. However, how much is this worth when you loose precious details in the process given that you went through all the effort of setting up the suggested high-fidelity tool chain?

AR1 This is a different application of the data. I do not think that I lost these details. We can still investigate the time series data or statistics if we want to, what we did in Sub-section 5.1 and 5.2. Other analysis are possible, but beyond the scope of this paper. – The description of power curve and AEP are supposed to be simple to be similar to that of conventional WT.

RC2 I meant details on the connection between the wind profiles and AEP. E.g.: how much of the AEP is attributed to certain wind conditions? This information is lost when casting the WRF results in the probability distributions. - That would be convenient. However, I would argue that more importantly they should provide a reasonable AEP prediction. This is definitely an interesting analysis, but the investigation of specific clusters/shapes on power output AEP is beyond the scope of this paper.

AR2 We propose to added this to future work.

RC1 After reading this section, I did not have a good understanding of how the p5/p50/p95 cluster profiles are used. I would argue that this is the most important part of the paper and therefore the approach taken there should be presented unambiguously.

Author Already we have implemented a clarification in sub-section Wind profile model and agree that if it was not clear before, this is important that ready have the correct statistical understanding of the method.

RC2 "From these sorted wind profiles, the 5th, 50th and 95th percentile profile are chosen and assumed to be representative of the spectrum of wind conditions within this cluster" - The clarification is still missing important details: Why 3 profiles? Representative how? Provide more arguments.

AR2 The profiles within each cluster are similar, by definition of the clustering algorithm. Similar in the sense of euclidean distance between profile velocities across height. To represent the variation within each cluster we chose high (P95), low(P5) and median(P50) wind speed profiles. This also reduces the computational cost. They are assumed to be representative of the wind conditions of this cluster, and including additional profiles in each cluster for analysis was not found to affect the results. We did not choose extreme cases because they will not represent the cluster as well as they are very rare and would only minimally contribute to overall integrated energy production. We also choose not to use the cluster centroid, because it is an averaged profile across all profiles in the cluster and not an actual profile seen in the wind speed data; the centroid is therefore potentially non-physical. The 3 profiles per cluster are directly implemented into the optimization as boundary conditions to derive power estimates. The entire point of clustering is to reduce the number of required profiles to get meaningfulness performance estimates.

AR2 We propose to clarify this in the text in Section "Wind conditions".

RC1 Which of the data points represent p5/50/95 profiles in fig 12?

AR1 Figure 12 shows operating heights for 3(3 profiles within each cluster)x20(number of clusters)=60 profiles . Therefore, it is not possible to indicate the percentile of each profile. p5 of a high wind speed cluster might be close to p95 of a low wind speed cluster.

RC2 You could distinguish with different markers. Same for the power points.

AR2 To get any information out of this you would need to indicate not only the P-value, but also the corresponding cluster. We feel this would make everything more confusing to the reader. The point of this figure is not to investigate the individual cluster, but the overall performance relative to reference wind speed, which is an easier to understand metric than clustered

percentile profile. Again, the point is not to investigate the clustering. The point is to use clustering to get representative profiles to get meaningful results with reduced computational cost.

AR2 We will clarify in the text the intent of presenting the clusters together in the figures vs identifying individual profiles. We will also include the actual profiles in online appendix data sets, should future researchers care to investigate further.

RC2 Regarding Fig 13: It's common practice in preliminary AEP calculations for WTs though it relies on quite big assumptions. Also I would say your method is quite different; in particular how you derive the probability functions. How do you come to the probability distributions? Do you use curve fitting again? Is the area underneath even 1?

AR2 Yes, the area underneath is 1. You can easily assess this by looking at the bar plot.

AR2 We propose to clarify in the text. The wind speed distribution of $100 \leq z \leq 400m$ is the distribution of average wind speed within this range (1st calc average of every profile within height range, 2nd derive distribution). The wind speed distribution at operating height is derived from the wind speeds along the flight trajectory of the simulated profiles.

**2 Reviewer 2**

Reviewer 2 asks for major revisions such as using a wind speed multiplier for each shape and normalizing the wind speed profiles. They would like discussion section amended and miss a discussion of the power curve and its definition in the conclusion. Most of the comments can be addressed by changing the wording and clarifying the text.

We do agree that normalizing the wind speed profiles and determining wind speed multipliers is a very useful contribution to apply the approach to a generalized site. Unfortunately, this is beyond the scope of this already quite lengthy paper. We propose to add this to the future works section as part of a general extension of the work for the purposes of long-term AEP assessment. We will add the requested reference and address the comments made directly in the PDF.

**2.1 Specific comment 1**

- Why are only calculated profiles used? As far as I know, the sites and times are correlating with LIDAR measurement campaigns.

  – Any kind of wind data can be used with this methodology. LiDAR measurements are available for the onshore location, but high quality data was only 6 months. WRF has the benefit that there is no missing data (e.g. power outage, low data availability / quality). We would have preferred using LiDAR data, but it was not possible to do an investigation of annual AWES performance. In any case, this type of re-analysis data is commonly used in practice. Also for a real site, only a specific location of data would be available, whereas an AWES wind farm would be operating over a fairly broad area and require the use of such reanalysis data in layout planning.

- Is it true that the QSM is neglecting force constraints and the retraction phase? Because they can be easily incorporated

  – Force constraints are used for the QSM. Could you please point out why you think they would not be, i.e What line? So that we can clarify.

  – We will include reel-in power losses in the QSM, by assuming reel-in speed and minimum tether tension ($F_{tether} \approx 0$), to reduce the power and AEP difference between the QSM and awebox model

- For the logarithmic optimization runs: Maybe 2m/s steps at 10m reference height is a bit rough

  – We see your point, but I want to avoid re-running optimization and post-processing the data. This choice was made, because we are investigating many different configurations and running one optimization takes more than 10 minutes, so practical computational limitations were taken in account in the study design.

- The implementation of Lagrange polynomials into the OCP and the difference between the grey and colored lines in Fig 11 and FigA3

  – The Lagrange polynomials actually go through all the implemented data points and deviate in between these points. Not every data point could be implemented, because a high number of data points leads to over-fitting (see Figure 2:

https://en.wikipedia.org/wiki/Overfitting and oscillation between data points. Therefore, the number of data points was reduced, which is why there is a small difference between the red and grey line. We propose to add this explanation to the text.

**References**

[1] T. Burton, D. Sharpe, N. Jenkins, and E. Bossanyi. *Wind Energy Handbook*. John Wiley & Sons, Inc., 2001.

[2] Gabriel Cuevas-Figueroa, Peter K. Stansby, and Timothy Stallard. Accuracy of wrf for prediction of operational wind farm data and assessment of influence of upwind farms on power production. *Energy*, 254:124362, 2022.

[3] Kunal K. Dayal, Gilles Bellon, John E. Cater, Michael J. Kingan, and Rajnish N. Sharma. High-resolution mesoscale wind-resource assessment of Fiji using the Weather Research and Forecasting (WRF) model. *Energy*, 232:121047, 2021.

[4] J. Díaz-Fernández, P. Bolgiani, D. Santos-Muñoz, L. Quitián-Hernández, M. Sastre, F. Valero, J.I. Farrán, J.J. González-Alemán, and M.L. Martín. Comparison of the wrf and harmonie models ability for mountain wave warnings. *Atmospheric Research*, 265:105890, 2022.

[5] Sven-Erik Gryning, Ekaterina Batchvarova, Rogier Floors, Alfredo Peña, Burghard Brümmer, Andrea N. Hahmann, and Torben Mikkelsen. Long-Term Profiles of Wind and Weibull Distribution Parameters up to 600 m in a Rural Coastal and an Inland Suburban Area. *Boundary-Layer Meteorology*, 150(2):167–184, 2014.

[6] A. Peña, K. Schaldemose Hansen, S. Ott, and M. P. van der Laan. On wake modeling, wind-farm gradients, and aep predictions at the anholt wind farm. *Wind Energy Science*, 3(1):191–202, 2018.

[7] M. Schelbergen, P. C. Kalverla, R. Schmehl, and S. J. Watson. Clustering wind profile shapes to estimate airborne wind energy production. *Wind Energy Science*, 5(3):1097–1120, 2020.

[8] Tija Sile, Andrea N. Hahmann, Bjorn Witha, Martin Dorenkamper, Magnus Baltscheffsky, and Stefan Soderberg. Applying numerical weather prediction models to the production of new european wind atlas: Sensitivity studies of the wind climate to the planetary boundary layer parametrization. In *2018 IEEE 59th International Scientific Conference on Power and Electrical Engineering of Riga Technical University (RTUCON)*, United States, 2019. IEEE. 59th International Scientific Conference on Power and Electrical Engineering of Riga Technical University , RTUCON 2018 ; Conference date: 12-11-2018 Through 13-11-2018.

[9] Jianzhou Wang, Zhenhai Guo, and Xia Xiao. Wind power assessment based on a wrf wind simulation with developed power curve modeling methods. *Abstract and Applied Analysis*, 2014.

---

## Author Response (AR6)

**Response to reviewers of wes-2020-120**

Markus Sommerfeld

June 4, 2023

**Author response**

Dear editor,
Thank you very much for your helpful comments and proposed changes to our manuscript, "Impact of offshore and onshore wind profiles on ground generation airborne wind energy system performance", wes-2020-120. We hope that this document will satisfy your standards.
Most of the requested changes could be implemented. Additionally, some figures have been edited and an additional figure to visualize the rotation of the wind vecotr has been added.
Sincerely, Markus Sommerfeld

**0.1 Specific comments**

I implemented most of the comments. Below you find comments on those I did not or could not implement.

- did not change $\varepsilon$ with $\beta$. $\beta$ is used for the kite's sideslip angle.

- Regarding Fig 6 +7, wind profile cluster: It would be interesting to relate your analysis to the one of Schelbergen et al (2020) who used very similar locations, in the Netherlands, so not too far away. What are the similarities and what are the differences between the two offshore/onshore resource assessments, both based on k-means clustering and similar number of clusters.

  - This would be interesting, but it is beyond the scope of this paper. I am not able to implement this within my available time

- No idea what you mean by "omnidirectional operation".

  - This refers to the fact that the device is able to operate under wind from any direction, i.e. operation is not restricted to a certain direction.

- Why equal tether drag distribution leads to underestimation of drag at kite? You can explain in very few words and do not need a referemce. You can leave the reference in place (never hurts) but if you can explain somethin with a few words then do so.

  - not implemented.

- why not using a function defined in two regions to approximate $cl_L$?, a linear function (for the lower alpha range) and a quadratic function (for the upper alpha range. At the point of connection, both functions should have the same value (continuous) and the same derivative (differentiable).

  - That would be a possible alternative, but was not implemented in the simulation (5 years ago)

- Better explain $U_ref$ and $z_ref$.

  - I hope I did?

- Better explain a priori guess

  - I hope I did?
  - this a priori guess was used to select the wind velocity profiles from the clustered dataset and then kept through the study.

- What is "instantaneous performance" in the OCP context? How do you investigate this?

  - it's the timeseries results, e.g. tether tension or power during the simulated pumping cycle.

- Can you add a scale to the map?

  - I tried, but couldn't make the old code work. I hope the longitude and latitude labels on the abscissa and ordinate will suffice.

**References**